# Open Thoughts

## DATA RECIPES FOR REASONING MODELS

**Etash Guha\*[1,2], Ryan Marten\*[3], Sedrick Keh\*[4], Negin Raoof\*[5], Georgios Smyrnis\*[6],**
**Hritik Bansal[ζ7], Marianna Nezhurina[ζ8,9,16], Jean Mercat[ζ4], Trung Vu[ζ3], Zayne Sprague[ζ6],**
**Ashima Suvarna[7], Benjamin Feuer[10], Liangyu Chen[1], Zaid Khan[11], Eric Frankel[2],**
**Sachin Grover[12], Caroline Choi[1], Niklas Muennighoff[1], Shiye Su[1], Wanjia Zhao[1], John Yang[1],**
**Shreyas Pimpalgaonkar[3], Kartik Sharma[3], Charlie Cheng-Jie Ji[3], Yichuan Deng[2],**
**Sarah Pratt[2], Vivek Ramanujan[2], Jon Saad-Falcon[1], Jeffrey Li[2], Achal Dave, Alon Albalak[13],**
**Kushal Arora[4], Blake Wulfe[4], Chinmay Hegde[10], Greg Durrett[6], Sewoong Oh[2],**
**Mohit Bansal[11], Saadia Gabriel[7], Aditya Grover[7], Kai-Wei Chang[7], Vaishaal Shankar,**
**Aaron Gokaslan[14], Mike A. Merrill[1], Tatsunori Hashimoto[1], Yejin Choi[1],**
**Jenia Jitsev[8,9,16], Reinhard Heckel[15], Maheswaran Sathiamoorthy[3],**
**Alexandros G. Dimakis[†3,5], Ludwig Schmidt[†1]**

[1]Stanford University, [2]University of Washington, [3]BespokeLabs.ai, [4]Toyota Research Institute,
[5]UC Berkeley, [6]UT Austin, [7]UCLA, [8]JSC, [9]LAION, [10]NYU, [11]UNC Chapel Hill,
[12]ASU, [13]Lila Sciences, [14]Cornell Tech [15]TUM [16]Open-Ψ (Open-Sci) Collective

## ABSTRACT

Reasoning models have made rapid progress on many benchmarks involving math, code, and science. Yet, there are still many open questions about the best training recipes for reasoning since state-of-the-art models often rely on proprietary datasets with little to no public information available. To address this, the goal of the OpenThoughts project is to create open-source datasets for training reasoning models. Our OpenThoughts2-1M dataset led to OpenThinker2-32B, the first model trained on public reasoning data to match DeepSeek-R1-Distill-32B on standard reasoning benchmarks such as AIME and LiveCodeBench. We then improve our dataset further by systematically investigating each step of our data generation pipeline with 1,000+ controlled experiments, which led to OpenThoughts3. Scaling the pipeline to 1.2M examples and using QwQ-32B as teacher yields our OpenThinker3-7B model, which achieves state-of-the-art results: **53%** on AIME 2025, **51%** on LiveCodeBench 06/24-01/25, and **54%** on GPQA Diamond – improvements of 15.3, 17.2, and 20.5 percentage points compared to the DeepSeek-R1-Distill-Qwen-7B. All of our datasets and models are available on openthoughts.ai.

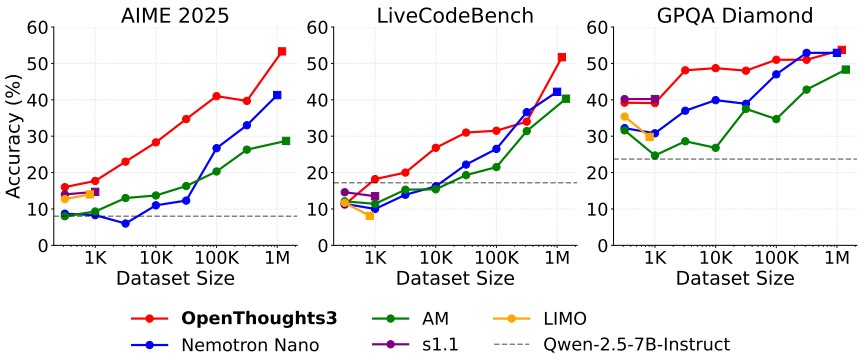

Figure 1: **OpenThoughts3 outperforms existing SFT reasoning datasets across data scales.** All models are finetuned from Qwen-2.5-7B-Instruct. We compare to large SFT datasets (AM, Nemotron Nano) and small curated datasets (s1.1, LIMO) on AIME 2025 (left), LiveCodeBench 06/24-01/25 (middle), and GPQA Diamond (right). Scaling curves for all evaluation benchmarks are in Figure 6.

---

\*,† denote equal contribution. ζ denotes additional core contributors. The order is determined randomly.

| Benchmark | OpenThinker3-7B | DS-R1-Qwen-7B | NemoNano-1M | AM-1.4M | OpenR1-Distill-7B | Nemotron-Nano-8B | AceReason-7B | Skywork-7B | Qwen2.5-7B-Instruct |
|---|---|---|---|---|---|---|---|---|---|
| Base Model | 🌀 | 🌀 M | 🌀 | 🌀 | 🌀 M | ∞ | 🐳 | 🐳 | N/A |
| Train Size | 1.2M | 800K | 1M | 1.4M | 350K | 3.9M | 57K | 119K | N/A |
| Method | SFT | SFT | SFT | SFT | SFT | SFT/RL | RL | RL | N/A |
| Trained by us | Yes | No | Yes | Yes | No | No | No | No | N/A |
| Open Data | 🟢 | 🔴 | 🟢 | 🟢 | 🟢 | 🟢 | 🟡 | 🟡 | N/A |
| Average | **55.3** | 42.9 | 47.3 | 42.1 | 47.2 | 53.2 | 52.9 | 51.6 | 24.0 |
| *Math* AIME24 | **69.0** | 51.3 | 55.0 | 28.3 | 57.7 | 62.0 | **71.0** | **68.3** | 15.0 |
| AMC23 | **93.5** | 92.0 | 87.0 | 82.2 | 87.0 | **94.0** | **93.8** | 91.0 | 53.0 |
| MATH500 | 90.0 | 88.0 | 86.8 | 87.4 | 88.0 | 89.4 | 89.8 | **90.2** | 70.8 |
| *Code* CodeElo | 31.0 | 19.9 | 28.6 | 21.0 | 30.1 | 30.9 | 32.9 | **37.0** | 5.5 |
| LCB 05/23-05/24 | 64.5 | 48.7 | 58.0 | 54.5 | 37.9 | **68.0** | 60.5 | 60.4 | 36.2 |
| CodeForces | **32.2** | 21.1 | 28.3 | 24.8 | 29.3 | **32.9** | 30.9 | **32.5** | 10.2 |
| *Sci* GPQA-D | 53.7 | 33.2 | 52.9 | 48.3 | **58.9** | 52.9 | 52.9 | 50.2 | 24.6 |
| JEEBench | **72.4** | 50.4 | 61.0 | 61.1 | 68.7 | 70.7 | 64.3 | 55.3 | 33.9 |
| *Held Out* HMMT 02/25 | **42.7** | 25.0 | 24.7 | 19.0 | 25.7 | 26.7 | 33.3 | 32.7 | 2.0 |
| HLE MCQ | 10.2 | 12.4 | 2.1 | 9.5 | 12.4 | 12.0 | 10.9 | 10.7 | **12.7** |
| AIME25 | **53.3** | 38.0 | 41.3 | 28.7 | 39.7 | 48.0 | 50.7 | 47.3 | 8.0 |
| LCB 06/24-01/25 | **51.7** | 34.5 | 42.2 | 40.3 | 30.7 | **50.9** | 44.3 | 43.8 | 16.3 |

Table 1: **OpenThinker3-7B outperforms all open-data 7B and 8B reasoning models across domains.** Our model also performs well on held out benchmarks which are not measured during our main experimentation, such as HMMT and AIME25. In our table, 🌀 denotes a model trained from Qwen-2.5-7B-Instruct, 🌀M for Qwen-2.5-Math-Base, ∞ for Llama-3.1-8B-Instruct, and 🐳 for DeepSeek-R1-Distill-Qwen-7B. "Base Model" denotes the starting checkpoint of the training strategy. "Method" denotes the model's optimization algorithm. In each row, we bold values within two standard errors of the highest-scoring model.

# 1 INTRODUCTION

Recent models, such as DeepSeek-R1 (Guo et al., 2025) and o3 (OpenAI, 2024), have demonstrated strong performance in reasoning-based domains, including math, coding, and science. These models often start from a strong base model, then introduce reasoning capabilities through a series of post-training techniques like supervised finetuning (SFT) or reinforcement learning (RL). This post-training process equips these models with the ability to output long chains of thought, or "thinking tokens," during inference time, which can guide the model toward the correct answer. Yet, the complete recipes for frontier reasoning models are not public, making research for building reasoning models difficult.

Innovating on SFT data curation is a powerful method for building reasoning models (Abdin et al., 2024; Lin et al., 2024). For instance, the R1-Distill models show that it is possible to get state-of-the-art small- to mid-scale reasoning models, with performance of 51% on AIME and 33% on GPQA, without any RL steps, based only on supervised fine-tuning on a large, carefully curated dataset of question-thinking tokens-answer triplets, where the thinking tokens and answers are generated using a reasoning teacher model or are taken from the real data that contains reasoning traces. Existing works, such as SkyT1 (NovaSky-Team, 2025b) and S1 (Muennighoff et al., 2025), adopt nearly identical model architectures and training setups as typical instruction tuning, yet still achieve performance improvements by focusing on improving the training datasets. These examples highlight the importance of curating high-quality SFT data as a key lever for reasoning performance.

Most of these projects, however, explore only a limited fraction of possible design choices, such as relying on human-written questions or using DeepSeek-R1 as the sole teacher. Recreating reasoning models requires exploring a large design space for various strategies of generating question-answer pairs for reasoning (Face, 2025). This exploration is prohibitively expensive for many researchers due to the high costs of teacher inference and model training. In the absence of these expensive experiments, many papers rely on existing heuristics and intuitions to inform their data design choices.

To demystify the SFT data curation process and gain a deeper understanding of what contributes to a strong reasoning SFT dataset, we conduct an empirical investigation into data curation techniques for improving reasoning capabilities. Through more than **1,000** ablation experiments across three data domains (math, code, and science), we develop a simple, scalable, and highly performant pipeline, as shown in Figure 2. We scale up this pipeline to produce **OpenThoughts3-1.2M**, and fine-tune Qwen2.5-7B-Instruct on it to yield **OpenThinker3-7B**. As seen in Table 1, the resulting OpenThinker3-7B is a state-of-the-art open-data model at the 7B scale on several reasoning benchmarks, outperforming the R1-Distill-7B model by 12.4 points on average across 12 tasks, and outperforming the next-best open-data model (Nemotron-Nano-8B) by 2.1 points. We release our models and data artifacts to the community and share several key findings from our study. Some of our insights include: (1) Sampling multiple answers per question from a teacher model is an effective technique to increase the size of a data source by at least $16\times$. The increased dataset scale drives significant performance gains. (2) Models with better performance are not necessarily better teachers. QwQ-32B is a stronger teacher than DeepSeek-R1, although it scores lower on target reasoning benchmarks. (3) We experimented with numerous verification and answer filtering methods, and none gave significant performance improvements. (4) Selecting questions from a small number (top 1 or 2) of high-quality sources leads to better downstream performance compared to optimizing for diversity (i.e., top 8 or 16 sources). (5) Filtering questions by LLM labeled difficulty or LLM response length yields better results than filters typical to pre-training data curation using embeddings or `fastText`.

## 2  RELATED WORK

The release of models such as Gemini (Gemini-Team et al., 2023), QwQ (Qwen-Team, 2025), and DeepSeek-R1 (Guo et al., 2025), which made long reasoning traces visible to users, opened the possibility of training small models via the distillation of traces from larger ones. DeepSeek released strong distilled models together with DeepSeek-R1 (e.g., DeepSeek-R1-Distill-Qwen-7B), showing how promising this strategy can be. Following this, many open-data efforts have attempted to replicate these models by building SFT reasoning datasets through distillation from teacher models such as QwQ-32B (NovaSky-Team, 2025b) or DeepSeek-R1 (Bespoke-Labs, 2025).

Many datasets target math, code, and science to develop reasoning capabilities. Datasets such as OpenR1(Face, 2025), OpenMathReasoning (Moshkov et al., 2025), and OpenCodeReasoning (Ahmad et al., 2025b) collect questions from public forums and competition sites like CodeForces, AoPS, and StackOverflow, while others like Natural Reasoning (Yuan et al., 2025) use large pre-training corpora as seed data for generating reasoning traces. Efforts like S1 (Muennighoff et al., 2025) and LIMO (Ye et al., 2025) emphasize manual curation of small datasets of challenging, high-quality prompts. In practice, many reasoning projects (e.g., DeepMath-103K (He et al., 2025c), OpenR1 (Face, 2025), and Nvidia Nemotron (Adler et al., 2024)) introduce innovations across multiple stages, such as data sourcing, filtering, and scaling. Beyond SFT, works such as AceReason (Chen et al., 2025) and Skywork-OR1 (He et al., 2025a) build reasoning datasets for reinforcement learning.

## 3  OPENTHOUGHTS3 DATA PIPELINE

### 3.1  EXPERIMENTAL SETUP

**Training**   Our goal is to create the best dataset of question-response pairs for SFT reasoning. The best dataset is the one that produces the highest-performing model. To approach this systematically, we ablate each step of our pipeline individually, isolating the effect of a given strategy while keeping the rest of the pipeline constant. For each experiment, we utilize the full pipeline to generate 31,600 data points for each data strategy, and we finetune Qwen2.5-7B-Instruct (Yang et al., 2024b) on each dataset. Our experiments are conducted at a dataset scale that is small enough to be cost-effective yet

Figure 2: **The OpenThoughts experiment pipeline aims to build the strongest reasoning dataset recipe.** We investigate (1) sourcing questions from existing and newly generated datasets, (2) mixing questions from the top-performing sources, (3) filtering for high-quality questions using `fastText` or LLMs, (4) deduplicating questions and sampling multiple answers per question, (5) filtering out low-quality answers using LLM verification or majority consensus, and (6) selecting the best teacher model.

large enough to provide a meaningful signal. We choose 31,600 as a log-scale midpoint between 10K and 100K, as $\sqrt{10} \approx 3.16$. These experiments inform the design choices for the final OpenThoughts3 pipeline. Appendix A contains details on hyperparameters and training setup.

**Evaluation Setup** We evaluate our models on a set of reasoning benchmarks containing math, code, and science questions. Per domain, these benchmarks are: AIME24 (MAA, 2024), AMC23 (MAA, 2023) and MATH500 (Hendrycks et al., 2021b) for math; CodeElo (Quan et al., 2025), CodeForces (Penedo et al., 2025a), and LiveCodeBench 05/23-05/24 (Jain et al., 2024) for code; GPQA Diamond (Rein et al., 2024) and JEEBench (Arora et al., 2023) for science. We score each model based on average performance on these eight tasks. Evalchemy (Raoof et al., 2025) is our primary evaluation tool, and we use the default setup provided for each benchmark. Further details on evaluation setup are in Appendix B. We also decontaminate our datasets against our benchmarks by removing samples with high similarity. Details for this process are in Appendix C. To measure generalization, our pipeline experiments exclude a held out set of benchmarks, which are only measured once pipeline experiments are over. This held out set consists of AIME 2025 (MAA, 2025), HMMT 02/25 (Balunović et al., 2025), Humanity's Last Exam (multiple choice questions subset) (Phan et al., 2025), and LiveCodeBench 06/24-01/25 (Jain et al., 2024).

**Pipeline** At each pipeline step, we select the top-performing approach based on the average benchmark score across all domains, and then proceed to the next step in the pipeline experimentation with this selection. DeepSeek-R1 is the default teacher model unless specified otherwise.

## 3.2 QUESTION SOURCING

The first step in our data generation pipeline is finding questions for each data domain. We can broadly categorize our question sourcing techniques into three types: (1) Fully synthetic – an existing LLM generates questions with little-to-no seed material. Examples include CodeAlpaca (Chaudhary, 2023) and CamelChemistry (Li et al., 2023a). These often involve prompting an LLM with a template to generate multiple questions. (2) Semi-synthetic – an LLM uses existing data sources such as CommonCrawl or FineWeb (Penedo et al., 2024a) as seeds to form questions. Examples include TigerLabMath (Yue et al., 2024) and AutoMathText (Zhang et al., 2024b). (3) Non-synthetic – humans write the questions. Examples include StackExchange and ShareGPTCode. These questions often arise from online forums, contests, chatbot interactions, and other sources.

Our experiments cover 27 different question sources for code questions, 21 sources for math, and 14 sources for science. The details of these sources are in Appendix O.1. The first step of our ablation is to generate 31,600 questions using each source. For sources that produce fewer datapoints, we repeat the questions until we reach the desired amount. We use GPT-4o-mini for all sources that we generate which require an LLM. Finally, we use DeepSeek-R1 to generate responses for each question, even if a pre-existing answer exists.

The experimental results are in Table 2. For code, CodeGolf questions from StackExchange and competitive coding questions from OpenCodeReasoning (Ahmad et al., 2025b) perform well, achieving scores of 25.3 and 27.5 on average on code benchmarks. For math, both LLM-generated questions in openmath-2-math (Toshniwal et al., 2024a) and human-written questions in NuminaMath (LI et al.,

| SFT Datasets | | | Benchmarks | | |
|---|---|---|---|---|---|
| Code Question Source | | Average | Code Avg | Math Avg | Science Avg |
| StackExchange-CodeGolf[*] | | $\mathbf{38.8}_{0.4}$ | $25.3_{0.6}$ | $\mathbf{50.9}_{1.1}$ | $40.7_{0.5}$ |
| OpenCodeReasoning | | $\mathbf{38.4}_{0.3}$ | $\mathbf{27.5}_{0.4}$ | $47.9_{0.7}$ | $40.7_{0.6}$ |
| KodCode-V1 | | $37.7_{0.3}$ | $23.9_{0.4}$ | $\mathbf{49.8}_{0.7}$ | $40.4_{0.3}$ |
| . . . | | . . . | . . . | . . . | . . . |
| bugdaryan/sql-create-. . . | | $21.6_{0.6}$ | $7.0_{0.7}$ | $34.1_{1.4}$ | $24.7_{0.9}$ |
| Math Question Source | | Average | Code Avg | Math Avg | Science Avg |
| OpenMath-2-Math | | $\mathbf{38.1}_{0.3}$ | $\mathbf{12.4}_{0.2}$ | $\mathbf{58.8}_{1.0}$ | $\mathbf{45.6}_{0.2}$ |
| NuminaMath-1.5 | | $37.4_{0.5}$ | $11.4_{0.5}$ | $\mathbf{58.5}_{1.0}$ | $\mathbf{45.0}_{1.2}$ |
| MathPile[*] | | $36.2_{0.5}$ | $11.5_{0.7}$ | $55.1_{0.9}$ | $44.6_{1.1}$ |
| . . . | | . . . | . . . | . . . | . . . |
| Lap1official/Math[*] | | $24.4_{0.3}$ | $7.3_{0.3}$ | $38.6_{1.0}$ | $28.5_{0.3}$ |
| Science Question Source | | Average | Code Avg | Math Avg | Science Avg |
| StackExchange-Physics[*] | | $\mathbf{34.3}_{0.4}$ | $\mathbf{11.9}_{0.5}$ | $50.9_{0.8}$ | $43.2_{0.7}$ |
| OrganicChemistry-PDF[*] | | $\mathbf{34.0}_{0.3}$ | $8.4_{0.3}$ | $\mathbf{52.1}_{0.7}$ | $\mathbf{45.3}_{0.8}$ |
| CQADupStack-Physics | | $33.3_{0.4}$ | $7.4_{0.3}$ | $\mathbf{51.9}_{1.1}$ | $\mathbf{44.1}_{0.9}$ |
| . . . | | . . . | . . . | . . . | . . . |
| AdapterOcean/biology_dataset. . . | | $21.9_{0.4}$ | $3.1_{0.3}$ | $41.3_{1.1}$ | $21.1_{0.8}$ |

Table 2: **Evaluating question sources and generation strategies.** We show only the top 3 scoring sources for each domain; descriptions of each source are in Appendix O.1 and full results are in Tables 31 to 33. Each row represents a unique source of questions. Question quality greatly affects performance, yielding a 17.2 gap between the strongest and the weakest code datasets. The * symbol denotes a new dataset we created with a programmatic generation strategy. Gray subscripts represent standard errors, and we bold values within two standard errors of the highest-scoring data strategy.

| SFT Datasets | | | Benchmarks | | |
|---|---|---|---|---|---|
| Code Question Mixing Strategy | | Average | Code Avg | Math Avg | Science Avg |
| Top 1 Code Sources | | $39.9_{0.6}$ | $23.1_{1.0}$ | $\mathbf{54.5}_{0.8}$ | $\mathbf{43.1}_{1.2}$ |
| Top 2 Code Sources | | $\mathbf{41.3}_{0.4}$ | $\mathbf{27.3}_{0.3}$ | $\mathbf{54.7}_{0.9}$ | $\mathbf{42.1}_{1.0}$ |
| Top 4 Code Sources | | $38.6_{0.4}$ | $24.2_{0.6}$ | $52.2_{0.8}$ | $39.8_{0.9}$ |
| Top 8 Code Sources | | $37.0_{0.4}$ | $21.8_{0.3}$ | $51.9_{1.2}$ | $37.7_{0.6}$ |
| Top 16 Code Sources | | $36.4_{0.4}$ | $20.8_{0.4}$ | $50.1_{0.9}$ | $39.1_{1.0}$ |

Table 3: **Mixing different code question sources**. Our experiments show that choosing only the two best question sources outperforms mixing more question sources. Similar results hold for science and math data domains. Full results including the math and science datasets are in Tables 34 to 36.

2024) score the highest, achieving 58.8 and 58.5 on average across math benchmarks. Lastly, for science, the highest-scoring question generation strategies are physics questions from StackExchange and LLM-extracted questions from organic chemistry textbooks, which achieve an average score of 43.2 and 45.3, respectively, on science benchmarks. No clear pattern emerges across question generation strategies – simple synthetic methods perform comparably to, and occasionally better than, more complex or manually curated pipelines. These top-performing question sources provide the foundation for subsequent stages of the pipeline.

| SFT Datasets | | | Benchmarks | | |
| --- | --- | --- | --- | --- | --- |
| Math Question Filtering Strategy | Average | Code Avg | Math Avg | Science Avg |
| Response Length Selection (GPT-4.1-mini) | $\mathbf{41.9}_{0.3}$ | $\mathbf{13.4}_{0.3}$ | $\mathbf{66.0}_{0.8}$ | $\mathbf{48.6}_{0.4}$ |
| Response Length Selection (GPT-4.1-nano) | $39.4_{0.3}$ | $11.0_{0.4}$ | $\mathbf{64.5}_{0.7}$ | $44.3_{0.7}$ |
| AskLLM Selection | $36.3_{0.4}$ | $9.5_{0.5}$ | $58.1_{1.1}$ | $43.8_{0.6}$ |
| FastText (P: Numina; N: Lap1official) | $35.6_{0.4}$ | $11.0_{0.2}$ | $54.9_{1.1}$ | $43.5_{0.8}$ |
| $\cdots$ | $\cdots$ | $\cdots$ | $\cdots$ | $\cdots$ |
| Code Question Filtering Strategy | Average | Code Avg | Math Avg | Science Avg |
| Difficulty-based Selection | $\mathbf{43.0}_{0.5}$ | $27.7_{0.4}$ | $\mathbf{56.0}_{1.3}$ | $46.4_{0.7}$ |
| Response Length Selection (GPT-4.1-nano) | $\mathbf{42.2}_{0.4}$ | $26.6_{0.5}$ | $\mathbf{55.4}_{1.3}$ | $\mathbf{46.0}_{0.2}$ |
| AskLLM Selection | $41.6_{0.5}$ | $\mathbf{28.8}_{0.5}$ | $52.1_{1.2}$ | $45.2_{0.8}$ |
| Response Length Selection (GPT-4o-mini) | $40.8_{0.5}$ | $25.6_{0.5}$ | $53.1_{0.9}$ | $\mathbf{45.2}_{1.1}$ |
| $\cdots$ | $\cdots$ | $\cdots$ | $\cdots$ | $\cdots$ |

Table 4: **Filtering questions provides an effective tool for extracting high-quality questions.** Using LLM-based methods to find the best questions from the question sources outperformed classical filtering methods such as `fastText` and embedding-based filters. This table shows the top-performing strategies. Full results including science datasets are reported in Tables 37 to 39.

## 3.3 MIXING QUESTIONS

After obtaining high-quality questions from various question sources, the challenge becomes how to combine them effectively – should we rely on a single best-performing strategy, or blend multiple strong ones to maximize downstream performance? This mixing strategy is a key design choice in many generation pipelines (Yue et al., 2024; Moshkov et al., 2025; Lambert et al., 2024). Intuitively, adding more question sources into the mix introduces the risk of incorporating lower-quality strategies in exchange for greater diversity. Our experiments aim to assess whether the additional question diversity justifies this tradeoff in terms of question quality.

For simplicity, we use the rankings of the previous step in our pipeline (Section 3.2) as a heuristic for candidate dataset selection. Our mixing strategy selects the top-ranked N datasets, randomly samples $\frac{31,600}{N}$ questions from each source, and concatenates them to form a dataset of size 31,600. We sweep values of $N \in \{1, 2, 4, 8, 16\}$. The results for the code domain are in Table 3 while the other results are in Appendix P.2. Our experiments show that mixing many question sources degrades performance: mixing at most two sources yields the best results across all data domains. Using two high-quality code question sources instead of 16 strategies results in a 5% accuracy increase on average across all benchmarks. This result indicates that downstream performance benefits from increased quality of source data rather than diversity induced by mixing multiple question sources.

> **Takeaway:** We use OpenMath-2-Math as our sole math question source, CodeGolf and Open-CodeReasoning as our code question sources, and StackExchangePhysics and OrganicChemistry-PDFs as our science question sources.

## 3.4 QUESTION FILTERING

Since each data source can contain millions of potential questions, answering and training on every possible question is infeasibly expensive. Therefore, the next step is to select a high-quality subset of questions from each source. Across the literature, a wide range of filtering strategies have consistently improved overall dataset quality (Soldaini et al., 2024; Su et al., 2024; Li et al., 2024; Wettig et al., 2025; Penedo et al., 2024a; Gao et al., 2020; Shum et al., 2025). Using the best question sources from Section 3.3 as a starting point, we extensively explore various filtering methods, including `fastText` classifiers, difficulty scores, and embedding distance, to select higher-quality questions. A detailed description of these filtering methods is in Appendix O.2. The results of these experiments for math and code are reported in Table 4 while the science results are in Appendix P.3 (Table 39).

| SFT Datasets | | Benchmarks | | |
|---|---|---|---|---|
| Science Answer Generation Strategy | Average | Code Avg | Math Avg | Science Avg |
| Exact Dedup w/ $16\times$ sampling | $\mathbf{36.2}_{0.5}$ | $9.0_{0.4}$ | $\mathbf{54.5}_{1.0}$ | $49.7_{1.2}$ |
| Fuzzy Dedup w/ $16\times$ sampling | $\mathbf{36.1}_{0.4}$ | $\mathbf{10.9}_{0.2}$ | $52.9_{1.3}$ | $48.8_{0.5}$ |
| Exact Dedup w/ $4\times$ sampling | $\mathbf{35.8}_{0.5}$ | $10.6_{0.7}$ | $51.8_{1.0}$ | $49.6_{1.2}$ |
| No Dedup w/ $4\times$ sampling | $\mathbf{35.8}_{0.4}$ | $10.0_{0.4}$ | $\mathbf{55.2}_{0.8}$ | $45.4_{0.9}$ |
| No Dedup w/ $16\times$ sampling | $35.7_{0.4}$ | $7.6_{0.5}$ | $53.8_{1.0}$ | $\mathbf{50.9}_{0.5}$ |
| No Dedup w/ $1\times$ sampling | $35.5_{0.3}$ | $9.3_{0.3}$ | $54.2_{1.1}$ | $46.9_{0.2}$ |
| Exact Dedup w/ $1\times$ sampling | $35.0_{0.4}$ | $7.6_{0.4}$ | $\mathbf{54.0}_{1.2}$ | $47.5_{0.5}$ |
| Fuzzy Dedup w/ $4\times$ sampling | $34.9_{0.4}$ | $7.4_{0.5}$ | $\mathbf{55.0}_{1.0}$ | $46.0_{0.7}$ |
| Fuzzy Dedup w/ $1\times$ sampling | $34.2_{0.3}$ | $5.8_{0.4}$ | $52.5_{0.7}$ | $49.5_{0.4}$ |

Table 5: **Deduplication and repeated teacher sampling provide an axis of scale.** Using fewer questions and annotating more times performs similarly or even outperforms annotating more questions fewer times. There does not seem to be a clear trend in types of deduplication that improve performance. Full results including math and code datasets are in Tables 40 to 42.

The two highest performing question filtering methods are difficulty-based filtering and response length filtering. Difficulty-based filtering asks an LLM (GPT-4o-mini) to assess the difficulty of each question, then retains the most difficult questions. Difficulty-based filtering is the winning strategy for code. Meanwhile, response length filtering asks an LLM to respond to each question directly, then selects the questions with the longest LLM-generated responses. Response length filtering performs the best for math and science. For math and code domains, using the best question filtering strategy resulted in an average improvement of 4% and 6% over the random filtering baseline, respectively. We test different LLMs such as GPT-4o-mini and GPT-4.1-nano for response length filtering and find that using stronger models for response length filtering typically outperforms using weaker models. For all domains, using LLM-based filtering methods outperformed classical filtering methods such as embedding-based and `fastText` filters.

> **Takeaway:** We use difficulty-based filtering with GPT-4o-mini for code questions, and response length filtering with GPT-4.1-mini for math and science questions.

## 3.5 DEDUPLICATION AND SAMPLING MULTIPLE ANSWERS PER QUESTION

Deduplication is a powerful strategy for improving dataset quality (Li et al., 2024; Lee et al., 2022; Penedo et al., 2024b; Fang et al., 2025; Liu et al., 2024). Our ablations investigate the effects of question deduplication on downstream reasoning performance. We explore three degrees of deduplication strictness: no deduplication, exact match deduplication, and fuzzy deduplication using a threshold-based string similarity. Further details are in Appendix O.3. While deduplication enhances question diversity by reducing repetition, a natural counterpart for enhancing answer diversity is to query the teacher multiple times to elicit distinct responses. This strategy trades off higher answer diversity for lower question diversity and provides another axis of data scale. We explore three levels of sampling multiple answers per question, at $1\times$, $4\times$, and $16\times$.

To address the interplay between naturally occurring duplicate questions in the source datasets and the need to query the teacher multiple times per question, we sweep all combinations of deduplication levels (none, fuzzy, exact) and sampling multiple answers ($1\times$, $4\times$, $16\times$). The results for the science domain are presented in Table 5, while the other domains' results are in Appendix P.4 (Table 40 and Table 41). For code and science data, various combinations of deduplication and multiple answer generation yield similar results. For example, the baseline of no deduplication with $1\times$ answer per question performs 0.7 points worse on average than exact deduplication with $16\times$ answers per question for the code domain. Meanwhile, for math, exact deduplication with $4\times$ answers per question performs the best, and $16\times$ answers per question is the second-best option. We adopt the second-best option moving forward, as it provides better scalability. Similar to Section 3.3, the results here indicate that the benefits of question diversity may be limited for the reasoning datasets we measure performance on, at least when answer diversity increases. Thus, for math and science, we

| Math Answer Filtering Strategy | Average | Code Avg | Math Avg | Science Avg |
|---|---|---|---|---|
| No Filtering (not compute-controlled) | **41.9**$_{0.4}$ | **15.2**$_{0.5}$ | **65.6**$_{0.9}$ | 46.4$_{0.7}$ |
| Random Filtering | **41.6**$_{0.4}$ | **14.9**$_{0.4}$ | **64.8**$_{0.9}$ | 46.7$_{0.5}$ |
| Shortest Answers Selection | **41.1**$_{0.4}$ | **14.8**$_{0.4}$ | 63.7$_{1.1}$ | 46.7$_{0.7}$ |
| Removing Non-English Answers | **41.1**$_{0.5}$ | **14.2**$_{0.5}$ | 63.1$_{1.0}$ | **48.6**$_{1.0}$ |
| . . . | . . . | . . . | . . . | . . . |
| GPT Verification | 40.0$_{0.5}$ | 13.1$_{0.3}$ | 61.4$_{1.1}$ | **48.3**$_{1.1}$ |
| Removing Long Paragraphs | 38.0$_{0.4}$ | 5.7$_{0.2}$ | **64.5**$_{0.9}$ | 46.8$_{1.0}$ |

Table 6: **Filtering answers did not improve over not filtering at all.** Using verification techniques such as majority consensus filtering or response-length-based filtering did not improve upon training on all samples. Full results including code and science datasets are in Tables 43 to 45.

| SFT Datasets | | | Benchmarks | | |
|---|---|---|---|---|---|
| Teacher for Code | Average | Code Avg | Math Avg | Science Avg |
| Qwen/QwQ-32B | **44.2**$_{0.5}$ | **29.5**$_{0.3}$ | **58.7**$_{1.1}$ | 44.6$_{1.0}$ |
| deepseek-ai/DeepSeek-R1 | 42.3$_{0.5}$ | 27.2$_{0.5}$ | 54.7$_{1.4}$ | **46.5**$_{0.3}$ |
| microsoft/Phi-4-reasoning-plus | 29.0$_{0.4}$ | 0.5$_{0.1}$ | 52.1$_{1.2}$ | 37.2$_{0.6}$ |

Table 7: **Using a weaker teacher outperformed using a stronger teacher.** Across all domains, QwQ-32B was the strongest teacher model, despite being a weaker model than DeepSeek-R1. Further results can be seen in Table 48.

select the optimal strategy, which is exact deduplication with $16\times$ answers per question. For code, we employ the second-best strategy, which involves no deduplication with $16\times$ answers per question.

> **Takeaway:** Our final pipeline uses $16\times$ answers per question for all domains. It uses exact deduplication for math and science and no deduplication for code.

## 3.6 ANSWER FILTERING

Verification or removing low-quality annotations is a common step in many reasoning data pipelines. Intuitively, removing data that may be incorrect should improve downstream performance. Our experiments explore various answer filtering techniques. To ensure that we can still obtain datasets of size 31,600 after filtering, we first generate 63,200 answers, apply each answer-filtering strategy, and then sample 31,600 question-answer pairs from the filtered dataset. Our ablations also include a baseline with no filtering, which is not compute-controlled, as it contains 63,200 questions.

Table 6 shows the result of each filtering method for math datasets, and the results for code and science datasets are shown in Appendix P.5. For math datasets, the random filtering baseline outperformed all other filtering methods. A `fastText` (Joulin et al., 2017) classifier was the best answer-filtering method for code question-answer pairs. The positives for the `fastText` classifier came from CodeForces (Penedo et al., 2025a) answered with DeepSeek-R1, and the negatives came from CodeForces answered with GPT-4o-mini. For science, keeping the top 8 longest answers was the strongest filtering strategy. However, across all domains, the no-filtering strategy (training on all samples without controlling compute) led to performance similar to that of all other methods of filtering. This result suggests that the benefits of answer filtering are not significant enough to justify reducing the number of samples in the dataset. As such, we opt to skip this part in the pipeline.

> **Takeaway:** We do not perform answer filtering because no filtering strategy outperformed the baseline, which uses all the answers.

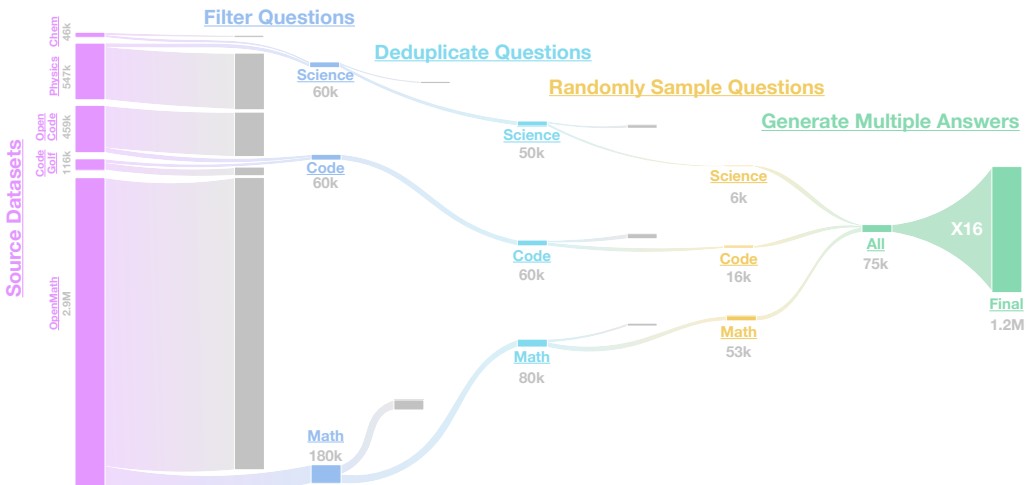

Figure 3: **The OpenThoughts3-1.2M Full Data Pipeline.**

## 3.7 TEACHER MODEL

The previous experiments have relied on using DeepSeek-R1 as a teacher model. However, there are many possible candidates for teacher reasoning models, including DeepSeek R1, Phi-4-Reasoning-Plus-14B (Abdin et al., 2025), and QwQ-32B. Our experiments measure the downstream effects of selecting different teacher models for each strategy, as described in Section 3.6. The sampling hyperparameters are kept constant across all teacher models we studied. The results of this experiment are in Table 7. Across all domains, using QwQ-32B as a teacher model outperforms all other teacher models, yielding an average accuracy improvement of 1.9% and 2.6% over using DeepSeek-R1 as a teacher for code and math, respectively. This is despite the fact that QwQ-32B scores lower on average when compared to DeepSeek-R1. For example, DeepSeek-R1 outperforms QwQ-32B by 9%, 8%, and 23% on CodeElo, GPQA Diamond, and JEEBench, respectively. A comparison of the empirical strengths of each teacher is in Table 28.

> **Takeaway:** We use QwQ-32B as the teacher model.

## 4 SCALING OUR PIPELINE TO OPENTHOUGHTS3-1.2M

Dataset scaling plays a key role in achieving strong performance. We investigate how well our pipeline scales by identifying the winning strategy in each successive pipeline step and plotting its performance from 316 to 31.6k examples. Figure 5 demonstrates that the scaling behavior improves as we successively stack the best choices from each stage in the pipeline. Additionally, Figure 5 also shows a strong positive correlation between scale and performance. This suggests that further scaling the dataset size could yield even greater gains. We thus mix and scale up the data pipelines from Section 3 to build OpenThoughts3-1.2M, our 1.2 million-sized dataset. OpenThoughts3-1.2M contains 850,000 math, 250,000 code, and 100,000 science datapoints. We chose this ratio following the OpenThoughts2-1M mixture used to train OpenThinker2, which exhibited strong and balanced performance on par with the DeepSeek-R1-Distill models. To arrive at the target number of samples in each domain, we work backwards to estimate how many questions we need at the beginning of the pipeline. Then, we apply the highest performing strategy at each stage in the pipeline, opting for the more scalable choices if performance is equal. This construction process of OpenThoughts3-1.2M is illustrated in Figure 3.

As seen in Table 1, OpenThinker3-7B is the best open-data reasoning model at the 7B scale, regardless of optimization algorithm choice (SFT, RL, or both). OpenThinker3-7B also generalizes well to evaluations held-out throughout the pipeline process, exhibiting the best scores on HMMT, AIME25, and LCB 06/24-01/25. Further results on scaling can be seen in Appendix D.

## 5 CONCLUSION

Through iterative experimentation, our pipeline surfaced several key insights into effective SFT reasoning data curation. These findings collectively shape our final pipeline, allowing us to build OpenThoughts3-1.2M, a state-of-the-art open-data SFT reasoning dataset, composed of science, math, and code data. Our final model, OpenThinker3-7B, trained on this data, is the SOTA open-data reasoning model at its model scale.

This work has several limitations. We did not explore datasets for reinforcement learning, a standard training regime for building reasoning models. Within the SFT realm, we did not explore the use of staged SFT or curriculum learning to further improve performance. We nonetheless believe this work serves as a valuable foundation for the community's continued progress on open reasoning models.

## 6 REPRODUCIBILITY STATEMENT

We have provided all adequate details to fully reproduce our datasets, including techniques used, code, and prompts. We have also discussed hyperparameters for the training and evaluation setup in the appendix. We believe this is sufficient information for reproducing our experiments.

## ACKNOWLEDGEMENTS

We would like to thank Alex Fang, Matthew Wallingford, Jiaqi Zeng, Oleksii Kuchaiev, Asad Aali, Benjamin Burchfiel, and Russ Tedrake for helpful discussions and support at various stages of the project. We would also like to thank Adam R. Klivans, Mike Garrison, and Satya Kotari for support with compute and infrastructure. Further thanks go for support provided by supercomputing facilities and their teams, especially to Damian Alvarez and Mathis Bode from Juelich Supercomputer Center (JSC, Germany) and to Laura Morselli from CINECA (Italy).

This research is supported by Toyota Research Institute (TRI), the Vista GPU Cluster through the Center for Generative AI (CGAI) and the Texas Advanced Computing Center (TACC) at UT Austin, the high-performance computer at the NHR Center of TU Dresden, and the compute resources of the AI Systems provided by the Leibniz Supercomputing Centre (LRZ) through the Munich Center for Machine Learning.

This research is also supported by NSF Grants AF 1901292, CNS 2148141, IFML CCF 2019844, and research gifts by UT Austin Machine Learning Lab (MLL) and the Onassis Foundation -Scholarship ID: F ZS 056-1/2022-2023. AGD was with UT Austin for a part of this work. HB is supported in part by AFOSR MURI grant FA9550-22-1-0380. MN and JJ acknowledge funding by the Federal Ministry of Education and Research of Germany (BMBF) under grant no. 01IS24085C (OPENHAFM), under the grant 16HPC117K (MINERVA) and partial funding under grant no. 01IS22094B (WestAI - AI Service Center West), as well as co-funding by EU from EuroHPC Joint Undertaking program under grant no. 101182737 (MINERVA) and from Digital Europe Programme under grant no. 101195233 (openEuroLLM).

This work would not be possible without the support of the Gauss Centre for Supercomputing e.V. through the John von Neumann Institute for Computing (NIC) on the supercomputer JUWELS Booster at Juelich Supercomputing Centre (JSC). We further acknowledge EuroHPC Joint Undertaking for providing this project access to the EuroHPC supercomputer LEONARDO, hosted by CINECA (Italy) and the LEONARDO consortium through an EuroHPC Extreme Access grant EHPC-EXT-2023E02-068, storage resources on JUST granted and operated by JSC and supported by Helmholtz Data Federation (HDF), compute resources provided via WestAI compute grant "Measuring and enhancing advanced reasoning capabilities of foundation models via local model deployment" (westai0007) at JSC and compute resources provided via JSC at JURECA. This work was supported in part by RS-2024-00457882, National AI Research Lab Project.

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

## A    TRAINING DETAILS

### A.1    TRAINING FRAMEWORK

Our training is done using LlamaFactory. This repository is publicly available at `https://github.com/hiyouga/LLaMA-Factory`.

### A.2    HYPERPARAMETERS

For the different scales in Figure 1, we use different hyperparameters for each scale. In general, we want to use larger batch sizes and larger learning rates, but doing this on datasets that are smaller would lead us to take too few steps. We performed hyperparameter sweeps to find an appropriate set of hyperparameters in each model scale. We ultimately ended up with 4 sets of hyperparameters – micro (for 0.3K scale), small (for 1K, 3K scale), medium (for 10K, 30K scale), and large (for 100K scale and above). These hyperparameter sets are identical, except for number of epochs, batch size, learning rate, and packing.

For all hyperparameter sets, we train on DeepSpeed v3 with a cosine learning rate, a warmup of 0.1, weight decay of 0, and with the AdamW optimizer with betas 0.9 and 0.999. The specific differences between the hyperparameter sets can be found in Table 8 below.

| Hyperparameter Set | Dataset Size | LR | Batch Size | Epochs | Packing |
|---|---|---|---|---|---|
| Micro | < 1K | 1e-5 | 32 | 13 | No |
| Small | 1K - 3.16K | 2e-5 | 96 | 7 | No |
| Medium | 3.16K - 31.6K | 4e-5 | 128 | 5 | No |
| Large | > 31.6K | 8e-5 | 512 | 5 | Yes |

Table 8: **Settings for different hyperparameter sets with corresponding dataset sizes**

As shown in Table 8 above, the micro, small, and medium sets are trained without example packing, while the large set is trained with packing. For the large hyperparameter set, we use packing in order to save compute time. On the other hand, for the micro, small, and medium sets, we do not use packing because we want to have a larger number of training steps. We show in Appendix A.3 below that the presence/absence of packing does not significantly affect our performance.

Aside from packing, we adjust certain hyperparameters to optimize for training speed. We use DeepSpeed v3 without memory offloading. In addition, we use persistent dataloader workers, with num_workers=4. We conducted small tests with these settings to ensure that they did not negatively affect performance.

### A.3    PACKING

We investigate the effect of sequence packing on model performance across diverse reasoning and coding benchmarks. Sequence packing concatenates multiple training examples into single sequences to improve computational efficiency, but may affect learning dynamics. Table 9 compares models trained with and without packing on a dataset with shorter sequences. Overall, we see that adding packing does not negatively affect performance. This observation is in contrast with some of the findings reported by Face (2025), where they found that packing negatively affected performance on their setup. We believe this drop in performance can likely be attributed to truncating long sequences into multiple parts, whereas in Llama Factory's packing implementation, packing is implemented in a greedy way, and only shorter sequences are packed together.

### A.4    CHAT TEMPLATE

One other axis we explore is the chat template. More specifically, this refers to how reasoning models can be prompted to produce thinking tokens. This often comes in the form of specialized templates. For instance, the R1 template encloses the thinking tokens in `<think>` and `</think>`. In Appendix A.4, we compare this R1 template with the SkyT1 template which

| Configuration | AIME24 | AMC23 | MATH500 | JEE | GPQA-D | LCB | CodeElo |
|---|---|---|---|---|---|---|---|
| AM100K | 18.7 | **62.0** | **79.6** | 45.6 | **46.5** | **34.7** | **6.3** |
| *w/o packing* | **22.0** | **62.0** | 77.6 | **46.4** | 34.7 | 31.7 | 5.3 |

Table 9: **Performance comparison between models trained with and without sequence packing.** Packing shows mixed effects.

uses `<|begin_of_thought|>` and `<|end_of_thought|>` and was originally used to train the initial OpenThinker-7B on OpenThoughts-114k. From Appendix A.4, we see that the results for the two different chat templates are roughly equivalent, suggesting that the chat template may not be as important as long as they exist (see Appendix A.5 below for ablations where we remove this completely). For simplicity, we stick with the R1 chat template of `<think>` and `</think>` in all of our pipeline experiments.

| Model | AIME24 | AIME25 I | AMC23 | MATH500 | GPQA-D | LCB 05/23-05/24 |
|---|---|---|---|---|---|---|
| OpenThinker-7B | 31.3 | **28.0** | 72.0 | **84.4** | 42.9 | **41.8** |
| w/ R1 template | **32.7** | 22.0 | **72.5** | 83.8 | **43.9** | 33.6 |

Table 10: **Performance comparison of OpenThinker-7B model with and without R1 template across different benchmarks.** The table shows that using simpler `<think>` and `</think>` tokens instead of the more complex SkyT1 tokens (`<|begin_of_thought|>` and `<|end_of_thought|>`) yields comparable or slightly improved performance on most benchmarks.

## A.5 SYSTEM PROMPT

In this subsection, we investigate the effect of removing the chat template completely and simply prompting the model directly. We report our results in Appendix A.5. Our evaluation reveals several key findings. First, enabling explicit reasoning consistently improves performance across mathematical and scientific reasoning tasks, with particularly dramatic improvements on AIME benchmarks (45.3% vs. 2.0% on AIME25). Second, even when reasoning is explicitly disabled, many responses still begin with `<think>` tokens (1681/3127), indicating the model has learned to engage reasoning mechanisms by default. Third, the choice between system prompts shows task-dependent effects: while "reasoning on" helps mathematical tasks, removing system prompts entirely can sometimes yield better results (70.0% vs. 61.3% on AIME24), suggesting that explicit instruction may occasionally constrain the model's natural reasoning patterns.

| Model/Configuration | AIME24 | AIME25 | AMC23 | MATH500 | GPQA-D | LCB 05/23-05/24 |
|---|---|---|---|---|---|---|
| *Llama-3.1-Nemotron-Nano-8B (Ours)* | | | | | | |
| Reasoning On | 61.3 | 45.3 | 94.0 | 89.0 | 55.9 | 68.4 |
| Reasoning Off | 4.0 | 2.0 | 38.0 | 48.8 | 34.7 | 67.1 |
| No System Prompt | 70.0 | 42.7 | 94.0 | 88.8 | 23.2 | 67.2 |
| *Llama-3.1-Nemotron-Nano-8B (Official)* | | | | | | |
| Reasoning On | – | 47.1 | – | 95.4 | 54.1 | – |
| Reasoning Off | – | 0.0 | – | 36.6 | 39.4 | – |

Table 11: **Performance comparison across reasoning benchmarks with reasoning enabled and not.** Turning reasoning on for existing models greatly improves performance. However, using no system prompt at all also performs similarly well to Reasoning On.

## B EVALUATION DETAILS

All model evaluations across benchmarks were performed using **Evalchemy** (Raoof et al., 2025), our unified, multi-GPU evaluation framework. Evalchemy partitions each task into independent

shards, runs them in parallel (via data parallel sharding), and streams per-shard metrics back to a central coordinator for real-time aggregation. This architecture ensured consistent generation settings, reproducible logging of model checkpoints and sampling parameters, and enabled us to compute average model accuracy and standard error over repeated runs. We used Evalchemy to cache full model completions for long chain-of-thought tasks (e.g., AIME, LiveCodeBench, GPQA Diamond), which reduced redundant inference costs and enabled easy analysis of failure cases.

For each benchmark, we report:

- AIME24, AIME25, AMC23, and HMMT: mean accuracy and SEM over 10 iterations
- LiveCodeBench: mean accuracy and SEM over 6 iterations
- CodeForces, CodeElo, GPQA Diamond, JEEBench, and HLE: mean accuracy and SEM over 3 iterations
- MATH500: single pass evaluation on the full 500 sample set

All runs used unified generation configurations: temperature = 0.7, top_p = 1.0, and max_new_tokens = 32,768.

Reasoning models with long CoT often generate multi-step explanations before providing a final answer. The response typically begins with a <think> token, includes intermediate reasoning steps, and ends with a </think> token to indicate the end of the reasoning process. The final answer follows this block. For code generation questions, the final answer is usually marked within a fenced code block with a language tag.

We provide brief descriptions for each of our benchmarks.

1. **AIME24**: a mathematics competition for high-school students held in 2024. It involves 30 questions of different levels of difficulty. Answers are a single integer from 0 to 999.

2. **AIME25**: a mathematics competition for high-school students held in 2025. It involves 30 questions of different levels of difficulty. Answers are a single integer from 0 to 999.

3. **AMC23**: a mathematics competition for high-school students held in 2023. It consists of 40 questions with different difficulty levels. The answers are numerical.

4. **MATH500**: consists of 500 diverse problems in probability, algebra, trigonometry, and geometry.

5. **CodeForces**: consists of 453 real-world programming problems sourced from the Code-Forces platform. The benchmark measures unit test-based execution accuracy with a human-comparable Elo rating.

6. **CodeElo**: consists of 391 real-world programming problems curated from a variety of contests. The benchmark measures unit test-based execution accuracy with a difficulty-calibrated Elo rating.

7. **LiveCodeBench**: a benchmark of real-world programming tasks that evaluate a model's ability to generate, execute, verify, and iteratively repair solutions using unit-test feedback. LiveCodeBench 05/23-05/24 subset has 511 problems released between May 2023 and May 2024, whereas the 06/24-01/25 subset has 369 problems released between May 2024 and Jan. 2025.

8. **GPQA Diamond**: a set of 198 challenging questions from the Graduate-Level Google-Proof Q&A Benchmark (GPQA). Questions are in multiple-choice format.

9. **JEEBench**: contains 515 questions spanning Physics, Chemistry and Mathematics subjects collected from the Joint Entrance Examination (JEE): Advanced held from 2016 to 2023. Questions are in multiple-choice and numerical formats.

10. **HMMT**: 30 questions from the HMMT high school mathematics competition held in February 2025. Questions are in Combinatorics, Number Theory, Algebra, and Geometry.

11. **HLE**: a subset of 512 multiple-choice, text-only questions from the Humanity's Last Exam (HLE) benchmark.

| Benchmark | Domain / Description | Number of Questions |
|---|---|---|
| **Code Generation** | | |
| CodeElo (Quan et al., 2025) | Code generation with human-comparable Elo ratings. | 391 |
| CodeForces (Penedo et al., 2025a) | Benchmarking competition-level code generation. | 453 |
| LiveCodeBench 05/23-05/24 (Jain et al., 2024) | Holistic code benchmark with iterative repair. | 511 |
| LiveCodeBench 06/24-01/25 (Jain et al., 2024) | Holistic code benchmark with iterative repair. | 369 |
| **Mathematical Problem Solving** | | |
| AIME 24 (MAA, 2024) | 2024 AIME math-reasoning dataset. | 30 |
| AIME 25 (MAA, 2025) | 2025 AIME math-reasoning dataset. | 30 |
| AMC 23 (MAA, 2023) | 2023 AMC math-reasoning dataset. | 40 |
| HMMT (Balunović et al., 2025) | High school mathematics competition. | 30 |
| MATH500 (Hendrycks et al., 2021b) | 500-problem split from "Let's Verify Step by Step." | 500 |
| **Science Tasks** | | |
| GPQA Diamond (Rein et al., 2024) | Graduate-level, Google-proof Q&A benchmark. | 198 |
| JEEBench (Arora et al., 2023) | Pre-engineering IIT JEE-Advanced exam questions. | 515 |
| **General Tasks** | | |
| HLE (Phan et al., 2025) | Subject-matter expert questions. | 512 |

Table 12: **We evaluate on 12 tasks across multiple data domains.** We validate experiments on 8 of these tasks, and keep the remaining 4 (AIME 2025, LiveCodeBench 06/24-01/25, HMMT, HLE) as held-out sets.

## C  DECONTAMINATION

Contamination with the evaluation datasets is an important issue, since it poses the danger of misleading results over the actual usefulness of the training set. It is expected that training data that contains evaluation questions in some form will lead to improved performance on those same questions. Such an effect could potentially affect the conclusions of our experiments. To avoid this issue, we perform decontamination against our evaluation sets, via two separate criteria.

The first method used is Indel similarity of each training sample and each evaluation sample. This similarity refers to the number of characters that need to be inserted or deleted from one sample to match the other, and is calculated relative to the sample length. More precisely, we consider the Normalized Indel similarity score between a pair of strings, as computed via the Longest Common Subsequence (LCS) metric:

$$\text{indel}_{\text{sim}} = 100 \times \frac{\text{LCS}_{\text{length}}(s_1, s_2)}{\max(|s_1|, |s_2|)} \tag{1}$$

We consider a similarity of 75% between our two strings to indicate contamination with respect to this metric.

Our second method is an N-gram-based similarity metric. In this setting, we first tokenize both the training and the evaluation sample using the same tokenizer as the Qwen2-7B-Instruct model. We then examine the sets of N-grams in each of the samples, for $N = 13$. If we find that the two samples share an N-gram with each other, then we consider the training sample to be contaminated.

For our pipeline, we consider a training sample contaminated if it is marked as contaminated by either of our methods, and we discard it. The thresholds for our methods are chosen empirically, in order to minimize both false negatives (samples that are contaminated but are not detected) and false positives (samples that are marked as contaminated but are in fact unrelated to the evaluation samples).

We systematically tune our decontamination schema through rigorous experimentation. Our testbed is a manually contaminated dataset; an ideal decontamination scheme can accurately filter out contaminated questions from the normal questions.

**Contaminated Dataset Construction**   Our experiments require a dataset of contaminated and noncontaminated questions. We construct this dataset from several sources.

1. We take test sets (MATH500, GPQA Diamond, LiveCodeBench) and sample exact questions from each test set.

2. We sample questions from test sets and apply three types of alteration. Our first alteration is embedding the question in a longer context, such as "Please help me solve this problem: ".

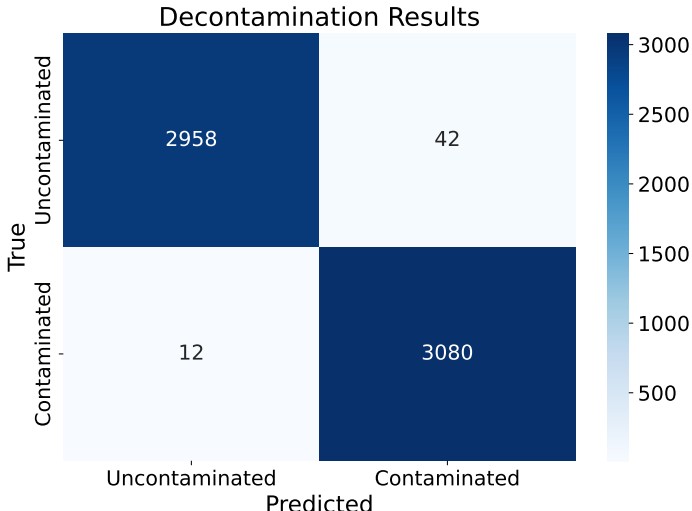

Figure 4: **Our decontamination algorithm accurately identifies contaminated samples.** Our decontamination algorithm has a $99.6\%$ true negative accuracy rate. The algorithm also throws out minimal amounts of noncontaminated samples.

The second alteration is replacing several words with synonyms, numerical expressions with equivalent expressions, and variable names. Our final alteration is changing the formatting of the question by altering paragraph breaks, sentence order, and punctuation.

3. We add uncontaminated questions by creating completely original questions manually.

Overall, our dataset has 3092 contaminated samples and 3000 uncontaminated samples. We tuned our decontamination algorithm to produce nearly 0 false negatives (marking contaminated questions as decontaminated) while not having many false positives. The results of our final decontamination schema are in Figure 4. Our final decontamination algorithm only misses 12 questions out of 3092 manually contaminated items, representing a $99.6\%$ true negative rate. Decreasing the threshold for fuzzy string matching or the $n$ in $n$-gram count significantly raises the false positive rates, which could potentially affect downstream performance. The decontamination schema only throws out $1.4\%$ of noncontaminated samples.

# D    ADDITIONAL SCALING EXPERIMENTS

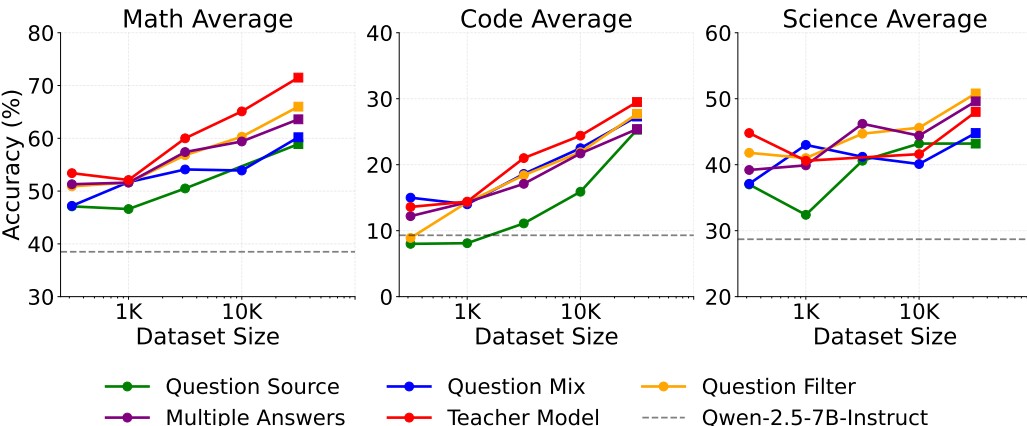

Figure 5: **Scaling the top strategies from each pipeline step.** Across dataset scales, the datasets created by subsequent stages in the pipeline shift the scaling curve upwards. The largest gains come from the selection of question sources, question filtering strategies, and teacher model selection. The average performance on math and code has a clearer scaling trend than the performance on science – the final "Teacher Model" curve not being the top science performer is a consequence of our design choice, where we select winning strategies based on average performance across domains.

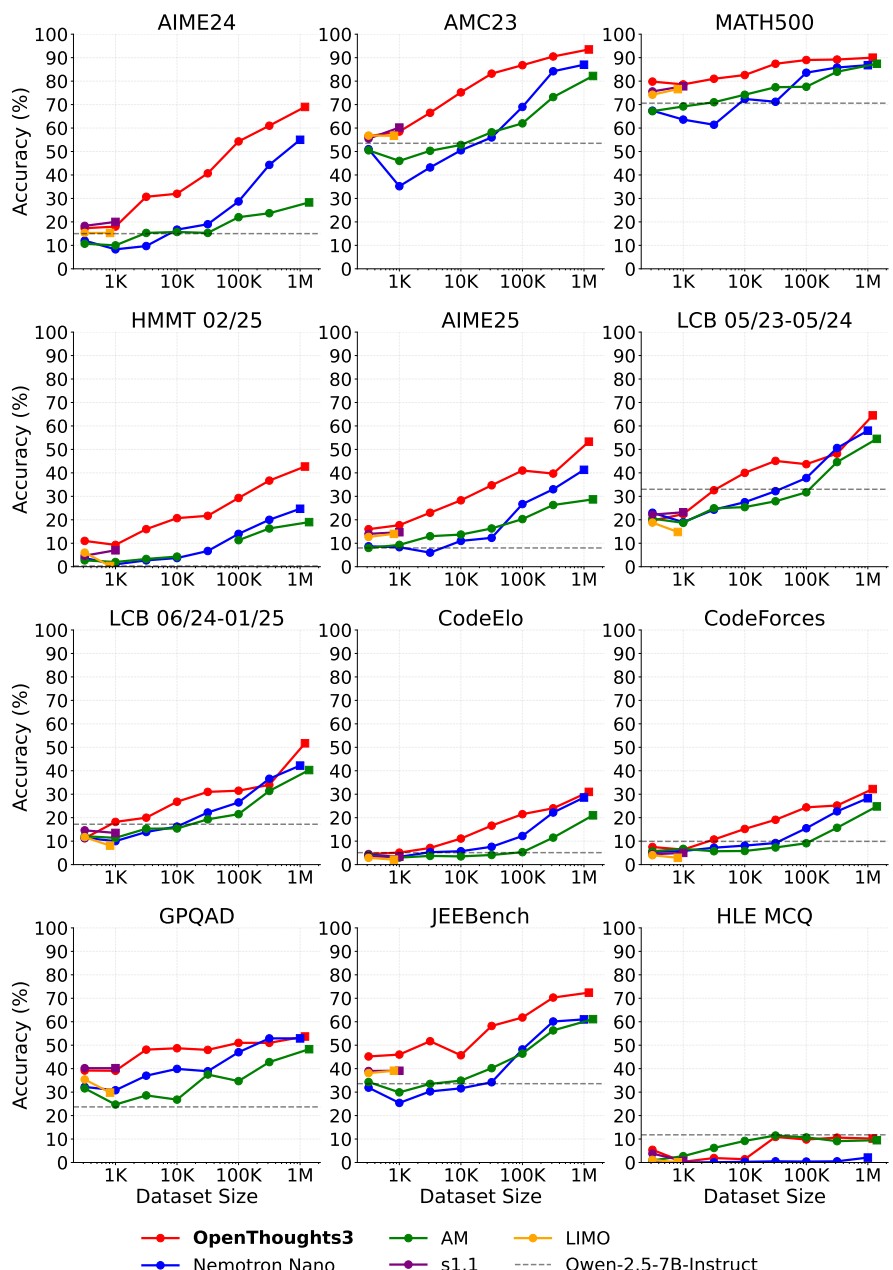

Figure 6: **Downstream model performance after finetuning Qwen-2.5-7B-Instruct on increasingly larger subsets from OpenThoughts3-1.2M**. Across a wide variety of math (AIME, AMC, MATH, HMMT), code (LiveCodeBench, CodeElo, CodeForces), and science (JEEBench, HLE) benchmarks, OpenThoughts3-1.2M outperforms existing reasoning datasets. HMMT, AIME 2025, LiveCodeBench 06/24-01/25, and HLE are "held out", which means that we did not use them to evaluate any intermediate models during our experiments to inform our data recipe.

Studying scaling trends allows us to see if a data recipe is consistent across scales and helps determine whether further scaling is promising. Figure 6 shows that the OpenThoughts3 recipe dominates other reasoning dataset strategies across scales and benchmarks.

Performance on many of the studied benchmarks continues to improve up to the 1M scale. However, some benchmarks are saturating (AMC23, MATH500) at the largest scale, and others do not respond

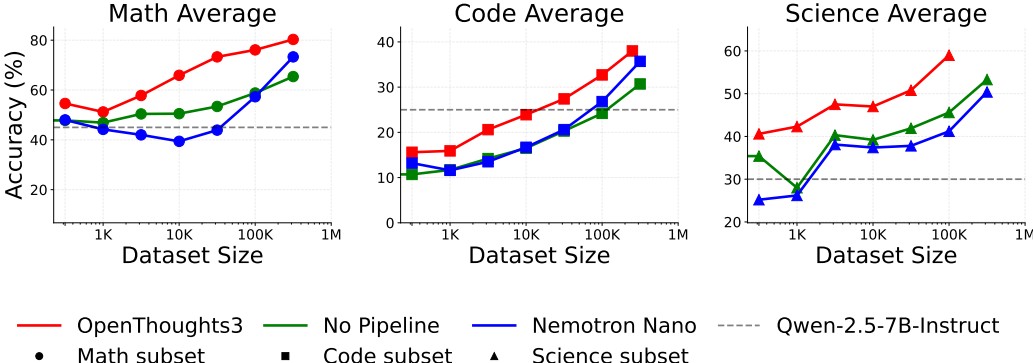

Figure 7: **The OpenThoughts3 data recipe within each domain shows strong scaling over baselines**. Math performance is averaged over AIME24, AMC32, and MATH500. Code performance is averaged over LCB 05/23-05/24, CodeElo, and CodeForces. Science is averaged over GPQA Diamond and JEEBench. The largest scale for the OpenThoughts3 math and science subsets are 250K and 100K, respectively.

to scale (HLE). Scaling the reasoning datasets to even larger sizes beyond 1M is an exciting future direction.

The scaling curves do not always exhibit smooth increases in performance, exhibiting dips and jumps. There is variance in our experimental procedure in training and evaluation, so even with a fixed dataset, the downstream performance will fluctuate. However, we found in our experimentation that re-training and re-evaluating models on a fixed dataset did not fully explain these dips and jumps.

To further study the scaling trends, we first isolate data from each domain and measure the average performance of the in-domain evaluation benchmarks. This matches the same setting as our pipeline experiments in Section 3, in which data recipes are swept for each domain (math, code, and science).

Figure 7 shows the individual domain recipes continue scaling nicely beyond the largest dataset size used in the pipeline experiments, 31.6K, for another order of magnitude. We chose these sizes to study scaling at half an order of magnitude resolution.

We include "No Pipeline" as a naive baseline to demonstrate the full gains from the data recipe determined by our extensive experimentation in Section 3. "No Pipeline" is constructed by taking the union of 31.6K samples from all candidate question sources from the first stage in the pipeline 3.2. Therefore, "No Pipeline" does not include the selection of questions only from high-quality sources, does not employ the filtering questions, uses DeepSeek-R1 instead of QwQ-32B as a teacher model, does not include multiple answer samples per question, and does not have any filtering based on answers. Figure 5 further breaks down the gains due to these choices individually step by step.

## D.1 BASE MODEL

We train OpenThoughts3 using the Llama-3.1-8B-Instruct models. This experiment shows that the scaling gains that we observe are not just limited to Qwen models, and our dataset is indeed scalable and generalizable. The results of this experiment are shown in Appendix D.1. Overall, we see that for some datasets, the Llama models see a more significant performance gain as compared to the Qwen models. This is most prominent in some math datasets such as AMC23 (from 15.8 to 75.2) and MATH500 (from 43.2 to 83.8), though the Qwen models, which start from a stronger starting point, still perform better overall. In the future, we would like to expand this to further consider stronger models such as the recent Qwen 3 series of models.

| Base Model | AIME24 | AIME25 | AMC23 | MATH500 | GPQA-D | LCB 05/23-05/24 |
|---|---|---|---|---|---|---|
| Qwen-2.5-7B-Instruct | **54.3** **(+39.3)** | **41.0** **(+33.0)** | **86.8** (+33.3) | **89.0** (+18.4) | **51.0** **(+27.3)** | 43.7 (+10.7) |
| Llama-3.1-8B-Instruct | 37.0 (+32.3) | 30.3 (+30.0) | 75.2 **(+59.4)** | 83.8 **(+40.6)** | 45.1 (+19.3) | **44.4** **(+31.3)** |

Table 13: **Performance comparison between base models when fine-tuning on 100k samples from OpenThoughts3**. The table shows the absolute performance scores achieved by fine-tuned models, with improvements over the respective base models shown in parentheses. Both fine-tuned models demonstrate substantial improvements on all benchmarks when trained on OpenThoughts3 data. While Llama-3.1-8B-Instruct experiences larger lifts on AMC23, MATH500, and LCB 05/23-05/24, using Qwen-2.5-7B-Instruct results in the overall best performance.

| Models | Average | Code Avg | Math Avg | Science Avg |
|---|---|---|---|---|
| OpenThinker-32B | **64.5**$_{0.3}$ | **45.8**$_{0.3}$ | **83.7**$_{1.0}$ | **63.7**$_{0.2}$ |
| OpenThinker-32B-Unverified | 62.1$_{0.3}$ | 43.8$_{0.4}$ | 81.4$_{0.9}$ | 60.5$_{0.3}$ |
| OpenThinker-7B-Unverified | **45.0**$_{0.4}$ | **24.2**$_{0.3}$ | **64.6**$_{0.9}$ | **46.9**$_{0.6}$ |
| OpenThinker-7B | 41.9$_{0.6}$ | 21.8$_{0.5}$ | 62.0$_{1.9}$ | 41.9$_{0.6}$ |

Table 14: **Impact of Verification on original OpenThinker models.** We see verification hurts at the 7B model scale but helps at the 32B model scale.

# E    ADDITIONAL DATA RECIPE EXPERIMENTS

Due to space constraints, we could not present all of our data curation ablations in the main text. This section discusses several interesting experiments that provide further insights into tools for improving reasoning models.

## E.1    VERIFICATION

Verification played a large role in OpenThoughts-114K. We examine how verification impacted OpenThoughts-114K in the following sections.

### E.1.1    VERIFICATION IMPACT ON THE ORIGINAL OPENTHOUGHTS

Verification played an important part in OpenThoughts-114K and OpenThoughts2. However, OpenThoughts3 does not rely on any form of verification. A natural question is how important empirically was verification for the original OpenThoughts experiments. Table 14 demonstrates the findings of this study. We trained a 7B and 32B model on the unverified version of OpenThoughts-114K and evaluated the difference between the unverified models and the original models. Our results show that verification may hurt performance at the 7B level but help at the 32B level.

### E.1.2    REMOVING PROOF-BASED QUESTIONS

We push further on the impact of verification on OpenThoughts-114K. Some math questions in OpenThoughts-114K are proof-based. This characteristic can make our numerical verification less accurate. A simple question is whether removing proof-based questions improves downstream performance due to more accurate verification. Table 15 contains the results of this ablation. Removing proofs degrades performance on relevant benchmarks despite being unverifiable with our methodology.

### E.1.3    EXTRACTION-BASED MATH VERIFICATION

Another verification strategy we explore is filtering question–answer pairs based on answer correctness. For math examples with a known ground truth answer, we compare the model-generated answer

| SFT Datasets | | | Benchmarks | | |
|---|---|---|---|---|---|
| Datasets | Average | Code Avg | Math Avg | Science Avg |
| OpenThinker-7B | **41.9**$_{0.6}$ | **21.8**$_{0.5}$ | **62.0**$_{1.9}$ | 41.9$_{0.6}$ |
| OpenThinker-7B w/o proofs | 39.4$_{1.3}$ | 15.2$_{2.9}$ | **60.4**$_{1.0}$ | **44.2**$_{0.3}$ |

Table 15: **Comparison of OpenThoughts with and without proof-based questions.** Throwing out proof-based questions harms performance overall by 5.6 points on average.

| Extraction Method (Training) | Dataset Size | Extraction Method (Evaluation) | AIME25 | MATH500 |
|---|---|---|---|---|
| LLM judge | 114K | Hendrycks-Math (default) | 31.3 | 84.4 |
| LLM judge | 114K | HF Math-Verify | **44.0** | **89.0** |
| Math-Verify | 83K | Hendrycks-Math (default) | 23.0 | 55.0 |
| Math-Verify | 83K | HF Math-Verify | 22.7 | 82.2 |

Table 16: **Comparison of math answer verification strategies.** We filter the OpenThoughts-114K dataset using either LLM-based or Math-Verify-based answer correctness. The resulting models are evaluated using both Hendrycks and Math-Verify answer extraction tools.

| SFT Datasets | | Benchmarks | |
|---|---|---|---|
| Datasets | LiveCodeBench | CodeElo | CodeForces |
| Verified via Unit Tests | 36.0 | 9.4 | 10.4 |
| Unfiltered (Random Sample) | **38.5** | **10.7** | **13.54** |

Table 17: **Effect of using LLM-Generated unit tests for code data verification.** Downstream performance of models trained on 16,000 code examples: one set filtered to include only samples verified by LLM-generated unit tests, and the other unfiltered. No improvement is observed from verification-based filtering.

to the reference and discard samples with incorrect responses. However, extracting and evaluating the model's final answer, which is often embedded in complex mathematical expressions, is non-trivial.

To address this, we experiment with two answer extraction methods: (i) using the Math-Verify toolkit from Hugging Face, and (ii) using an LLM-based extractor (OpenThinker-7B).

We apply both methods to filter the OpenThoughts-114K dataset and train downstream models. For evaluation, we again compare Math-Verify against the default answer extractor from Hendrycks et al. (2021b). Table 16 summarizes the results across two benchmarks—AIME25 and MATH500—under different combinations of data generation and evaluation verifiers.

### E.1.4 LLM-GENERATED UNIT TEST VERIFICATION

We investigate the effect of filtering code examples by LLM-generated unit tests. From an initial pool of 45,000 question–answer pairs, we use GPT4o-mini to (1) detect which answers contain Python code and (2) generate a standalone, executable unit test for each Python instance. We then apply our "verification" filter only to those Python examples, while non-Python examples remain untouched. Next, we fine-tune Qwen2.5-7B-Instruct on two separate subsets of 16,000 samples each: one filtered to include only examples whose generated tests pass, and one drawn at random without filtering. The results shown in Table 17 suggest that an LLM-generated unit test verification does not improve downstream code-generation accuracy.

### E.2 TEACHER MODEL

In this section, we study Claude 3.7 (with thinking mode) as an annotator. First, we show that Claude-thinking traces contribute to better performance in code, maths, and general question answer-

ing. Longer thinking traces lead to better results in all three categories of benchmarks. Then we demonstrate that using Claude 3.7 to re-annotate the S1K Muennighoff et al. (2025) dataset (math) as well as our science or code data actually leads to worse performance than using R1.

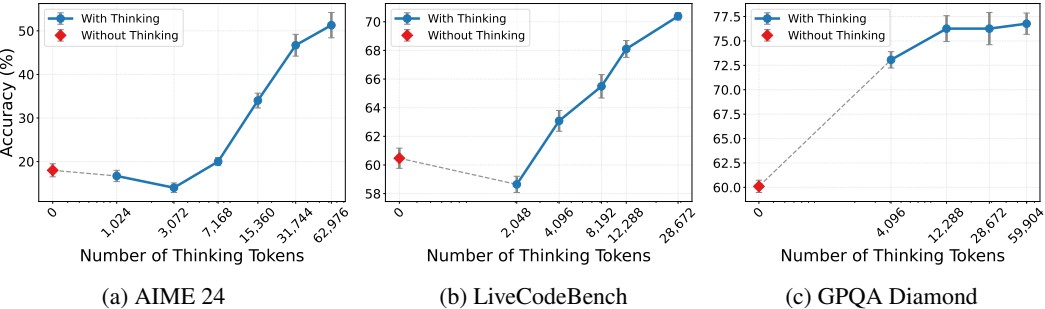

(a) AIME 24       (b) LiveCodeBench       (c) GPQA Diamond

Figure 8: **Claude 3.7 accuracy improves consistently with larger thinking-token budgets across three benchmarks.** Each panel plots mean accuracy (markers) and $\pm 1$ standard error (error bars) over multiple independent runs (5 for AIME 24, 3 for LCB and GPQA Diamond). The horizontal axes are logarithmic in the number of *thinking tokens*; the answer budget is 1 024 tokens for AIME 24 and 4 096 tokens for LCB and GPQA Diamond and is not counted in the thinking tokens budget. *AIME 24:* accuracy rises from a no-thinking baseline of $18.0\%$ (red diamond) to $51.3\%$ when the model is allowed 62 976 thinking tokens. *LCB:* performance climbs steadily from $60.5\%$ to $70.4\%$ at 28 672 thinking tokens. *GPQA Diamond:* accuracy increases from $60.1\%$ without thinking to $\approx 76\%$ at 12 288 tokens, after which the curve plateaus, illustrating diminishing returns beyond this budget.

**Claude 3.7 with thinking.** The API interface to Claude 3.7 allows the user to set a budget for the number of thinking tokens. This permits us to study the evolution of the benchmark performance as test-time compute is increased. Figure 8 shows that for all types of tasks tested (mathematical reasoning, coding, and question answering), increasing test time is beneficial. This is especially true for mathematical reasoning and coding; GPQA Diamond performance saturates earlier.

**Claude vs R1 as an annotator for code.** To assess Claude 3.7 as an annotator for code, we consider the OpenThoughts-114K dataset and re-annotated 10K random coding problems from OpenThoughts-114K with Claude and verified them, which yields 5.8K verified examples. We mixed this with the OpenThoughts math and science parts, so that the proportions of the code, math, and science parts of the resulting dataset are the same as for OpenThoughts 1. This swaps the code annotator from R1 to Claude. Table 18 shows that Claude performs slightly worse as a code annotator in this experiment.

| Code Annotators | AIME24 | GPQA | MATH500 | LiveCodeBench |
|---|---|---|---|---|
| R1 | 0.233 | 0.399 | **0.816** | **0.341** |
| Claude | **0.266** | **0.404** | 0.806 | 0.323 |

Table 18: **Scores for switching the code annotator from R1 to Claude.**

**Claude vs R1 as an annotator for science** To assess Claude 3.7 as an annotator for science, we consider the OpenThoughts 1 dataset and re-annotated and verified the science part from OpenThoughts 1 with Claude, analogously as above for code, we swapped the science annotator from R1 to Claude. The results in Table 19 show that Claude performs slightly worse as a science annotator in this context.

**Claude vs R1 vs Gemini as annotators for S1** To further assess annotators for math, we consider the S1K dataset (Muennighoff et al., 2025) and re-annotated its answers and reasoning traces with Claude and R1. Table 20 shows much better performance with R1 annotations. However, Claude annotations did not show as strong improvements.

| Science Annotators | AIME24 | AIME25 | AMC23 | MATH500 | GPQA | LiveCodeBench |
|---|---|---|---|---|---|---|
| Claude 3.7 | 0.3733 | 0.2733 | 0.740 | 0.840 | **0.2138** | 0.4207 |
| R1 | **0.3867** | **0.2933** | **0.765** | **0.872** | 0.2121 | **0.4586** |

Table 19: **Scores for switching the science annotator from R1 to Claude.**

| Models | AIME24 | AIME25 | MATH500 | GPQA |
|---|---|---|---|---|
| Gemini simplescaling/s1K | **56.7** | 26.7 | 93.0 | 59.6 |
| R1 simplescaling/s1K-1.1 | **56.7** | **60.0** | **95.4** | **63.6** |
| Claude-3-7 simplescaling/s1K-claude-3-7-sonnet | 40.0 | – | 87.0 | 51.5 |

Table 20: **Scores for switching the annotator from Gemini to R1 or Claude.**

### E.3 COMPRESSING REASONING TRACES

So far, we performed supervised fine-tuning on long reasoning traces (up to 16K tokens) before predicting the final answer. Recent work (Chen et al., 2024; NovaSky-Team, 2025a) highlights the significant inference cost associated with long reasoning traces and the tendency of reasoning models to overthink. In this section, we study how reducing reasoning traces during training affects downstream performance. We employ two methods: (a) removing self-reflection components from the reasoning traces, and (b) filtering out instances where the reasoning trace length is above a specified threshold.

To examine the first approach, we begin with a random subset of 12K instances from the OpenThoughts3 dataset, ensuring that each instance contains a complete thought (i.e., `<\think>` is present in the reasoning trace). Typically, reasoning traces are long due to the model's self-reflective behavior, where it (re-)analyzes prior solutions and proposes alternative approaches to the problem. To investigate the role of self-reflection, we remove keywords such as "wait", "but wait", and "but the question" from the reasoning traces, following the approach of Deng et al. (2025). This reduces the average reasoning trace length from 11.6K to 0.3K tokens. We present the results in Table 21. Notably, we observe that removing self-reflection leads to an average relative performance drop of $49.1\%$ across diverse downstream benchmarks. These findings suggest that self-reflection and long-form reasoning structures are essential for enhancing the reasoning capabilities of OpenThoughts3 models.

Our second approach to reducing the reasoning trace is to filter instances whose lengths exceed a given threshold (e.g., 2048, 4096, or 8192 tokens). This method reduces both the length of the reasoning traces and the overall dataset size, while preserving the self-reflection capabilities within the retained traces. We present the results in Table 21. We observe that the downstream performance of models trained on the filtered datasets drops significantly compared to the default dataset. Specifically, the filtered-2048 setting results in a relative performance degradation of $33.3\%$ across downstream benchmarks. Furthermore, higher filtering thresholds show improved model performance. This indicates that the presence of long reasoning structures in the dataset is beneficial, and that self-reflection alone is not sufficient for achieving strong reasoning performance. Nevertheless, the ability to reduce overthinking post-hoc remains an active and highly relevant area of research (Sui et al., 2025).

### E.4 POORLY PERFORMING SFT DATASETS FROM WEAK QUESTIONS

When benchmarking existing SFT reasoning datasets, we finetune models on the full sample sets from multiple available sources. Evaluating downstream performance on our fixed benchmark suite, we observe a wide range of dataset quality.

To investigate further, we finetune on a small subset of 31,600 randomly sampled question–answer pairs from each original dataset and re-annotate those questions with DeepSeek-R1, using the same procedure described in the sourcing stage of our pipeline (Section 3.2). Once again, we observe significant variance in downstream performance, which appears to stem primarily from differences in the quality and nature of the question sources.

| Setup | Avg. length | AIME24 | MATH500 | GPQA | LCB 05/23-05/24 | Average |
|---|---|---|---|---|---|---|
| Baseline | 11593 | **34.0** | **84.0** | **45.6** | **40.7** | **51.4** |
| No Self-Reflection | 328 | 5.0 | 61.8 | 31.5 | 19.2 | 26.3 (-49.1%) |
| Filter > 2048 | 1343 | 16.7 | 70.2 | 32.3 | 26.9 | 34.2 (-33.3%) |
| Filter > 4096 | 2305 | 18.3 | 74.6 | 42.6 | 31.7 | 39.9 (-22.3%) |
| Filter > 8192 | 4775 | 22.0 | 79.6 | 44.4 | 30.3 | 42.3 (-17.6%) |

Table 21: **Evaluating the role of compressing reasoning traces on downstream performance.** The first row shows the performance of the model trained on the default OpenThoughts3 dataset (12K instances), where the average reasoning trace length is 11.5K tokens. The second row reports performance when self-reflection capabilities are removed, reducing the average trace length to 0.3K tokens. The subsequent rows present results for a filtering strategy that removes instances with reasoning trace lengths exceeding a certain threshold (2048, 4096, or 8192 tokens). Overall, the results underscore the importance of both self-reflection and long reasoning structures for achieving strong performance across diverse evaluation benchmarks.

| | Benchmark | SYNTHETIC-1-SFT-Data | KodCode-V1-SFT-R1 | code_codegolf | code_kodcode | code_stack_exchange | code_understanding | real_world_swe |
|---|---|---|---|---|---|---|---|---|
| | Train Size | 894K | 268K | 31.6K | 31.6K | 31.6K | 31.6K | 31.6K |
| | Method | SFT | SFT | SFT | SFT | SFT | SFT | SFT |
| | Average | **42.6** | 26.9 | 37.3 | 36.2 | 27.3 | 26.9 | 29.8 |
| *Math* | AIME24 | **40.7** | 15.0 | 17.7 | 20.3 | 14.0 | 15.7 | 12.3 |
| | AMC23 | **78.5** | 50.5 | 58.0 | 56.3 | 52.2 | 50.7 | 52.2 |
| | MATH500 | **87.6** | 71.8 | 77.0 | 72.8 | 74.0 | 70.0 | 72.4 |
| | MMLUPro | **31.0** | 28.0 | 30.6 | 28.2 | 25.2 | 25.6 | 24.4 |
| *Code* | CodeElo | **17.2** | 7.4 | 14.4 | 12.5 | 3.8 | 2.6 | 7.8 |
| | LCB 05/23-05/24 | **48.3** | 39.1 | 44.4 | 43.9 | 9.3 | 0.6 | 15.9 |
| | CodeForces | **21.8** | 10.5 | 17.2 | 15.3 | 4.4 | 3.2 | 8.4 |
| *Sci* | GPQA-D | **45.1** | 31.6 | 42.6 | 41.6 | 33.7 | 42.4 | 38.6 |
| | JEEBench | **57.2** | 31.0 | 38.8 | 39.3 | 29.5 | 31.9 | 36.3 |

Table 22: **Performance comparison: Full-scale datasets vs. controlled ablation study.** SYNTHETIC-1-SFT-Data (894K) achieves the highest performance with an average of **42.6**, significantly outperforming all controlled datasets. Among the size-controlled 31.6K datasets, code_codegolf serves as the best baseline (average score of 37.3), with code_kodcode achieving competitive performance (average score of 36.2). The vertical line separates full-scale mixed datasets from sample-size controlled ablation experiments.

## E.5 STACKING GAINS FROM OUT OF DOMAIN TRANSFER

During our experimentation on the dataset pipeline in Section 3, it was clear that reasoning ability transferred across domains. For example, scores on science evaluations would increase when a model was finetuned on only math reasoning data. We often observed such significant out-of-domain performance gains between candidate pipeline choices that the model with the highest in-domain average performance would not be the same as the model with the highest overall average performance.

We studied whether these gains due to cross-domain transfer persisted when all the domains are mixed together. To do this, we selected datasets from the answer filtering 3.6 portion of the pipeline experiments - the strongest performing science reasoning dataset on in-domain science evaluations (longest answer filtering), the strongest performing code reasoning dataset on out-of-domain science evaluations (longest answer filtering), and the weakest performing code reasoning dataset on out-of-domain science evaluations (shortest answer filtering).

Then, we compared the downstream science performance between the mixes created by combining the science dataset with the two different code datasets. The large difference in the GPQA Diamond scores of the two code datasets did not show up after mixing with the strong science dataset. In other words, the gains from the out-of-domain transfer seen on code datasets disappear when the in-domain science data is mixed in.

| Finetuning Dataset | JEE | GPQA-D | LCB 05/23-05/24 | CodeElo | CodeForces |
|---|---|---|---|---|---|
| Science | 48.7 | 48.8 | 21.8 | 6.3 | 7.9 |
| Code (high GPQA) | 46.6 | 47.3 | 44.6 | 15.9 | 18.9 |
| Code (low GPQA) | 44.3 | 36.7 | 45.8 | 15.1 | 19.7 |
| Science + Code (high GPQA) | 50.4 | 52.7 | 20.2 | 15.3 | 17.6 |
| Science + Code (low GPQA) | 51.1 | 52.7 | 20.7 | 14.6 | 19.6 |

Table 23: **Out-of-domain (code to science transfer) gains do not persist when mixed with the in-domain (science) dataset**. The individual datasets (above the midline) contain only 31K samples from that domain and the mixes (below the midline) contain 62K samples of both code and science. When combining a code dataset with strong performance on science evaluations with a strong science dataset, there is no difference in the downstream model GPQA-D scores over mixing with a code dataset with weak performance on science evaluations.

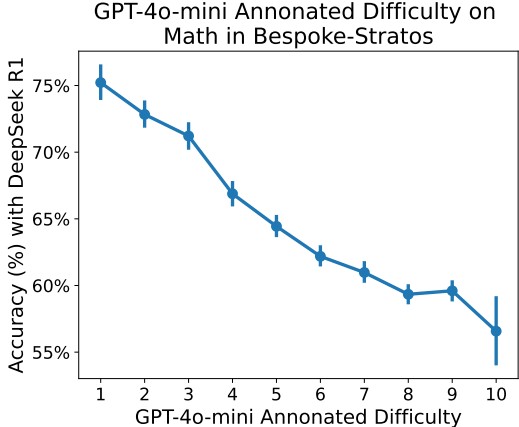

Figure 9: GPT-4o-mini can reliably determine the relative difficulty of questions. DeepSeek R1 performs substantially worse on the hardest questions (difficulty ten).

# F    MODEL REASONING PERFORMANCE ANALYSIS

## F.1    PERFORMANCE ON MATH DIFFICULTY

While developing OpenThoughts, we explored using language models to filter math data questions by difficulty. We applied the difficulty labeling prompt from Sky-T1 NovaSky-Team (2025b) to the math questions from Bespoke-Stratos with GPT-4o-mini as the annotator. This prompt uses example problems to judge questions on a 1-10 scale, with one corresponding to beginners questions and ten corresponding to the hardest IMO problems.

We found that GPT-4o-mini could reliably predict which questions DeepSeek-R1 would correctly answer. At the lowest level of difficulty (one) R1 scored over 75%, while at the highest (ten) R1 scored less than 57% (Figure 9). This built confidence that LLM-annotated difficulty labeling could be used to filter the hardest (and therefore potentially most useful for training) questions.

## F.2    PERFORMANCE SCALING BY CODE DIFFICULTY

While testing the scaling of OpenThoughts3 (shown in figure 1 of the main paper), we noticed a slight drop in performance on all code benchmarks at 100K scale. To study this phenomenon in more detail, we studied LiveCodeBench 05/23-05/24 and represented the contributions of each difficulty level to the average accuracy. We expected to see a saturation of the easy category and very low performance levels in the hard category. Surprisingly, in figure 10, we observe that the models' accuracies are actually increasing nicely with scale for both the medium and hard tasks, and the slight drop is entirely happening in the easy category.

## F.3    SAMPLING BY LONGEST, SHORTEST, MAJORITY

When sampling multiple responses, we have multiple aggregation strategies to predict the final answer as a number or an MCQ choice: (1) "shortest": using the answer of the shortest response as the final answer; (2) "longest": using the answer of the longest response as the final answer; (3) "majority": using the majority prediction as the final answer.

Our experiments reveal that *the shortest response strategy consistently outperforms the longest response strategy* across models and datasets. While majority voting often achieves the best overall performance, the shortest strategy provides an effective single-response selection method that typically outperforms both vanilla sampling and longest response selection.

On the AIME24 mathematical reasoning dataset (Table 24), majority voting generally achieves the best performance, but the shortest response strategy still outperforms the longest strategy for most models. As shown in Table 1, while majority voting achieves the highest scores (e.g., 76.67% for

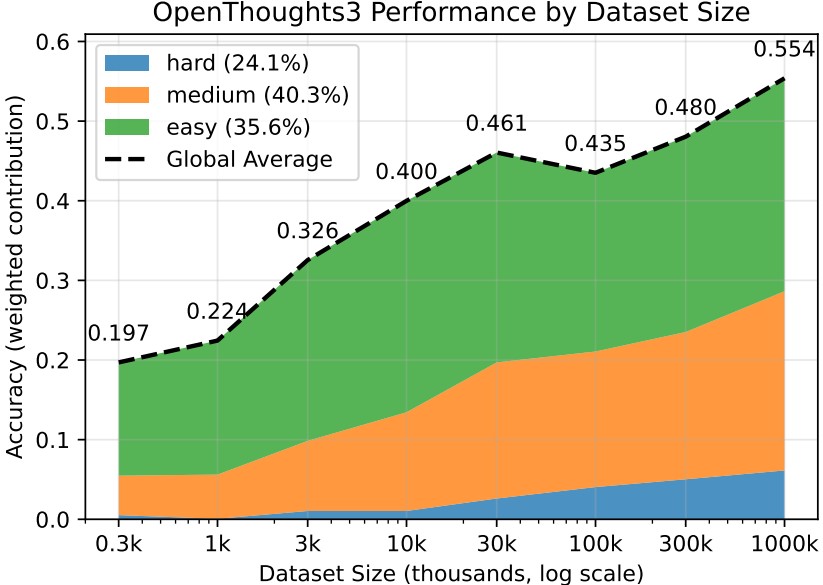

Figure 10: OpenThoughts3 performance scaling on LiveCodeBench 05/23-05/24 as the dataset size is increased. The contribution of the performance on each category of problem is represented.

DeepSeek-R1-Distill-Qwen-7B), the shortest strategy (66.67%) significantly outperforms the longest strategy (36.67%).

| Model | Shortest | Longest | Majority | Vanilla |
|---|---|---|---|---|
| DeepSeek-R1-Distill-Qwen-7B | 66.67 | 36.67 | **76.67** | 55.00 |
| OpenThinker-7B | **43.33** | 13.33 | **43.33** | 30.33 |
| simplescaling-s1-32B | 40.00 | 30.00 | **46.67** | 35.00 |
| NovaSky-Sky-T1-32B | 30.00 | 26.67 | **43.33** | 30.67 |

Table 24: **Performance comparison of sampling strategies on AIME24 dataset.** "Vanilla" refers to the average pass rate across all sampling.

The trend is even more pronounced on the GPQA Diamond scientific reasoning dataset. Table 25 presents results from our most comprehensive experiments with higher run counts. The shortest strategy consistently outperforms the longest strategy across most models, with improvements ranging from 7-11 percentage points.

| Model | Runs | Shortest | Longest | Majority | Vanilla |
|---|---|---|---|---|---|
| DeepSeek-R1-Distill-Qwen-7B | 43 | **51.01** | 43.94 | 29.80 | 48.81 |
| OpenThinker-7B | 81 | **44.44** | 39.39 | 24.75 | 42.28 |
| simplescaling-s1-32B | 44 | **50.00** | 48.99 | 30.30 | 52.79 |
| NovaSky-Sky-T1-32B | 65 | 40.91 | **42.42** | 27.78 | 49.99 |

Table 25: **Performance comparison on GPQA Diamond dataset (high-run experiments)**

**Response Length Analysis** An interesting pattern emerges when examining the relationship between response length and correctness. Table 26 shows the average token lengths for correct versus incorrect responses in vanilla sampling. Across most models, *incorrect responses tend to be significantly longer than correct ones*, suggesting that verbose reasoning may actually indicate uncertainty or error-prone reasoning paths.

| Dataset | Model | Correct | Incorrect | Difference |
|---------|-------|---------|-----------|------------|
| AIME24 | DeepSeek-R1-Distill-Qwen-7B | 7,817 | 18,198 | +10,381 |
| | OpenThinker-7B | 8,966 | 19,517 | +10,551 |
| | simplescaling-s1-32B | 5,048 | 7,481 | +2,433 |
| | NovaSky-Sky-T1-32B | 1,802 | 3,327 | +1,525 |
| GPQA Diamond | DeepSeek-R1-Distill-Qwen-7B | 5,034 | 6,622 | +1,588 |
| | OpenThinker-7B | 7,471 | 8,772 | +1,301 |
| | simplescaling-s1-32B | 3,617 | 3,790 | +173 |
| | NovaSky-Sky-T1-32B | 911 | 896 | -15 |

Table 26: **Average response length comparison: Correct vs. Incorrect responses**

The superiority of the shortest response strategy suggests that concise reasoning often captures the most direct and correct solution path. Longer responses may indicate the model is uncertain, exploring multiple approaches, or getting lost in unnecessary complexity. This finding has important implications for practical deployment, as selecting the shortest response is computationally efficient if you stop generating all samples and often yields better results than more complex aggregation methods.

However, it's important to note that the shortest strategy is not universally optimal. For some models, like NovaSky-Sky-T1-32B on certain datasets, other strategies may perform better, indicating that the optimal sampling strategy may be model-dependent. Additionally, majority voting can sometimes achieve the highest performance when computational resources allow for multiple response generation and aggregation.

## G    SURPASSING THE TEACHER WITH DISTILLATION FOR LEGAL REASONING

In this work, we focus primarily on reasoning for math, science, and coding. However, reasoning can also be beneficial for other domains, for example, for legal reasoning. For such domains, reasoning models out of the box can be significantly improved through finetuning.

We consider a classification task from the Lawma benchmark (Dominguez-Olmedo et al., 2025), specifically, we consider the task of classifying the ideological direction of an opinion from the Supreme Court as conservative, liberal, or unspecificable. This is a challenging task, since general models perform relatively poorly on this task, for example, GPT4 only performs slightly above $50\%$. Dominguez-Olmedo et al. (Dominguez-Olmedo et al., 2025) provide 5.44K labeled training examples as well as 1.52K labeled test examples.

We consider three data generation strategies: i/ We finetune Qwen2.5-7B on a dataset obtained by taking 2K examples, each annotated 5x independently by R1, and verified with a majority vote. Only the traces where the outcome agrees with the majority are kept (8.3K many of the 10K), the others are filtered out. This strategy does not use the provided expert labels. ii/ We finetune Qwen2.5-7B on a dataset obtained by taking 2K examples, each annotated $5\times$ independently by R1, and verified with the expert labels, resulting in 7.36K verified examples. iii/ We finetune Qwen2.5-7B on a dataset obtained by taking all 5.4K examples each R1-annotated once and verified with the expert-provided label, resulting in 4.02K verified examples.

Table 27 contains the accuracies in predicting the ideological direction of an opinion from the Supreme Court as conservative, liberal, or unspecificable for the different data generation strategies. It can be seen that all finetuned Qwen2.5-7B models outperform the much larger annotator (i.e., R1) by a significant margin.

Those results demonstrate that finetuning with a strong teacher model, like R1, followed by verification can yield strong models for specialized tasks, such as this legal reasoning task. Interestingly, finetuning without the expert labels based on consensus/majority verification performs almost as good as using the expert labels for verification.

| Model / Setup | Accuracy |
|---|---|
| Qwen2.5-7B (no finetuning) | 0.271 |
| Qwen2.5-7B finetuned on 2K examples ($5\times$ R1-annotated, majority verified) | 0.819 |
| Qwen2.5-7B finetuned on 5.4K examples (annotated, verified) | 0.820 |
| Qwen2.5-7B finetuned on 2K examples ($5\times$ R1-annotated, verified) | **0.828** |
| R1 (annotator) accuracy | 0.739 |

Table 27: **Comparison of model performance under different finetuning setups.**

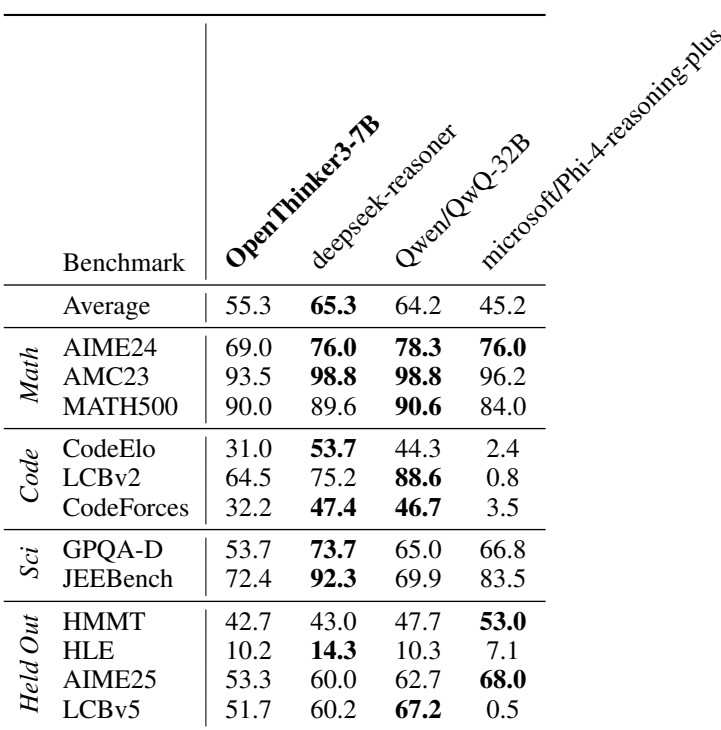

| | Benchmark | OpenThinker3-7B | deepseek-reasoner | Qwen/QwQ-32B | microsoft/Phi-4-reasoning-plus |
|---|---|---|---|---|---|
| | Average | 55.3 | **65.3** | 64.2 | 45.2 |
| Math | AIME24 | 69.0 | **76.0** | **78.3** | **76.0** |
| | AMC23 | 93.5 | **98.8** | **98.8** | 96.2 |
| | MATH500 | 90.0 | 89.6 | **90.6** | 84.0 |
| Code | CodeElo | 31.0 | **53.7** | 44.3 | 2.4 |
| | LCBv2 | 64.5 | 75.2 | **88.6** | 0.8 |
| | CodeForces | 32.2 | **47.4** | 46.7 | 3.5 |
| Sci | GPQA-D | 53.7 | **73.7** | 65.0 | 66.8 |
| | JEEBench | 72.4 | **92.3** | 69.9 | 83.5 |
| Held Out | HMMT | 42.7 | 43.0 | 47.7 | **53.0** |
| | HLE | 10.2 | **14.3** | 10.3 | 7.1 |
| | AIME25 | 53.3 | 60.0 | 62.7 | **68.0** |
| | LCBv5 | 51.7 | 60.2 | **67.2** | 0.5 |

Table 28: **Comparison of OpenThinker3-7B to teacher models.** We see that DeepSeek-R1 is empirically the best model overall and our actual teacher model, QwQ-32B, is empirically worse. Phi-4-Reasoning-Plus performs empirically poorly on code evaluations since it outputs code without code tags which Evalchemy marks as incorrect.

## H ALL TEACHERS ABLATIONS

We report the benchmarks of all of our teacher models in Table 28. Our results indicate that DeepSeek-R1 is the most performant model despite QwQ-32B being the stronger teacher. Phi-4-Reasoning-Plus is also strong on certain benchmarks, but performs poorly on code. This is because it often fails to produce code tags such as ""'python" which Evalchemy uses for code extraction. One notable number is that OpenThinker3-7B outperforms QwQ-32B on JEEBench, demonstrating a single example of weak-to-strong generalization.

## I SAFETY ANALYSIS OF OPENTHINKER MODELS

As we enhance the reasoning capabilities of open-source models, we aim to ensure that our models refuse to respond to unsafe requests while complying with benign requests. To this end, we evaluate the safety capabilities of Openthinker models on the following benchmarks:

| Model | Harmbench ($\downarrow$) | XSTEST ($\downarrow$) |
|---|---|---|
| Qwen2.5-7B-Instruct | 14.5 | 4.4 |
| DeepSeek-R1-Distill-Qwen-7B | 30.5 | 2.4 |
| OpenThinker-7B | 36.8 | 4.4 |
| OpenThinker2-7B | 42.8 | 2.4 |
| OpenThinker3-7B | 55.5 | 5.6 |

Table 29: **Performance of OpenThinker models on safety and over-refusal benchmarks.** Here, we report the harmfulness rate and the over-refusal rate.

- **XSTEST** Röttger et al. (2023) consists of 250 safe prompts that syntactically resemble unsafe prompts. We report the over-refusal rate based on whether GPT-4o classifies the response as refusal or compliance.
- **Harmbench** Mazeika et al. (2024) consists of 400 prompts based on harmful behaviors such as cybercrime, unauthorized intrusion, handling of copyrighted material and misinformation/disinformation. We use GPT-4o as the evaluator and report the proportion of cases that got the maximum score of 5 as the harmfulness rate.

In Table 29, we observe that supervised fine-tuning on reasoning inadvertently degrades the pre-existing safety alignment of Qwen2.5 models, consistent with prior findings Qi et al. (2023). Among the OpenThinker models, OpenThinker-7B achieves a relatively low harmfulness rate (36.8) alongside a moderate over-refusal rate (4.4). In the subsequent generation, OpenThinker2-7B slightly improves the over-refusal rate (2.4), but this comes at the cost of increased harmfulness (42.8). OpenThinker3-7B continues this trend, reaching the highest harmfulness score (55.5) and a modest rise in over-refusal rate (5.6). Notably, OpenThinker3-7B was trained without any explicit safety-tuning or alignment-focused data, which likely contributes to its degraded safety performance.

Interestingly, when comparing Tables 29 and 1, we find a clear trade-off between reasoning capabilities and safety. These findings underscore the challenge of balancing safety and utility in reasoning models Wang et al. (2025); Bercovich et al. (2025). Future work on the OpenThoughts would benefit from incorporating safety-specific datasets to mitigate these risks while preserving their strong reasoning capabilities.

## J EXISTING FRONTIER MODEL EVALUATIONS

Table 30 shows the benchmark results of the models available through APIs. Gemini-2.5-pro displays the strongest performance despite some of its answers being empty (and thus incorrect) due to running out of token budget when its thinking process is too long (especially visible in JEEBench). Claude 3.7 was given a 32K token budget. o3 was used with its default medium reasoning effort. Our OpenThinker3-7B model outperforms, on average, the models to the right of the separator.

## K TESTING REASONING ROBUSTNESS: ALICE IN WONDERLAND EVALUATION

Here, we build on the work on Alice in Wonderland problems (Nezhurina et al., 2024), which use variations in simple problem templates that do not change both problem and solution structure. Given an instance of a problem $P$ and its corresponding solution $S$, reasoning (i.e. abstract solution) $R$ and the final answer $A$, the reasoning-invariant perturbation to the problem statement $P_i^*$ will have a solution $S_i^*$ and answer $A_i^*$. $i$ indicates an arbitrary ID of a perturbation, depending on a problem template, it can have infinitely many perturbations, e.g. if varying variables that hold arbitrary natural numbers. Importantly, $P_i^*$ will have the same abstract solution (or reasoning) $R$ as the original template $P$. Models capable of strong generalization should show similar performance solving the problem across all its structure-preserving variations

We measure model sensitivity to reasoning-invariant problem perturbations to test models' ability to generalize. We follow Nezhurina et al. (2024) in our evaluation setup: we set sampling temperature to 0.1 and sample 100 times, we set a maximum number of output tokens to 30720. Assuming a beta-binomial distribution for models' answers for each variation, we find the mean (average

| | | gemini-2.5-pro-preview-05-06 | o4-mini-2025-04-16 | o3-2025-04-16 | deepseek-reasoner | claude-3-7-sonnet-thinking | gpt-4.1-mini | xai/grok-3 | gpt-4.1-2025-04-14 | gpt-4.1-nano |
|---|---|---|---|---|---|---|---|---|---|---|
| | API Provider | ✦ | ⑤ | ⑤ | 🐋 | ✳ | ⑤ | ∅ | ⑤ | ⑤ |
| | Average | **69.6** | 67.2 | 67.0 | 63.4 | 56.3 | 53.0 | 50.7 | 47.4 | 35.3 |
| *Math* | AIME24 | **92.3** | 81.0 | 80.7 | 76.0 | 47.0 | 48.3 | 60.0 | 50.7 | 31.0 |
| | AMC23 | **100.0** | 99.0 | 97.5 | 99.0 | 73.2 | 87.5 | 89.8 | 83.2 | 71.7 |
| | MATH500 | **93.8** | 90.8 | 86.0 | 91.6 | 78.2 | 88.0 | 85.0 | 83.6 | 79.2 |
| *Code* | CodeElo | **59.5** | 52.9 | 35.2 | 32.1 | 58.4 | 29.5 | 29.8 | 31.1 | 8.4 |
| | LCB 05/23-05/24 | 73.7 | 64.8 | **79.2** | 76.6 | 72.0 | 72.1 | 32.7 | 65.6 | 39.5 |
| | CodeForces | **57.5** | 55.4 | 37.5 | 47.4 | 56.3 | 37.2 | 35.8 | 35.4 | 14.5 |
| *Sci* | GPQA-D | **82.5** | 77.9 | 80.0 | 73.7 | 80.8 | 64.3 | 66.5 | 34.5 | 46.0 |
| | JEEBench | 44.6 | 80.8 | 86.2 | **88.5** | 71.4 | 73.4 | **88.5** | 78.3 | 55.7 |
| *Unseen* | HMMT | **76.7** | 58.0 | 62.3 | 43.0 | 28.0 | 29.7 | 33.3 | 18.7 | 10.0 |
| | HLE | 15.8 | 16.3 | **22.7** | 14.3 | 14.4 | 9.5 | 8.6 | 8.4 | 12.6 |
| | AIME25 | **76.3** | 72.3 | 70.3 | 60.0 | 39.7 | 41.3 | 50.0 | 33.0 | 23.3 |
| | LCB 06/24-01/25 | 62.3 | 57.5 | **66.8** | 59.0 | 55.7 | 55.2 | 28.3 | 46.6 | 31.5 |

Table 30: **Performance of current API-based frontier models.** A significant gap remains between current open-source reasoning models and frontier reasoning models. The vertical line denotes the division between models that underperform and overperform OpenThinker3-7B according to average benchmark performance. The largest gaps arise from benchmarks such as CodeElo, CodeForces, and GPQA Diamond. Gemini-2.5-pro is the most performant model.

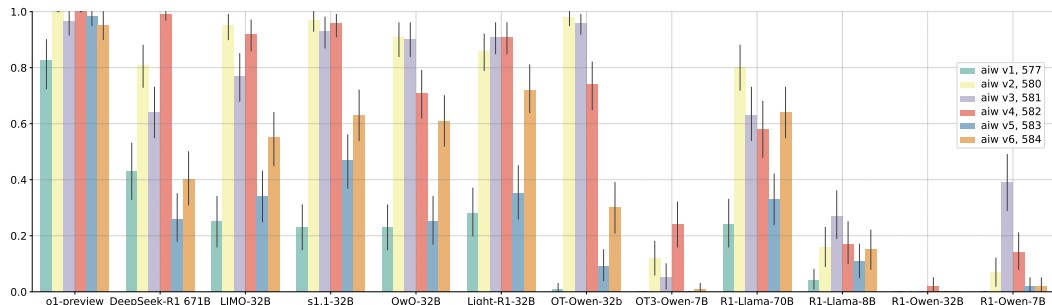

Figure 11: **Distilled reasoning models show deficits in generalization.** All distilled reasoning models exhibit strong performance fluctuations on AIW Friends variations 1-6 Nezhurina et al. (2024), despite the variations not changing problem structure at all. This points to generalization deficits. The fluctuations affect to same extent SFT only (eg S1.1 32B, LIMO-32B) and SFT+RL (eg Light-R1-32B, QwQ 32B) reasoning models. Smaller scale models, eg OpenThinker3-7B, perform worse than larger scale ones, eg OpenThinker-32B, showing overall lower correct response rates. Larger scale 32B models, while having higher overall correct response rate, show strong fluctuations, e.g. OpenThinker-32B going from close to 1 on variations 2 and 3 down to close to 0 on variation 1. For reference, o1-preview and o3-mini are shown, which have much smaller fluctuations and higher overall correct response rates. Distilled models still do not possess robust zero-shot generalization on simple problems. Numbers in the legend are prompt IDs, see Nezhurina et al. (2024) and AIW repo

correct response rate) and the variance $\sigma^2$. We visualize the correct response rate for each model and problem variant as a bar with corresponding error bars to indicate variance (Fig. 11). Despite improved performance on average (compared to non-reasoning models, Fig. 12), distilled reasoning models still show substantial fluctuations on simple task variations, in accord with what was observed in Nezhurina et al. (2024), Fig. 11.

As evident from Fig. 12, reasoning models clearly and strongly outperform their conventional LLM counterparts, showing higher average correct response rates. Despite still persisting clear generalization deficits, reasoning models exhibit a strong boost across AIW problems compared to previous SOTA LLMs, with mid-scale reasoning models (32B) strongly outperforming LLMs trained at the largest scales (e.g. Llama 3.1 405B or DeepSeek v3 671B).

> **Takeaway:** Distilled reasoning models, while strongly improving over standard LLMs, still suffer from generalization deficits, as evident from strong performance fluctuations across natural problem and solution structure preserving variations in problem templates.

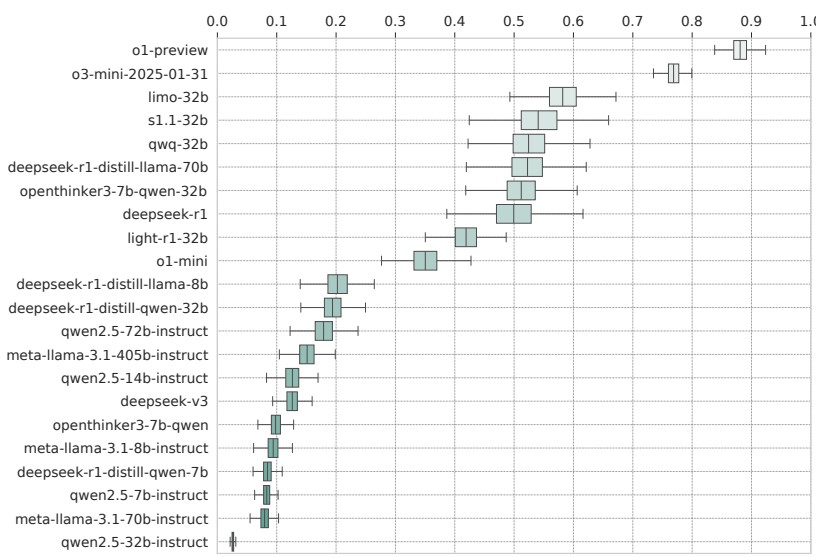

Figure 12: **Reasoning models outperform conventional LLMs.** Albeit suffering from strong fluctuations, larger-scale distilled reasoning models set themselves apart from the conventional language models from which they were distilled. Shown are correct response rates averaged across all variations of AIW Friends, AIW Plus, and AIW Circles Colleagues problems Nezhurina et al. (2024). Larger reasoning models on a 32B scale are populating the upper correct response rate range (the only exception being R1-Qwen-32B). Conventional LLMs, including the largest scale Llama 3.1 405B and DeepSeek v3 671B, stay confined to the low correct response rate region below 0.2. As a reference, we show closed reasoning models o3-mini and o1-preview that show only weak fluctuations and settle in the upper performance region above 0.7%

## L   COMPUTE REQUIREMENTS

The three main compute requirements for this effort are for annotation, training, and evaluation. Annotating OpenThoughts3-1.2M with QwQ-32B required 22,000 H100 GPU hours on 16 1xNvidia GH200 nodes. One single run of training OpenThinker3-7B required 25,000 A100 GPU hours on 128 nodes, each equipped with 4x Nvidia A100 GPUs (512 GPUs in total). One evaluation run for OpenThinker3-7B required 32 GPU-hours on 16 1xNvidia GH200 nodes. Throughout the pipeline experiments, annotating each 31,600-size dataset cost roughly $300 in API costs through the DeepSeek API.

## M   SOURCING REASONING TRACES FROM THE WEB

In this work, we generate reasoning traces with annotator models. As an alternative, we also experimented with finding reasoning traces from the web, then processing them with language models in order to bring them in a suitable form for SFT, following Yue et al. (2024), which generated instruction-finetuning data in that manner.

We searched for long reasoning traces in DCLM-RefinedWeb (Li et al., 2024), which is a large text corpus sourced from CommonCrawl text data heuristically filtered and deduplicated. We trained a FastText classifier by taking as positive data the reasoning traces from OpenThoughts-114K, and as negative data an equal amount of text from DCLM-RefinedWeb. We then examined some of the sequences that are most similar to the reasoning traces according to the FastText classifier (see Figure 13 for an example), and found that those are not similar to the long reasoning traces from OpenThoughts-114K. This indicates that CommonCrawl does not contain large amounts of long reasoning traces similar to those used by current reasoning models. While CommonCrawl can contains useful questions and answers that can serve a base for effective instruction-answer pairs for instruction finetuning and as question and answers as a bases for reasoning traces, it does not seem to contain many long reasoning traces.

```
Algebra 1
Published by Prentice Hall
ISBN 10: 0133500403
ISBN 13: 978-0-13350-040-0

Chapter 4 - An Introduction to Functions
Chapter Review - 4-2 Patterns and Linear Functions
Page 282:8

Domain: 0, 1, 2, 3
Range: 18, 21, 24, 27

Work Step by Step:
The relationship is between the number of snacks
purchased and the total cost.
If 0 snack is purchased, then the cost is 18.
If 1 snack is purchased, then the cost is 21.
If 2 snacks are purchased, the cost is 24.
If 3 snacks are purchased, then the cost is 27.
We can see a pattern in the range:
each cost term is separated by 3.
So, we can assume that each snack costs 3.00.
Update this answer! Update this answer
```

Figure 13: **Example of a reasoning trace found in DCLM-Baseline.** This is an example reasoning trace found in DCLM-Baseline. This text does not resemble reasoning traces from modern reasoning models.

## N    LICENSES OF EXISTING ASSETS

- **Qwen-2.5-7B-Instruct** model is distributed under Apache 2.0 license as indicated in Qwen-2.5-7B-Instruct. (Yang et al., 2024a).
- **Open2Math** dataset is distributed under Creative Commons Attribution 4.0 license as indicated in openmath-2-math. (Toshniwal et al., 2024b).
- **StackExchange CodeGolf** dataset is distributed under cc-by-sa 4.0 license as indicated in StackExchange Data Dump. (Stack Exchange, 2024).
- **OpenCodeReasoning** dataset is distributed under cc-by-4.0 license as indicated in Open-CodeReasoning. (Ahmad et al., 2025a).
- **StackExchange Physics** dataset is distributed under cc-by-sa 4.0 license as indicated in StackExchange Data Dump. (Stack Exchange, 2024).
- **OpenAI Models** are distributed under OpenAI Terms of Use.
- **QwQ-32B** model is distributed under Apache license 2.0 as indicated in QwQ-32B. (Yang et al., 2024b).
- **SCP-116K** dataset is distributed under cc-by-nc-sa-4.0 license as indicated in SCP-116K. (Lu et al., 2025).
- **Organic Chemistry** by Jonathan Clayden, Nick Greeves, and Stuart Warren has all rights reserved. (Clayden et al., 2012b).
- **Organic Chemistry/Fourth Edition** by Francis A. Carey has all rights reserved. (Carey, 1996).
- **Organic Chemistry** by John McMurry is distributed under the Creative Commons Attribution Non-Commercial ShareAlike 4.0 International License. (McMurry, 2023).
- **Principles of Organic Chemistry** by James Flack Norris has standard copyright (Norris, 1922).
- **Organic Chemistry** by Robert V. Hoffman with all rights reserved (Hoffman, 2004).
- **March's Advanced Organic Chemistry** by Michael B. Smith and Jerry March has all rights reserved. (Smith & March, 2007).
- **Essentials of Organic Chemistry** by Paul M. Dewick has all rights reserved. (Dewick, 2011).
- **Fundamentals of Organic Chemistry** by John McMurry has all rights reserved. (McMurry, 2010).
- **Advanced Organic Chemistry** by Francis A. Carrey and Richard J. Sundberg have all rights reserved. Carey & Sundberg (2007).
- **Introduction to Organic Chemistry** has no license.
- **Principles and Techniques** by Unknown Author is distributed under NCERT - Educational Use Permitted.
- **Organic Chemistry** by Francis A. Carey has all rights reserved. (Carey, 2010).
- **A Concise Textbook of Organic Chemistry** by C. G. Lyons, S. McClintock, and Nora Lumb with all rights reserved. (Lyons et al., 2016).
- **The Practical Methods of Organic Chemistry** by Ludwig Gatterman with all rights reserved and Community Resource - Educational Use. (Gattermann et al., 1917).
- **Modern Methods of Organic Synthesis** by W. Carruthers and Iain Coldham has all rights reserved. (Carruthers & Coldham, 1978).
- **Organic Chemistry** by J. Clayden, Greeves, Warren, and Wothers has all rights reserved. (Clayden et al., 2012a).
- **Techniques in Organic Chemistry** by Jerry R. Mohrig, Christina Hammond, and Paul Schatz has all rights reserved. (Mohrig et al., 2010).
- **Organic Chemistry** by David J. Hart, Christopher Hadad, Leslie Craine, and Harold Hart has all rights reserved. (Hart et al., 2011).

## O  Pipeline Details

In this Section we go into more details for each step of our pipeline, which we briefly described in Section 3.

### O.1  Question Generation Strategies

We will now go over the different ways we generated questions.

#### O.1.1  Code Question Generation Strategies

We begin by detailing all the code question generation strategies:

- **StackExchange CodeGolf** (Number of Questions: 85.9K): A StackExchange forum of coding puzzles, specifically aimed at solutions with the least number of characters possible. (Stack Exchange, 2024).

- **OpenCodeReasoning** (Number of Questions: 459K) A large reasoning-based synthetic dataset to date for coding, comprises 735,255 samples in Python across 28,319 unique competitive programming questions (Ahmad et al., 2025a).

- **cognitivecomputations/dolphin-coder** (Number of Questions: 101K): Synthetic questions evolved from LeetCode questions. (Hartford et al., 2024).

- **m-a-p/CodeFeedback-Filtered-Instruction** (Number of Questions: 150K): Mixture of synthetic and real coding questions filtered by an LLM. (Zheng et al., 2024).

- **KodCode/KodCode-V1** (Number of Questions: 384K): Fully synthetic and diverse coding dataset with questions ranging from algorithmic to package specific knowledge. (Xu et al., 2025).

- **Multilingual-Multimodal-NLP/McEval-Instruct** (Number of Questions: 35.8K): Multilingual code dataset on code-understanding, completion, and generation. (Chai et al., 2024).

- **christopher/rosetta-code** (Number of Questions: 75.4K): Multilingual code dataset on basic coding exercises. (Code, 2022).

- **glaiveai/glaive-code-assistant-v3** (Number of Questions: 946K): Code problems and solutions generated using Glaive's synthetic data generation platform. (Glaive, 2023).

- **StackExchange CodeReview** (Number of Questions: 183K): Code review questions from codereview.meta.stackexchange.com. (Stack Exchange, 2024).

- **prithivMLmods/Coder-Stat** (Number of Questions: 41.9K): Coding dataset for the analysis of coding patterns, error types, and performance metrics. We transform code and an associated error into a question for the LLM to solve using GPT-4o-mini with the prompt in Figure 14. (prithivMLmods, 2024).

- **OpenCoder-LLM/opc-sft-stage2**: A mixture of synthetic python questions generated from python documentation, educational material and more. We use both OpenCoder-LLM/opc-sft-stage2 and OpenCoder-LLM/opc-sft-stage1. Specifically, we use the package_instruct subset of OpenCoder-LLM/opc-sft-stage2 and the filtered_infinity_instruct, largescale_diverse_instruct, and realuser_instruct subsets of OpenCoder-LLM/opc-sft-stage1. (Huang et al., 2024).

- **ise-uiuc/Magicoder-OSS-Instruct-75K** (Number of Questions: 73.4K): Coding instruction-tuning set generated with gpt-3.5-turbo-1106. (Wei et al., 2023).

```
You are to generate a question or task for a language model based on
    the following error and code pairs.

Error Type: {{original_status}}
Code: {{original_src}}

Include only the new question and task. Do not include anything like "
    Here is the instruction". Include the code in your question and
    make the task sound like what a human would ask a language model.
```

Figure 14: **Coder Stat Prompt**

- **codeparrot/apps** (Number of Questions: 3.7K): Python dataset for generating code from natural language specifications. (Hendrycks et al., 2021a).
- **ajibawa-2023/Code-290k-ShareGPT** (Number of Questions: 283K): Human-asked questions to ChatGPT regarding code. (ajibawa 2023, 2023).
- **nampdn-ai/tiny-codes** (Number of Questions: >1M): Dataset of coding questions from textbooks transformed into questions using an LLM. (Panda et al., 2024).
- **bigcode/commitpackft** (Number of Questions: >1M): Commits on GitHub turned into coding questions using an LLM. We specifically look at the Python, C++, Java, C, C#, CSS, JavaScript, Shell, and Ruby commits. We ask GPT-4o-mini to turn a pair of commit message and code into a question using GPT-4o-mini and the prompt in Figure 15 (Muennighoff et al., 2023).
- **deepmind/code_contests** (Number of Questions: 8.8K): Competitive programming questions. (Li et al., 2022).
- **SenseLLM/ReflectionSeq-GPT** (Number of Questions: 9.7K): Python dataset with questions formed using compiler feedback with an LLM. Ren et al. (2024). (Sun et al., 2024).
- **MatrixStudio/Codeforces-Python-Submissions** (Number of Questions: 538K): Set of programming coding questions from CodeForces website. (Penedo et al., 2025b).
- **bigcode/self-oss-instruct-sc2-exec-filter-50k** (Number of Questions: 47.6K): Questions generated by using an LLM to turn code snippets from GitHub into difficult questions. (BigCode, 2024).
- **Magpie-Align/Magpie-Qwen2.5-Coder-Pro-300K-v0.1** (Number of Questions: 299K): Coding questions generated by letting Qwen2.5 Coder 32B Instruct generate coding questions. (Magpie-Align, 2025).
- **PrimeIntellect/real-world-swe-problems** (Number of Questions: 69.6K): Real SWE problems generated by PrimeIntellect. (PrimeIntellect, 2025).
- **StackExchange StackOverflow**: Coding questions from the StackOverflow online forum. (Stack Exchange, 2024).
- **cfahlgren1/react-code-instructions** (Number of Questions: 70.4K): LLM generated questions regarding the React framework. (Fahlgren, 2025).
- **PrimeIntellect/stackexchange-question-answering** (Number of Questions: 309K): Curated questions from StackExchange StackOverflow. (Prime Intellect, 2025).
- **PrimeIntellect/synthetic-code-understanding** (Number of Questions: 59.9K): Coding questions to teach an LLM to predict the output of coding snippet. (PrimeIntellect, 2025).
- **bugdaryan/sql-create-context-instruction** (Number of Questions: 78.6K): Coding questions about SQL from the WikiSQL and Spider forums. (b mc2, 2023).

```
You are to generate a question or task for a language model based on
    the following instruction and code pairs.

Instruction: {{message}}
Code: {{old_contents}}

Include only the new question and task. Do not include anything like "
    Here is the instruction". Include
the code in your question and make the task sound like what a human
    would ask a language model.
```

Figure 15: **CommitPack Prompt**

### O.1.2 MATH QUESTION GENERATION STRATEGIES

We also detail all the math question generation strategies:

- **ai2-adapt-dev/openmath-2-math** (Number of Questions: >1M): The MATH subset of OpenMathInstruct2 (Allen Institute for AI, 2025).

- **AI-MO/NuminaMath-1.5** (Number of Questions: 853K): Scanned math problems from competition math problem sources. (LI et al., 2024).

- **GAIR/MathPile** (Number of Questions: 99.5K): Math text shards that become seed information for generating math questions with GPT-4o-mini. Specifically, MathPile ontains unstructured text about topics in math. GPT-4o-mini uses the prompt in Figure 16 to turn each text into a question. (Wang et al., 2024).

- **MetaMath-AIME** (Number of Questions: >1M): The MetaMath pipeline applied to the AIME and AOPS sections of NuminaMATH. We reproduce this pipeline using GPT-4o-mini since the original MetaMath dataset was based on GSM8K and MATH train sets (Yu et al., 2023).

- **math-ai/AutoMathText** (Number of Questions: >1M): Math text shards that become seed information for generating math questions with GPT-4o-mini. Specifically, math-ai/AutoMathText contains unstructured text about topics in math. GPT-4o-mini uses the prompt in Figure 16 to turn each text into a question. (Zhang et al., 2025).

- **OpenMathInstruct2-AIME** (Number of Questions: >1M): The OpenMathInstruct pipeline applied to the AIME and AOPS sections of NuminaMath. We only do the question augmentation part of the OpenMathInstruct pipeline and use GPT-4o-mini for our augmentation. (Toshniwal et al., 2024b).

- **zwhe99/DeepMath-103K** (Number of Questions: 95.9K): Curated math questions from several different sources filtered for difficulty. (He et al., 2025b).

- **TIGER-Lab/MathInstruct** (Number of Questions: 256K): Mixture of existing math datasets with questions generated with LLMs and Common-Crawl. (Yue et al., 2023).

- **nvidia/OpenMathInstruct-2** (Number of Questions: >1M): Synthetic questions sourced from MATH and GSM8K train sets. (Toshniwal et al., 2024b).

- **ddrg/named_math_formulas** (Number of Questions: >1M): Math text shards that become seed information for generating math questions with GPT-4o-mini. Specifically, we take each formula and put it in the prompt for Figure 17 and ask GPT-4o-mini to form a question. (Drechsel et al., 2025).

- **facebook/natural_reasoning** (Number of Questions: >1M): High-quality challenging reasoning questions backtranslated from DCLM and FineMath pre-training corpora. (Yuan et al., 2025).

- **SynthLabsAI/Big-Math-RL-Verified** (Number of Questions: 45.6K): Heavily filtered verifiable math questions. (Albalak et al., 2025).

```
You are to reform the following math text snippet into a question with
    a quantitative or verifiable answer such as one that would be
included in the USAMO or the Putnam Exam.

Text: {{text}}

Include only the new question or task. Do not include anything like "
    Here is the instruction". You can either extract a
question from the text or form a new one based on the text. Make the
    question sound like what a human would ask a language model.
```

Figure 16: **AutoMathText Prompt**

```
You are to reform the following math text snippet into a question with
    a quantitative or verifiable answer such as one that would be
included in the USAMO or the Putnam Exam.

Text: {{formula}

Include only the new question or task. Do not include anything like "
    Here is the instruction". You can either extract a
question from the text or form a new one based on the text. Make the
    question sound like what a human would ask a language model.
```

Figure 17: **Formulas**

- **Asap7772/hendrycks-math-mc-llama** (Number of Questions: 79.9K): No details provided. (Asap7772, 2025).
- **TIGER-Lab/MATH-plus** (Number of Questions: 847K): Mixture of MetaMath, MATH-orca and some additional MATH-augmented dataset with GPT-4. (TIGER-Lab, 2025).
- **ibivibiv/math_instruct** (Number of Questions: >1M): No information provided. (ibivibiv, 2023).
- **BAAI/InfinityMATH** (Number of Questions: 99.9K): A scalable instruction tuning dataset for programmatic mathematical reasoning. (Zhang et al., 2024a).
- **ajibawa-2023/Maths-College** (Number of Questions: 937K): Questions spanning a diverse domains of college level mathematics. (Ajibawa, 2023).
- **MetaMath** (Number of Questions: >1M): Our reproduction of MetaMath. The original MetaMath was built with GPT-3.5-turbo. We replace this with GPT-4o-mini in our pipeline (Yu et al., 2023).
- **allenai/math_qa** (Number of Questions: 29.7K): Math word problems sourced from AQuA-RAT. (Amini et al., 2019).
- **deepmind/math_dataset** (Number of Questions: 1M): Math questions at roughly a school level. We specifically use questions from the algebra__linear_2d_composed, probability__swr_p_level_set, polynomials__evaluate_composed, polynomials__simplify_power, calculus__differentiate_composed and probability__swr_p_sequence subsets. (Saxton et al.).
- **Lap1official/Math** (Number of Questions: >1M): No information provided. (Lap1official, 2024).

### O.1.3 SCIENCE QUESTION GENERATION STRATEGIES

We also detail all the science question generation strategies:

- **StackExchange Physics** (Number of Questions: 547K): Questions from https://physics.stackexchange.com, the Physics StackExchange Forum. (Stack Exchange, 2024).

- **Organic Chemistry PDF Pipeline** (Number of Questions: 46.2K): LLM extracted organic chemistry questions from SCP PDFs and more Organic Chemistry Textbooks. We start the PDFs from SCP-116k alongside organic chemistry textbooks, solution manuals, and more. We use gemini/gemini-2.0-flash-lite-preview-02-05 with the prompt in Figure 22 to extract the text from the PDFs. GPT-4o-mini then uses the prompt in Figure 21 to extract the question and answers from each page of the extracted text. GPT-4o-mini then refines the questions into cleaner questions using the prompt in Figure 20. GPT-4o-mini then filters out questions not related to math, science, and code using a prompt in Figure 19. We use structured decoding to get a classification as a boolean and a reasoning as a string. GPT-4o-mini then further filters questions not related to organic chemistry with the prompt in Figure 18. We use the same structured decoding technique for the organic chemistry filtering as we did for math, science, and code questions.

- **mteb/cqadupstack-physics** (Number of Questions: 38.3K): A dataset for community question-answering research focussed on physics. We use the prompt in Figure 23 to turn each unstructured text into a question with GPT-4o-mini (Hoogeveen et al., 2015).

- **Camel-AI/Physics** (Number of Questions: >1M): Our reproduction of the Physics questions from the Camel pipeline. We reproduce this pipeline from scratch using GPT-4o-mini. (Li et al., 2023b).

- **Josephgflowers/Par-Four-Fineweb-Edu-Fortified-Chemistry-Physics-Astronomy-Math-Reason** (Number of Questions: 988K): LLM generated questions from science text on FineWeb. We use the prompt in Figure 23 to turn each unstructured text into a question with GPT-4o-mini (Flowers, 2024).

- **millawell/wikipedia_field_of_science** (Number of Questions: 304K): LLM generated questions from Wikipedia science articles. We use the prompt in Figure 23 to turn each unstructured text into a question with GPT-4o-mini. (millawell, 2024).

- **zeroshot/arxiv-biology** (Number of Questions: 1.2K): LLM generated questions using Arxiv Biology papers. We take the abstracts from the original source use the prompt in Figure 24 to turn the abstract into a question using GPT-4o-mini. (Clement et al., 2019).

- **Camel-AI/Chemistry** (Number of Questions: >1M): Our reproduction of the chemistry questions from the Camel pipeline. We reproduce this pipeline from scratch using GPT-4o-mini. (Li et al., 2023b).

- **StackExchange Biology** (Number of Questions: 60.3K): Questions from https://biology.stackexchange.com, the Biology StackExchange Forum. (Stack Exchange, 2024).

- **Camel-AI/Biology** (Number of Questions: >1M): Our reproduction of the biology questions from the Camel pipeline. We reproduce this pipeline from scratch using GPT-4o-mini. (Li et al., 2023b).

- **AdapterOcean/biology_dataset_standardized_unified**: (Number of questions: 22K) No information provided. (AdapterOcean, 2024).

```
Yes or No, is this question an organic chemistry question?
Question:
{{problem}}
```

Figure 18: **Organic Chemistry Filtering**

```
Yes or No, is the question an answerable difficult science, math, or
    coding question? If the question refers to content (figures,
    equations, additional text) that you cannot see, then it is
    unanswerable.
Provide your reasoning.
Question:
{{improved_question_solution}}
```

Figure 19: **Code, Math, and Science Question Filtering Prompt**

```
You are an instructor creating exam questions.
I will provide you with a given question and the text from which it was extracted from.
You will ensure that the question is answerable, meaning that there is enough context to answer the
    question.
To do this, you will look at the extracted text and ensure that nothing is missing from the current
    questions instantiation. If there is, you will provide the new extra text before restating the
    question but be sure you always add the question itself at the end.
Because you are an instructor creating exam questions, you will never include the solution in the extra
    text or question.

Here is an example of your task:
Extracted Question: Calculate the chemical amount (in mol or mmol) of nitric acid that reacts with the
5.000 g sample of this mineral.
Extracted Text: A sample of a different mineral is analysed by the same methods. This mineral also
    contains only Pb2, CO3, OH, and O2 ions.
When a 5.000 g sample of this mineral is treated with 25.00 mL of 2.000 mol L nitric acid
(HNO3), 0.5214 g of carbon dioxide is released, and 0.01051 mol of the acid remains.
When subjected to thermal decomposition, 5.000 g of this mineral loses 0.5926 g.
(g) Calculate the chemical amount (in mol or mmol) of nitric acid that reacts with the
5.000 g sample of this mineral.

You would tell me:

A sample of a different mineral is analysed by the same methods. This mineral also contains only Pb2, CO3
    , OH, and O2 ions.
When a 5.000 g sample of this mineral is treated with 25.00 mL of 2.000 mol L nitric acid
(HNO3), 0.5214 g of carbon dioxide is released, and 0.01051 mol of the acid remains.
When subjected to thermal decomposition, 5.000 g of this mineral loses 0.5926 g. Calculate the chemical
    amount (in mol or mmol) of nitric acid that reacts with the
5.000 g sample of this mineral.

---

Here is the question: {{extracted_question}}
Here is the extracted text: {{output_extraction}}

---

Do not include any filler like "here is the improved question". Include only the relevant information and
    the question itself. Include all answer choices if applicable.
Do not include the solution if you see it. This is an exam, so you should NOT include the final answer in
    the question.
```

Figure 20: **Question Refinement Prompt**

```
Extract the questions, answer choices, and solutions from the extracted text from the pdf below.

Format your response as below:
QUESTION: "the question from the text and all relevant context excluding the answer and choices" (i.e. if
    the question is "What was Elvis Presley's (a) favorite sandwich? (b) most hated song? (c) birthday
    ?" You should have a question "QUESTION: What was Elvis Presley's favorite sandwich..." and then
    another question "QUESTION: What was Elvis Presley's most hated song?..." etc.)
ANSWER CHOICES: "answer choice 1" ||| "answer choice 2" ... (if no answer choices are available just say
    "free response", if answer choices are available please write them out i.e. "A: 0.57 ||| B: John
    Hancock ||| ..." etc.)
SOLUTION: "correct answer choice" or "free response answer". If you cannot determine the correct solution
    say "NO ANSWER DETECTED"

It is important that each QUESTION: is self contained, meaning, if you were to read "QUESTION: ..." by
    itself, you should be able to answer it. Given the example "What was Elvis Presley's (a) favorite
    sandwich? (b) most hated song? (c) birthday?", you should NOT say "QUESTION: favorite sandwich?" as
    this makes no sense. Instead you should say "QUESTION: What was Elvis Presley's favorite sandwich
    ?"
To be clear. All questions with subquestions should be restated as individual questions themselves with
    all relevant information required to answer them.
Only break up the subquestions. I.E. only break up questions that have (a) (b) and (c) or (1) (2) and (3)
    , do not break up questions that do not have markers indicating they have parts or subquestions
    please.
We are creating a bank of questions that are automatically extracted from pdfs, so it is imperative you
    get this right.
---
Extracted Text:

{{output_extraction}}
```

Figure 21: **Question Extraction Prompt**

```
Extract all text from this PDF, including all text from images. Example extractions could look like this:
html><body><table><tr><td>2</td><td>1 1</td><td>1 6</td><td>-14</td></tr><tr><td rowspan="2"></td><td>2
    3</td><td>7</td><td>14 0</td></tr><tr><td>1</td><td></td><td></td></tr></table></body></html>

Notice that the final sum is zero, telling us that 2 is a zero. But the nice thing about synthetic
    division, is that long division, the other coefficients are the coefficients of the quotient
    polynomial.

The Remainder Theorem states that when dividing a polynomial by a factor of the form $(x-a)$ there is a
    quotient polynomial and a constant remainder. The remainder is P(a) since $P(x)=(x-a)\cdot Q(x)+r\
    Rightarrow P(a)=(a-a)Q(a)+r=r.$ .

# Miscellaneous Facts;

It is obvious in any polynomial that $\mathrm(0)$ is the y-intercept and equally as obvious that $\mathrm
    (1)$ is the sum of all the coefficients.

# Problems:

1. Given the cubic polynomial $P(x)=x^-7x^-4x+28$ . Two of the zeros are additive inverses. Find the
    zeros.
3. If $\mathrm(\mathbf)$ is a polynomial with rational coefficients and roots at 0, 1, $\sqrt$ , and $1
    -(\sqrt(3))$ , then the degree of $\mathfrak(p)(\ensuremath(\mathbf(x)))$ is at least:
4. When Madison's dog chewed up her mathematics assignment, one particular equation was ripped apart. I
    found a piece of the beginning of the equation and a piece at the end, but the middle was missing.
    The beginning piece was $x^(5)-9x^(4)+$ and the ending piece was $+11=0$ . Fortunately the teacher
    had promised that all of the roots would be integers. How many times is $^(-1)$ a root? [Furman
    ????]
5. The following is a polynomial. Find the sum of the squares of its coefficients. $\sqrt[3](x^(9)-3x^(8)
    +18x^(7)-28x^(6)+84x^(5)-42x^(4)+98x^(3)+72x^+15x+1)$ . FURMAN
6. If a cubic polynomial $\operatorname(p)(\mathbf(x))$ has roots at -1, 2, and 3, and if $\mathfrak(p)
    (0)=1$ , then the remainder when $\mathfrak(p)(\ensuremath(\mathbf(x)))$ is divided by $\mathbf(X)
    -1$ is:
7. If 2 is a solution of $x^(3)+h x+10=0$ , then h equals:
8. The number of distinct real solutions of the equation $4x^(3)-8x^(2)+5x-1=0$ is:
9. What is the sum of the squares of the roots of $x^(4)-5x^(2)+6=0$
10. For how many integers $_\mathrm(N)$ is $N^(4)+6N<6N^(3)+N^(2)$ ?
11. How any times does the graph of $f(x)=x^(3)-x^(2)+2x+4$ cross the $\mathbf(X)$ axis?
12. Madison's dog chewed on her homework before she could finish it. The fragment saved from the horrible
    canine's mouth reveal only the two terms of highest degree of the polynomial $\mathfrak(p)(\
    ensuremath\mathbf(x)))$

Now please give me your extraction of all text, including text in images.
```

Figure 22: **Gemini OCR Prompt**

```
You are to reform the following math text snippet into a question with
    a quantitative or verifiable answer such as one that would be
included in the Biology, Physics or Chemistry Olympiad.
Text: {{text}}

Include only the new question or task. Do not include anything like "
    Here is the instruction". You can either extract a
question from the text or form a new one based on the text. Make the
    question sound like what a human would ask a language model.
```

Figure 23: **Prompt for Generating Science Questions**

```
You are to reform the following math text snippet into a question with
    a quantitative or verifiable answer such as one that would be
included in the Biology, Physics or Chemistry Olympiad.
Text: {{abstract}}

Include only the new question or task. Do not include anything like "
    Here is the instruction". You can either extract a
question from the text or form a new one based on the text. Make the
    question sound like what a human would ask a language model.
```

Figure 24: **Prompt for Generating Science Questions**

## O.2 Information on question filtering strategies

We will now provide more information on our question filtering strategies.

### O.2.1 FastText Details

We provide some more details on our FastText filters.

- Our hidden dimension of the FastText Filter was 256.
- We train the classifier for 3 epochs.
- We use a learning rate of $0.1$.
- We use bigrams or $n = 2$.
- We use a minimum $n$-gram count of 3.

### O.2.2 Code Filtering Strategies

We provide more details about our code filtering strategies.

- **Difficulty-based Selection**: Ask GPT-4o-mini to rate the question on a scale of 1 to 10 using a rubric for ICPC problems and take the hardest rated problems. When asking GPT-4o-mini to help with question filtering, we only use temperature set to $1.0$. We do not set other sampling hyperparameters. The prompt for difficulty filtering is in Figure 25. We use structured decoding to extract a numerical response from GPT-4o-mini. For AskLLM Filtering, we request a numerical response as an integer and a string response for reasoning.
- **Length-based Selection (GPT-4.1-nano)**: Annotate questions with GPT-4.1-Nano and keep the questions with longest responses.
- **AskLLM Selection**: Ask GPT-4o-mini to rate on a scale of 1 to 100 how similar a question is to a set of good questions and different from a set of bad questions. When asking GPT-4o-mini to help with question filtering, we only use temperature set to $1.0$. We do not set other sampling hyperparameters. The prompt for AskLLM filtering is in Figure 26. We use structured decoding to extract a numerical response from GPT-4o-mini. For AskLLM Filtering, we request a numerical response as an integer and a string response for reasoning.
- **Length-based Selection (GPT-4o-mini)**: Annotate questions with GPT-4o-mini and keep the questions with longest responses.
- **FastText (P: Codeforces; N: CodeReview)**: Classify questions with a FastText classifier trained with positives that are CodeForces and negatives that are questions from StackExchange Code Review. More info is in Appendix O.2.1.
- **Length-based Selection (GPT-4.1-mini)**: Ask GPT-4.1-mini to rate on a scale of 1 to 100 how similar a question is to a set of good questions and different from a set of bad questions.
- **Random Selection**: Randomly select questions.
- **FastText (P: LeetCode; N: SQL)**: Classify questions with a FastText classifier trained with positives that are from LeetCode and negatives that are questions from bugdaryan/sql-create-context-instruction. More info is in Appendix O.2.1.
- **FastText (P: Code Golf; N: SQL)**: Classify questions with a FastText classifier trained with positives that are from StackExchange Code Golf and negatives that are questions from bugdaryan/sql-create-context-instruction. More info is in Appendix O.2.1.
- **FastText (P: All; N: SQL)**: Classify questions with a FastText classifier trained with positives that are from CodeForces, LeetCode, Code Golf, and IOI and negatives that are questions from bugdaryan/sql-create-context-instruction. More info is in Appendix O.2.1.
- **FastText (P: IOI; N: SQL)**: Classify questions with a FastText classifier trained with positives that are from IOI and negatives that are questions from bugdaryan/sql-create-context-instruction. More info is in Appendix O.2.1.
- **FastText (P: Codeforces; N: All)**: Classify questions with a FastText classifier trained with positives that are CodeForces and negatives that are questions from StackExchange Code Review and bugdaryan/sql-create-context-instruction. More info is in Appendix O.2.1.

- **FastText Selection (P: Codeforces; N: SQL)**: Classify questions with a FastText classifier trained with positives that are CodeForces and negatives that are questions from bugdaryan/sql-create-context-instruction. More info is in Appendix O.2.1.

- **Embedding-based Selection**: Embed a set of positives, which are CodeForces, and a set of negatives, which are bugdaryan/sql-create-context-instruction, using OpenAI's embedding model, text-embedding-3-large. First, embed a given question and measure its mean cosine similarity to positives and mean cosine similarity to negatives, and generate the final score by taking the difference.

### O.2.3 MATH QUESTION FILTERING

- **Length-based Selection (GPT-4.1-nano)**: Annotate questions with GPT-4.1-Nano and keep the questions with longest responses.

- **AskLLM Selection**: Ask GPT-4o-mini to rate on a scale of 1 to 100 how similar a question is to a set of good questions and different from a set of bad questions. When asking GPT-4o-mini to help with question filtering, we only use temperature set to 1.0. We do not set other sampling hyperparameters. The prompt for AskLLM filtering is in Figure 26. We use structured decoding to extract a numerical response from GPT-4o-mini. For AskLLM Filtering, we request a numerical response as an integer and a string response for reasoning.

- **Random Selection**: Randomly select questions.

- **Embedding-based Selection**: Embed a set of positives, which are the "amc_aime" and "olympiads" subsets of AI-MO/NuminaMath-CoT, and a set of negatives, which is Lap1official/Math using OpenAI's embedding model, text-embedding-3-large. First, embed a given question and measure its mean cosine similarity to positives and mean cosine similarity to negatives, and generate the score by taking the difference.

- **FastText (P: S1.1; N: Lap1official)**: Classify questions with a FastText classifier trained with positives that are questions from S1 and negatives that are questions from Lap1official/Math_small_corpus. More info is in Appendix O.2.1.

- **FastText (P: Olympiad; N: Lap1official)**: Classify questions with a FastText classifier trained with positives that are questions from brando/olympiad-bench-imo-math-boxed-825-v2-21-08-2024 and negatives that are questions from Lap1official/Math_small_corpus. More info is in Appendix O.2.1.

- **Difficulty-based Selection**: Ask GPT-4o-mini to rate the question on a scale of 1 to 10 using a rubric for AOPS problems and take the hardest rated problems. When asking GPT-4o-mini to help with question filtering, we only use temperature set to 1.0. We do not set other sampling hyperparameters. The prompt for Difficulty filtering is in Figure 27. We use structured decoding to extract a numerical response from GPT-4o-mini. For AskLLM Filtering, we request a numerical response as an integer and a string response for reasoning.

- **FastText (P: OpenR1; N: Lap1official)**: Classify questions with a FastText classifier trained with positives that are questions open-r1/OpenR1-Math-220K and negatives that are questions from Lap1official/Math_small_corpus. More info is in Appendix O.2.1.

- **FastText (P: All; N: Lap1official)**: Classify questions with a FastText classifier trained with positives which are the "amc_aime" and "olympiads" subset of AI-MO/NuminaMath-CoT, brando/olympiad-bench-imo-math-boxed-825-v2-21-08-2024, open-r1/OpenR1-Math-220K, and S1 and negatives that is questions from Lap1official/Math_small_corpus. More info is in Appendix O.2.1.

- **FastText (P: Numina; N: All)**: Classify questions with a FastText classifier trained with positives, which are the "amc_aime" and "olympiads" subsets of AI-MO/NuminaMath-CoT, and negatives, that are questions from Lap1official/Math_small_corpus and facebook/natural_reasoning. More info is in Appendix O.2.1.

- **FastText (P: Numina; N: Natural Reasoning)**: Classify questions with a FastText classifier trained with positives, which are the "amc_aime" and "olympiads" subsets of AI-MO/NuminaMath-CoT, and negatives, that are questions from facebook/natural_reasoning. More info is in Appendix O.2.1.

```
You will be given a code problem. Your job is to grade the difficulty
    level from 1-10 according to the ICPC standard.
Here is the standard:
A 10-point scale for ICPC problems could be structured as follows,
    where level 1 represents the easiest problems and level 10
    represents the most challenging:
Level 1: Basic implementation problems requiring simple input/output
    handling and straightforward calculations. Typically solvable with
     a single loop or basic conditional statements. Examples include
    summing numbers or finding the maximum in an array.
Level 2: Problems involving basic data structures like arrays and
    strings, requiring simple algorithms like linear search or basic
    sorting. May include simple mathematical concepts like prime
    numbers or basic geometry.
Level 3: Problems requiring knowledge of standard algorithms like
    binary search, complete sorting algorithms, or basic graph
    traversal (DFS/BFS). May include simple dynamic programming
    problems with clear state transitions.
Level 4: Problems combining multiple basic concepts, requiring careful
     implementation and moderate optimization. Includes medium-
    difficulty dynamic programming problems and basic graph algorithms
     like shortest paths.
Level 5: Problems requiring solid understanding of data structures
    like segment trees, binary indexed trees, or disjoint set unions.
    May include more complex graph algorithms like minimum spanning
    trees or network flow basics.
Level 6: Advanced dynamic programming problems with non-obvious state
    representations. Problems requiring combination of multiple
    algorithms or data structures. May include basic game theory or
    basic number theory concepts.
Level 7: Problems requiring advanced algorithmic knowledge like heavy-
    light decomposition, suffix arrays, or advanced geometric
    algorithms. Includes complex optimization problems and harder
    network flow applications.
Level 8: Problems requiring deep mathematical insights combined with
    complex algorithmic implementations. May include advanced number
    theory, complex geometric algorithms, or problems requiring
    multiple non-obvious observations.
Level 9: Problems requiring extensive knowledge of advanced algorithms
     and mathematical concepts, often needing multiple key insights to
     solve. May include advanced string algorithms like suffix
    automata, or complex mathematical optimizations.
Level 10: The most challenging problems, often requiring novel
    approaches or insights not covered in standard competitive
    programming material. These problems might combine multiple
    advanced concepts in non-obvious ways, require complex proofs for
    correctness, or need highly optimized implementations to meet
    strict time limits.
This scale corresponds roughly to the difficulty progression you might
     see from early regional contests (levels 1-4) through regional
    finals (levels 4-7) to world finals problems (levels 7-10).
Problem to be labeled: {{question}}."
```

Figure 25: **Prompt for Code Difficulty Filtering.** This text is the prompt for Code Difficulty Filtering.

- **Length-based Selection (GPT-4o-mini)**: Annotate questions with GPT-4o-mini and keep the questions with longest responses.

```
I want you to judge whether the following question is like the positive examples provided or like the
    negative examples. Here are a few positive examples of questions:

Positive Questions
1. A plane contains $40$ lines, no $2$ of which are parallel. Suppose that there are $3$ points where
    exactly $3$ lines intersect, $4$ points where exactly $4$ lines intersect, $5$ points where exactly
     $5$ lines intersect, $6$ points where exactly $6$ lines intersect, and no points where more than
    $6$ lines intersect. Find the number of points where exactly $2$ lines intersect.
2. A spin-half particle is in a linear superposition0.8|\uparrow\rangle+0.6|\downarrow\rangle of its spin
    -up and spin-down states. If |\uparrow\rangle and |\downarrow\rangle are the eigenstates of \sigma_{
    z} , then what is the expectation value up to one decimal place, of the operator 10\sigma_{z}+5\
    sigma_{x} ? Here, symbols have their usual meanings.
3. An established group of scientists are working on finding solution to NP hard problems. They claim
    Subset Sum as an NP-hard problem. The problem is to determine whether there exists a subset of a
    given set S whose sum is a given number K. You are a computer engineer and you claim to solve this
    problem given that all numbers in the set are non-negative. Given a set S of size N of non-negative
     integers, find whether there exists a subset whose sum is K. Input First line of input contains T,
     the number of test cases. T test cases follow. Each test case contains 2 lines. First line
    contains two integers N and K. Next line contains N space separated non-negative integers (each
    less than 100000). 0 < T < 1000 0 < N < 1000 0 < K < 1000 Output Output T lines, one for each test
    case. Every line should be either 0 or 1 depending on whether such a subset exists or not. Example
    Input: 2 5 10 3 4 6 1 9 3 2 1 3 4 Output: 1 0
4. Let $S$ be the set of positive integer divisors of $20^9.$ Three numbers are chosen independently and
    at random with replacement from the set $S$ and labeled $a_1,a_2,$ and $a_3$ in the order they are
    chosen. The probability that both $a_1$ divides $a_2$ and $a_2$ divides $a_3$ is $\tfrac{m}{n},$
    where $m$ and $n$ are relatively prime positive integers. Find $m.$
5. What is the concentration of calcium ions in a solution containing 0.02 M stoichiometric Ca-EDTA
    complex (we assume that the pH is ideal, T = 25C). KCa-EDTA = 5x10\^10.

Negative Questions:
1. Solve 0 = 19*z - 17*z for z.
2. Simplify ((-2*(-2*sqrt(1210) - sqrt(1210) - sqrt(20)/sqrt(2)*-6))/((sqrt(1800)*2 + sqrt(1800) + sqrt
    (1800) + sqrt(1800))*-1)*3)**2.\n
3. Given a list of objects that have an `is_organized` method that returns a boolean value, write a
    Python function that takes the list and returns a new list of those objects for which `is_organized`
     returns True.
4. Can you provide a Python code snippet that demonstrates how to use a decorator to log the execution
    time of a function?
5. Is sodium hydroxide (NaOH) an acid or base?

Here is your question:
{{question}}

Return a score between 1 and 100, where 100 means exactly like the positive questions whereas 1 is
    exactly like the negative questions.
```

Figure 26: **Prompt for AskLLM Filtering.** This text is the prompt for AskLLM Filtering

- **Length-based Selection (GPT-4.1-mini)**: Annotate questions with GPT-4.1-mini and keep the questions with longest responses

- **FastText Selection (P: Numina; N: Lap1official)**: Classify questions with a FastText classifier trained with positives, which are the "amc_aime" and "olympiads" subsets of AI-MO/NuminaMath-CoT, and negatives, that is, questions from Lap1official/Math_small_corpus. More info is in Appendix O.2.1.

### O.2.4 SCIENCE QUESTION FILTERING

- **FastText (P: ExpertQA; N: Arxiv)**: Classify questions with a FastText classifier trained with positives that are questions from katielink/expertqa and negatives that are questions from AdapterOcean/biology_dataset_standardized_unified. More info is in Appendix O.2.1.

- **Length-based Selection (GPT-4o-mini)**: Annotate questions with GPT-4o-mini and keep the questions with longest responses.

- **Length-based Selection (GPT-4.1-nano)**: Annotate questions with GPT-4.1-Nano and keep the questions with longest responses.

- **Length-based Selection (GPT-4.1-mini)**: Annotate questions with GPT-4.1-mini and keep the questions with longest responses.

- **AskLLM Selection**: Ask GPT-4o-mini to rate on a scale of 1 to 100 how similar a question is to a set of good questions and different from a set of bad questions. When asking GPT-4o-mini to help with question filtering, we only use temperature set to 1.0. We do not set other sampling hyperparameters. The prompt for AskLLM filtering is in Figure 26. We

```
You will be given a math problem. Your job is to grade the difficulty level from 1-10 according to the AoPS standard. \
Here is the standard: \
All levels are estimated and refer to averages. The following is a rough standard based on the USA tier system AMC 8 – AMC 10 – AMC 12 – AIME – USAMO/USAJMO –
    IMO, \
representing Middle School – Junior High – High School – Challenging High School – Olympiad levels. Other contests can be interpolated against this. \
Notes: \
Multiple choice tests like AMC are rated as though they are free-response. Test-takers can use the answer choices as hints, and so correctly answer more AMC
    questions than Mathcounts or AIME problems of similar difficulty. \
Some Olympiads are taken in 2 sessions, with 2 similarly difficult sets of questions, numbered as one set. For these the first half of the test (questions
    1-3) is similar difficulty to the second half (questions 4-6). \
Scale \
1: Problems strictly for beginners, on the easiest elementary school or middle school levels (MOEMS, MATHCOUNTS Chapter, AMC 8 1-20, AMC 10 1-10, AMC 12 1-5,
    and others that involve standard techniques introduced up to the middle school level), most traditional middle/high school word problems. \
2: For motivated beginners, harder questions from the previous categories (AMC 8 21-25, harder MATHCOUNTS States questions, AMC 10 11-20, AMC 12 5-15, AIME
    1-3), traditional middle/high school word problems with extremely complex problem solving. \
3: Advanced Beginners problems that require more creative thinking (harder MATHCOUNTS National questions, AMC 10 21-25, AMC 12 15-20, AIME 4-6). \
4: Intermediate-level problems (AMC 12 21-25, AIME 7-9). \
5: More difficult AIME problems (10-12), simple proof-based Olympiad-style problems (early JBMO questions, easiest USAJMO 1/4). \
6: High-leveled AIME-styled questions (13-15). Introductory-leveled Olympiad-level questions (harder USAJMO 1/4 and easier USAJMO 2/5, easier USAMO and IMO
    1/4). \
7: Tougher Olympiad-level questions, may require more technical knowledge (harder USAJMO 2/5 and most USAJMO 3/6, extremely hard USAMO and IMO 1/4, easy-
    medium USAMO and IMO 2/5). \
8: High-level Olympiad-level questions (medium-hard USAMO and IMO 2/5, easiest USAMO and IMO 3/6). \
9: Expert Olympiad-level questions (average USAMO and IMO 3/6). \
10: Historically hard problems, generally unsuitable for very hard competitions (such as the IMO) due to being exceedingly tedious, long, and difficult (e.g.
    very few students are capable of solving on a worldwide basis). \
Examples \
For reference, here are problems from each of the difficulty levels 1-10: \
<1: Jamie counted the number of edges of a cube, Jimmy counted the numbers of corners, and Judy counted the number of faces. They then added the three numbers
    . What was the resulting sum? (2003 AMC 8, Problem 1) \
1: How many integer values of $x$ satisfy $|x| < 3\pi$? (2021 Spring AMC 10B, Problem 1) \
2: A fair $6$-sided die is repeatedly rolled until an odd number appears. What is the probability that every even number appears at least once before the
    first occurrence of an odd number? (2021 Spring AMC 10B, Problem 18) \
3: Triangle $ABC$ with $AB=50$ and $AC=10$ has area $120$. Let $D$ be the midpoint of $\overline{AB}$, and let $E$ be the midpoint of $\overline{AC}$. The
    angle bisector of $\angle BAC$ intersects $\overline{DE}$ and $\overline{BC}$ at $F$ and $G$, respectively. What is the area of quadrilateral $FDBG$?
    (2018 AMC 10A, Problem 24) \
4: Define a sequence recursively by $x_0=5$ and\[x_{n+1}=\frac{x_n^2+5x_n+4}{x_n+6}\]for all nonnegative integers $n.$ Let $m$ be the least positive integer
    such that\[x_m\leq 4+\frac{1}{2^{20}}.\]In which of the following intervals does $m$ lie? \
$\textbf{(A) } [9,26] \qquad\textbf{(B) } [27,80] \qquad\textbf{(C) } [81,242]\qquad\textbf{(D) } [243,728] \qquad\textbf{(E) } [729,\infty)$ \
(2019 AMC 10B, Problem 24 and 2019 AMC 12B, Problem 22) \
5: Find all triples $(a, b, c)$ of real numbers such that the following system holds:\[a+b+c=\frac{1}{a}+\frac{1}{b}+\frac{1}{c},\]\[a^2+b^2+c^2=\frac{1}{a
    ^2}+\frac{1}{b^2}+\frac{1}{c^2}.\](JBMO 2020/1) \
6: Let $\triangle ABC$ be an acute triangle with circumcircle $\omega,$ and let $H$ be the intersection of the altitudes of $\triangle ABC.$ Suppose the
    tangent to the circumcircle of $\triangle HBC$ at $H$ intersects $\omega$ at points $X$ and $Y$ with $HA=3,HX=2,$ and $HY=6.$ The area of $\triangle
    ABC$ can be written in the form $m\sqrt{n},$ where $m$ and $n$ are positive integers, and $n$ is not divisible by the square of any prime. Find $m+n.
    $ (2020 AIME I, Problem 15) \
7: We say that a finite set $\mathcal{S}$ in the plane is balanced if, for any two different points $A$, $B$ in $\mathcal{S}$, there is a point $C$ in $\
    mathcal{S}$ such that $AC=BC.$ We say that $\mathcal{S}$ is centre-free if for any three points $A$, $B$, $C$ in $\mathcal{S}$, there is no point $P$
    in $\mathcal{S}$ such that $PA=PB=PC.$ \
Show that for all integers $n\geq 3$, there exists a balanced set consisting of $n$ points. \
Determine all integers $n\geq 3$ for which there exists a balanced centre-free set consisting of $n$ points. \
(IMO 2015/1) \
8: For each positive integer $n$, the Bank of Cape Town issues coins of denomination $\frac1n$. Given a finite collection of such coins (of not necessarily
    different denominations) with total value at most most $99+\frac{1}{2}$, prove that it is possible to split this collection into $100$ or fewer
    groups, such that each group has total value at most $1$. (IMO 2014/5) \
9: Let $k$ be a positive integer and let $S$ be a finite set of odd prime numbers. Prove that there is at most one way (up to rotation and reflection) to
    place the elements of $S$ around the circle such that the product of any two neighbors is of the form $x^2+x+k$ for some positive integer $x$. (IMO
    2022/3) \
10: Prove that there exists a positive constant $c$ such that the following statement is true: Consider an integer $n > 1$, and a set $\mathcal S$ of $n$
    points in the plane such that the distance between any two different points in $\mathcal S$ is at least 1. It follows that there is a line $\ell$
    separating $\mathcal S$ such that the distance from any point of $\mathcal S$ to $\ell$ is at least $cn^{-1/3}$. \
(A line $\ell$ separates a set of points S if some segment joining two points in $\mathcal S$ crosses $\ell$.) (IMO 2020/6)
Problem to be labeled: {{question}}"
```

Figure 27: **Prompt for Math Difficulty Filtering.** This text is the prompt for Math Difficulty Filtering.

use structured decoding to extract a numerical response from GPT-4o-mini. For AskLLM Filtering, we request a numerical response as an integer and a string response for reasoning.

- **FastText (P: SciQ; N: Wikipedia w/ Arxiv)**: Classify questions with a FastText classifier trained with positives that are questions from allenai/sciq and negatives that is questions from AdapterOcean/biology_dataset_standardized_unified and questions generated from millawell/wikipedia_field_of_science as in Appendix O.1.3. More info is in Appendix O.2.1.

- **Embedding-based Selection**: Embed a set of positives, which are questions from allenai/sciq, and a set of negatives, which is AdapterOcean/biology_dataset_standardized_unified, using OpenAI's embedding model, text-embedding-3-large. First, embed a given question and measure its mean cosine similarity to positives and mean cosine similarity to negatives, and generate a score by taking the difference.

- **FastText (P: SCP, ExpertQA, SciQ; N: Arxiv)**: Classify questions with a FastText classifier trained with positives that are questions from allenai/sciq, EricLu/SCP-116K, and katielink/expertqa and negatives that are questions from AdapterOcean/biology_dataset_standardized_unified and questions generated from millawell/wikipedia_field_of_science as in Appendix O.1.3. More info is in Appendix O.2.1.

- **Random Selection**: Randomly select questions.

- **Difficulty-based Selection**: Ask GPT-4o-mini to rate the question on a scale of 1 to 10 using a rubric for science Olympiad problems and select the hardest-rated problems. When asking GPT-4o-mini to help with question filtering, we only use temperature set to 1.0. We do not set other sampling hyperparameters. The prompt for Difficulty filtering is in Figure 28. We

use structured decoding to extract a numerical response from GPT-4o-mini. For AskLLM Filtering, we request a numerical response as an integer and a string response for reasoning.

## O.3 DEDUPLICATION AND TEACHER SAMPLING

There are several methods for performing deduplication. We choose to use Indel Similarity to measure the similarity between two questions. This similarity refers to the number of characters that need to be inserted or deleted from one sample to match the other, and is calculated relative to the sample length. More precisely, we consider the Normalized Indel similarity score between a pair of strings, as computed via the Longest Common Subsequence (LCS) metric:

$$\text{indel}_{\text{sim}} = 100 \times \frac{\text{LCS}_{\text{length}}(s_1, s_2)}{\max(|s_1|, |s_2|)} \tag{2}$$

## O.4 QUESTION ANSWER FILTERING

We will now discuss the details of each question-answer filtering method.

### O.4.1 QUESTION ANSWER FILTERING FOR MATH

- **Comprehensive Large Dataset**: Take all 63,200 responses for one dataset without filtering. This is not compute-controlled.
- **Random Filtering**: Filtering questions-answer pairs randomly.
- **Shortest Answers Selection**: For any given question, take the 8 shortest responses out of the 16 responses and create 8 question-answer pairs for the dataset.
- **Majority Consensus Selection**: For any given question, provide all responses to GPT-4o-mini. Ask GPT-4o-mini to return a list of indices of responses that agree with the majority. Only use the last $1,000$ characters of each response to get the final answer. Use temperature 1.0. The prompt is in Figure 29.
- **FastText Selection**: Classify question-answer pairs with a FastText filter. Form the query strings by the following format: "Question: {question} \nAnswer: {answer_column}". We do this for both training and using the FastText classifier. The classifier's positives are S1.1, which contains responses from DeepSeek R1. The classifier's negatives are from mlfoundations-dev/stratos_verified_mix annotated with GPT-4o-mini. More info is in Appendix O.2.1.
- **Longest Answers Selection**: For any given question, take the 8 longest responses out of the 16 responses and create 8 question-answer pairs for the dataset.
- **GPT Verification**: Ask GPT-4o-mini whether a provided answer is the correct answer for the provided question. The sampling hyperparameters are temperature of 0.0, top_p of 1.0, and presence penalty of 1.0. We use structured decoding to get a response, which is a boolean value, and a reasoning, which is a string. The full prompt is in Figure 30.
- **Removing Non-English Answers**: Ask GPT-4o-mini whether a provided answer contains only English. The sampling hyperparameters are temperature of 0.0, top_p of 1.0, and presence penalty of 1.0. We use structured decoding to get a response, which is a boolean value, and a reasoning, which is a string. The full prompt is in Figure 32.
- **Removing Long Paragraphs**: Ask GPT-4o-mini whether a provided answer is free of long paragraphs. The sampling hyperparameters are temperature of 0.0, top_p of 1.0, and presence penalty of 1.0. We use structured decoding to get a response, which is a boolean value, and a reasoning, which is a string. The full prompt is in Figure 33.

### O.4.2 CODE QUESTION ANSWER FILTERING

- **FastText Selection**: Classify question-answer pairs with a FastText filter. Form the query strings by the following format: "Question: {question} \nAnswer: {answer_column}". We do this for both training and using the FastText classifier. The classifier's positives are CodeForces which contains responses from DeepSeek R1. The classifier's negatives are CodeForces annotated with GPT-4o-mini. More info is in Appendix O.2.1.

```
You will be given a science problem. Your job is to grade the difficulty level from 1-10 according to the
    international science olympiad standard. \
Here is the standard: \
A 10-point scale for international science olympiad problems could be structured as follows, where level
    1 represents the easiest problems and level 10 represents the most challenging:
Level 1: Basic Knowledge Application - Straightforward recall of fundamental scientific facts and
    principles. Simple calculations requiring only basic formulas. Direct application of a single
    scientific concept. Problems typically solvable in 1-2 steps. Content typically covered in standard
    high school curriculum. Examples include identifying simple chemical compounds, basic circuit
    calculations, or classifying organisms.
Level 2: Multi-Step Basic Applications - Problems requiring 2-3 distinct steps to solve. Application of
    multiple basic concepts within a single field. Basic data interpretation from graphs or tables.
    Simple laboratory techniques and measurements. Content typically found in advanced high school
    courses. Examples include stoichiometry calculations, basic kinematics problems, or analyzing
    simple biological processes.
Level 3: Advanced Application of Standard Concepts - Integration of multiple scientific concepts.
    Moderate quantitative reasoning with multi-step calculations. Interpretation of experimental data
    requiring analytical thinking. Problems requiring deeper understanding beyond memorization. Typical
     of challenging high school science competition questions. Examples include problems combining
    thermodynamics and kinetics, multi-step mechanics problems, or ecological relationship analysis.
Level 4: Early National Olympiad Level - Problems requiring specialized knowledge in specific scientific
    domains. Application of advanced concepts not typically covered in regular curriculum. Moderate
    laboratory techniques and experimental design understanding. Analytical thinking with non-obvious
    solution paths. Typical of early rounds in national science olympiads. Examples include chemical
    equilibrium problems with multiple variables, circuit analysis with non-ideal components, or
    molecular biology mechanisms.
Level 5: National Olympiad Standard - Problems integrating concepts across multiple scientific domains.
    Creative application of standard principles in non-standard contexts. Analysis of complex
    experimental setups and data. Multiple conceptual hurdles requiring insight. Typical of national
    olympiad final rounds. Examples include complex organic synthesis pathways, non-ideal thermodynamic
     systems, or advanced genetics problems.
Level 6: Advanced National/Early International Level - Problems requiring deep conceptual understanding
    beyond standard curriculum. Integration of theoretical knowledge with practical laboratory
    techniques. Creative problem-solving with multiple possible approaches. Application of mathematical
     models to complex scientific phenomena. Typical of international olympiad preparation camps.
    Examples include quantum mechanical models, complex biochemical pathways, or statistical analysis
    of biological systems.
Level 7: International Olympiad Standard - Problems at the level of IChO, IPhO, or IBO theoretical
    examinations. Requires specialized knowledge combined with creative insight. Complex quantitative
    modeling of scientific phenomena. Integration of concepts across scientific disciplines. Multiple
    conceptual layers requiring systematic analysis. Examples include advanced spectroscopy
    interpretation, complex physical systems with multiple forces, or detailed biochemical mechanism
    analysis.
Level 8: Advanced International Olympiad - Problems requiring both breadth and depth of scientific
    knowledge. Novel applications of scientific principles not typically taught. Sophisticated
    experimental design and analysis. Multiple solution pathways requiring evaluation and selection.
    Typical of challenging international olympiad problems. Examples include challenging quantum
    chemistry problems, advanced laboratory protocols with multiple variables, or complex evolutionary
    or ecological models.
Level 9: Elite International Olympiad - Problems requiring exceptional scientific insight and creativity.
     Integration of cutting-edge scientific knowledge. Multiple conceptual breakthroughs needed for
    solution. Problems that challenge even the most talented students. Reserved for the most difficult
    questions in international competitions. Examples include novel applications of physical principles,
     complex multi-step synthesis with stereochemical considerations, or systems biology analysis.
Level 10: Historically Challenging Problems - Problems of legendary difficulty in science competitions.
    Requires innovative approaches beyond standard methodologies. May integrate advanced university-
    level concepts. Problems that very few competitors worldwide can solve completely. Often remembered
     as particularly challenging in olympiad history. Examples include problems that required creation
    of new approaches or that stumped almost all participants in a given year.

This scale corresponds roughly to the difficulty progression you might see from school science
    competitions (levels 1-3) through national selection rounds (levels 4-5) to international olympiad
    problems (levels 6-10).

Subject-Specific Notes:
Physics (IPhO): Levels 1-3 cover standard high school physics content (mechanics, electricity,
    thermodynamics); Levels 4-6 include advanced topics like wave optics, basic quantum physics, and
    non-ideal systems; Levels 7-10 incorporate university-level content including quantum mechanics,
    statistical physics, and relativity.

Chemistry (IChO): Levels 1-3 cover basic inorganic, organic, and analytical chemistry concepts; Levels
    4-6 include complex reaction mechanisms, advanced analytical methods, physical chemistry; Levels
    7-10 incorporate sophisticated laboratory methods, quantum chemistry, and cutting-edge chemical
    concepts.

Biology (IBO): Levels 1-3 cover basic cellular, molecular, and organismal biology; Levels 4-6 include
    advanced cellular processes, genetics, evolutionary biology, and ecology; Levels 7-10 incorporate
    complex experimental design, advanced biochemistry, systems biology, and bioinformatics.
Problem to be labeled: {{question}}."
```

Figure 28: **Prompt for Science Difficulty Filtering.** This text is the prompt for Science Difficulty Filtering.

```
I will provide you the last words of 16 math problem solutions.
They are candidate solutions to a problem.

They will typically contain the solution to a math problem. I want you
    to return the indices of the responses with the most common final
    numerical answer.
Only the final numerical answer matters.

Question: What is 3 x 5?

Solution 0:
answer is 15.

Solution 1:
15.0 is the solution to this problem.

Solution 2:
The answer is 14.

You would return: [0, 1] since they are both saying 15 is the same
    answer and only one response is saying 14 is the answer.

Here is your question:
{{question}}

Here are your candidate solutions:
{{list of all solutions}}

 Now tell me the solutions. Please remember to zero index these
    solutions. Do not include the number 16 as an index.
```

Figure 29: **Prompt for Math Question-Answer Majority Consensus Verification.** This text is the prompt for Math Question-Answer Majority Consensus Verification

```
Is the provided answer a correct solution to the following problem?

Question: {{question}}
Response: {{answer}}
```

Figure 30: **Prompt for Math Question-Answer GPT Verification.** This text is the prompt for Math Question-Answer GPT Verification

- **Comprehensive Large Dataset**: Take all 63,200 responses for one dataset without filtering. This is not compute-controlled.
- **Shortest Answers Selection**: For any given question, take the 8 shortest responses out of the 16 responses and create 8 question-answer pairs for the dataset.
- **Python Tag Based Selection**: Filter out any questions that don't have python tags: "```python".
- **Majority Consensus Selection**: For any given question, provide all responses to GPT-4o-mini. Ask GPT-4o-mini to return a list of indices of responses that agree with the majority. Use temperature 1.0. The prompt is in Figure 35.
- **Longest Answers Selection**: For any given question, take the 8 longest responses out of the 16 responses and create 8 question-answer pairs for the dataset.

- **GPT Verification**: Ask GPT-4o-mini whether a provided answer is the correct answer for the provided question. The sampling hyperparameters are temperature of 0.0, top_p of 1.0, and presence penalty of 1.0. We use structured decoding to get a response, which is a boolean value, and a reasoning, which is a string. The full prompt is in Figure 31.

- **Removing Non-English Answers**: Ask GPT-4o-mini whether a provided answer contains only English. The sampling hyperparameters are temperature of 0.0, top_p of 1.0, and presence penalty of 1.0. We use structured decoding to get a response, which is a boolean value, and a reasoning, which is a string. The full prompt is in Figure 32.

- **Random Filtering**: Filtering questions-answer pairs randomly.

- **Removing Long Paragraphs**: Ask GPT-4o-mini whether a provided answer is free of long paragraphs. The sampling hyperparameters are temperature of 0.0, top_p of 1.0, and presence penalty of 1.0. We use structured decoding to get a response, which is a boolean value, and a reasoning, which is a string. The full prompt is in Figure 33.

### O.4.3 SCIENCE QUESTION ANSWER FILTERING

- **FastText Selection**: Classify question-answer pairs with a FastText filter. Form the query strings by the following format: "Question: {question} \nAnswer: {answer_column}". We do this for both training and using the FastText classifier. The classifier's positives are S1.1, which contains responses from DeepSeek R1. The classifier's negatives are from mlfoundations-dev/stratos_verified_mix annotated with GPT-4o-mini. More info is in Appendix O.2.1.

- **Comprehensive Large Dataset**: Take all 63,200 responses for one dataset without filtering. This is not compute-controlled.

- **Longest Answers Selection**: For any given question, take the 8 longest responses out of the 16 responses and create 8 question-answer pairs for the dataset.

- **Removing Non-English Answers**: Ask GPT-4o-mini whether a provided answer contains only English. The sampling hyperparameters are temperature of 0.0, top_p of 1.0, and presence penalty of 1.0. We use structured decoding to get a response, which is a boolean value, and a reasoning, which is a string. The full prompt is in Figure 32.

- **Random Filtering**: Filtering questions-answer pairs randomly.

- **Shortest Answers Selection**: For any given question, take the 8 shortest responses out of the 16 responses and create 8 question-answer pairs for the dataset.

- **Removing Long Paragraphs**: Ask GPT-4o-mini whether a provided answer is free of long paragraphs. The sampling hyperparameters are temperature of 0.0, top_p of 1.0, and presence penalty of 1.0. We use structured decoding to get a response, which is a boolean value, and a reasoning, which is a string. The full prompt is in Figure 33.

- **Majority Consensus Selection**: For any given question, provide all responses to GPT-4o-mini. Ask GPT-4o-mini to return a list of indices of responses that agree with the majority. Only use the last $1,000$ characters of each response to get the final answer. Use temperature 1.0. The prompt is in Figure 34.

- **GPT Verification**: Ask GPT-4o-mini whether a provided answer is the correct answer for the provided question. The sampling hyperparameters are temperature of 0.0, top_p of 1.0, and presence penalty of 1.0. We use structured decoding to get a response, which is a boolean value, and a reasoning, which is a string. The full prompt is in Figure 30.

```
Is the provided code snippet a correct solution to the following
    problem?

Question: {{question}}
Response: {{answer}}
```

Figure 31: **Prompt for Code Question-Answer GPT Verification.** This text is the prompt for Code Question-Answer GPT Verification

```
Does the provided answer only contain one language?

Question: {{question}}
Response: {{answer}}
```

Figure 32: **Prompt for English Verification.** This text is the prompt for English Verification.

```
Is the provided answer free of any long paragraphs?
A paragraph is any block of text separated by a blank line.
A paragraph is *too long* if it has **>750 words**.

Question: {{question}}
Response: {{answer}}
```

Figure 33: **Prompt for Long Paragraphs Verification.** This text is the prompt for Long Paragraphs Verification.

```
I will provide you the last words of 16 science problem solutions.
They are candidate solutions to a problem.

They will typically contain the solution to a science problem. I want
    you to return the indices of the responses with the most common
    answer.

Here is your question:
{{question}}

Here are your candidate solutions:
{{list of all solutions}}

Now tell me the solutions. Please remember to zero index these
    solutions. Do not include the number 16 as an index.
```

Figure 34: **Prompt for Science Question-Answer Majority Consensus Verification.** This text is the prompt for Science Majority Consensus Verification

```
I will provide you 16 code_samples.
They are candidate solutions to a coding problem.

I want you to compare all of the code samples functionally and return a list of indices corresponding to
the samples that constitute the most common solutions that are functionally equivalent. If there are sets
of solutions that are of the same size, pick one of the sets at random. I want you to also provide your
      reasoning
for the indices you respond being functionally equivalent. Here is an example:

Question: Solve fizzbuzz.

Solution 0:
def fizzbuzz1(n):
    for i in range(1, n + 1):
        output = ''

        if i % 3 == 0:
            output += 'Fizz'
        if i % 5 == 0:
            output += 'Buzz'

        print(output or i)

Solution 1:
def fizzbuzz2(n):
    for i in range(1, n + 1):
        if i % 3 == 0 and i % 5 == 0:
            print('FizzBuzz')
        elif i % 3 == 0:
            print('Fizz')
        elif i % 5 == 0:
            print('Buzz')
        else:
            print(i)

Solution 2:
def fizzbuzz3(n):
    for i in range(1, n + 1):
        # Multiple logical errors:
        if i % 3 == 0: # Notice no brackets needed for simple if statements
            print('Fizz')
        elif i % 5 == 0:
            print('Buzz')
        elif i % 3 == 0 or i % 5 == 0: # Wrong logic order and operator
            print('FizzBuzz')
        else:
            print(i)

You would return: [0, 1] since they are functionally equivalent but the third response is different.

 Here is your question:
{{question}}

Here are your candidate solutions:
{{list of all solutions}}
```

Figure 35: **Prompt for Code Question-Answer Majority Consensus Verification.** This text is the prompt for Code Question-Answer Majority Consensus Verification

| SFT Datasets | | | Benchmarks | | |
|---|---|---|---|---|---|
| Question Generation Strategy | Average | Code Avg | Math Avg | Science Avg |
| StackExchange CodeGolf | **38.8**$_{0.4}$ | 25.3$_{0.6}$ | **50.9**$_{1.1}$ | 40.7$_{0.5}$ |
| nvidia/OpenCodeReasoning | **38.4**$_{0.3}$ | **27.5**$_{0.4}$ | 47.9$_{0.7}$ | 40.7$_{0.6}$ |
| KodCode/KodCode-V1 | 37.7$_{0.3}$ | 23.9$_{0.4}$ | **49.8**$_{0.7}$ | 40.4$_{0.3}$ |
| cognitivecomputations/dolphin-coder | 37.1$_{0.5}$ | 20.8$_{0.4}$ | 49.2$_{1.6}$ | **43.3**$_{0.7}$ |
| m-a-p/CodeFeedback... | 37.0$_{0.4}$ | 18.9$_{0.3}$ | **51.8**$_{1.1}$ | 41.8$_{0.6}$ |
| Multilingual-Multimodal-NLP/McEval-... | 35.4$_{0.3}$ | 16.2$_{0.4}$ | 49.5$_{0.7}$ | **42.9**$_{0.6}$ |
| OpenCoder-LLM/opc-sft-stage2 | 35.4$_{0.4}$ | 16.7$_{0.2}$ | **51.0**$_{1.1}$ | 39.8$_{0.5}$ |
| ajibawa-2023/Code-290k-ShareGPT | 35.1$_{0.4}$ | 18.5$_{0.4}$ | 49.4$_{1.0}$ | 38.7$_{0.6}$ |
| christopher/rosetta-code | 35.0$_{0.4}$ | 14.7$_{0.3}$ | **49.6**$_{0.7}$ | **43.3**$_{1.0}$ |
| glaiveai/glaive-code-assistant-v3 | 35.0$_{0.3}$ | 16.6$_{0.3}$ | **50.2**$_{1.0}$ | 40.0$_{0.3}$ |
| prithivMLmods/Coder-Stat | 34.2$_{0.4}$ | 14.1$_{0.7}$ | **49.7**$_{0.9}$ | 41.2$_{0.7}$ |
| ise-uiuc/Magicoder-OSS-Instruct-75K | 33.9$_{0.5}$ | 17.7$_{0.4}$ | 47.4$_{1.0}$ | 37.9$_{1.2}$ |
| codeparrot/apps | 33.5$_{0.4}$ | 15.6$_{0.6}$ | **49.8**$_{0.9}$ | 35.8$_{0.8}$ |
| StackExchange Codereview | 33.4$_{0.4}$ | 13.5$_{0.5}$ | 48.4$_{0.9}$ | 40.7$_{0.9}$ |
| nampdn-ai/tiny-codes | 32.8$_{0.4}$ | 12.0$_{0.5}$ | 48.0$_{0.9}$ | 41.1$_{0.5}$ |
| bigcode/commitpackft | 32.1$_{0.5}$ | 12.2$_{0.3}$ | 46.9$_{0.9}$ | 39.6$_{1.5}$ |
| deepmind/code_contests | 31.8$_{0.4}$ | 18.5$_{0.5}$ | 45.1$_{1.1}$ | 31.8$_{0.5}$ |
| SenseLLM/ReflectionSeq-GPT | 31.5$_{0.4}$ | 14.8$_{0.4}$ | 45.6$_{0.9}$ | 35.2$_{1.0}$ |
| MatrixStudio/Codeforces-Python... | 31.4$_{0.5}$ | 19.3$_{0.4}$ | 39.3$_{1.3}$ | 37.7$_{1.1}$ |
| Magpie-Align/Magpie-Qwen2.5-... | 31.2$_{0.5}$ | 12.3$_{0.4}$ | 47.8$_{1.3}$ | 34.6$_{1.1}$ |
| bigcode/self-oss-instruct-sc2-... | 31.2$_{0.4}$ | 13.7$_{0.4}$ | 46.1$_{0.9}$ | 35.0$_{0.6}$ |
| PrimeIntellect/real-world-swe-problems | 30.5$_{0.4}$ | 10.7$_{0.2}$ | 45.7$_{1.3}$ | 37.4$_{0.7}$ |
| StackExchange | 30.3$_{0.4}$ | 8.5$_{0.4}$ | 47.5$_{1.0}$ | 37.3$_{0.3}$ |
| cfahlgren1/react-code-instructions | 28.5$_{0.4}$ | 7.8$_{0.3}$ | 45.5$_{1.3}$ | 34.0$_{0.5}$ |
| PrimeIntellect/stackexchange... | 27.6$_{0.3}$ | 5.9$_{0.2}$ | 46.8$_{1.0}$ | 31.6$_{0.5}$ |
| PrimeIntellect/synthetic... | 27.2$_{0.3}$ | 2.2$_{0.2}$ | 45.5$_{0.7}$ | 37.2$_{0.9}$ |
| bugdaryan/sql-create-... | 21.6$_{0.6}$ | 7.0$_{0.7}$ | 34.1$_{1.4}$ | 24.7$_{0.9}$ |

Table 31: **Full Ablation for Code Question Sources**

## P  PIPELINE EXPERIMENTS EXPANDED RESULTS

Due to spatial constraints, we do not show every table and every data strategy in the main text. We elaborate and show the full results here.

### P.1  QUESTION SOURCING

Our results for the code question sourcing ablation are in Table 31. Our results for the math question sourcing ablation are in Table 32. Our results for the science question sourcing ablation are in Table 33. The gap between the top-performing datasets and the lowest-performing datasets is large. In the code data domain, this difference is 17.2 points on average

| SFT Datasets | | | Benchmarks | | |
| --- | --- | --- | --- | --- | --- |
| Question Generation Strategy | Average | | Code Avg | Math Avg | Science Avg |
| ai2-adapt-dev/openmath-2-math | $\mathbf{38.1}_{0.3}$ | | $\mathbf{12.4}_{0.2}$ | $\mathbf{58.8}_{1.0}$ | $\mathbf{45.6}_{0.2}$ |
| AI-MO/NuminaMath-1.5 | $37.4_{0.5}$ | | $11.4_{0.5}$ | $\mathbf{58.5}_{1.0}$ | $\mathbf{45.0}_{1.2}$ |
| openmathinstruct_aime[*] | $37.2_{0.5}$ | | $\mathbf{12.5}_{0.3}$ | $57.1_{0.9}$ | $44.3_{1.4}$ |
| GAIR/MathPile[*] | $36.2_{0.5}$ | | $11.5_{0.7}$ | $55.1_{0.9}$ | $44.6_{1.1}$ |
| MetaMath from Numina[*] | $35.8_{0.6}$ | | $10.9_{0.8}$ | $56.3_{1.4}$ | $42.5_{1.0}$ |
| math-ai/AutoMathText[*] | $35.7_{0.4}$ | | $8.7_{0.6}$ | $55.5_{0.8}$ | $\mathbf{46.6}_{0.8}$ |
| nvidia/OpenMathInstruct-2 Aime | $35.4_{0.4}$ | | $10.7_{0.4}$ | $55.8_{1.2}$ | $41.8_{0.4}$ |
| zwhe99/DeepMath-103K | $34.8_{0.6}$ | | $7.2_{0.4}$ | $55.6_{1.9}$ | $44.8_{1.0}$ |
| ddrg/named_math_formulas[*] | $33.9_{0.4}$ | | $10.3_{0.5}$ | $53.9_{1.0}$ | $39.4_{0.4}$ |
| TIGER-Lab/MathInstruct | $33.8_{0.5}$ | | $\mathbf{12.4}_{0.6}$ | $50.6_{0.8}$ | $40.8_{1.1}$ |
| nvidia/OpenMathInstruct-2 | $33.5_{0.5}$ | | $7.9_{0.5}$ | $55.2_{1.1}$ | $39.1_{1.0}$ |
| facebook/natural_reasoning | $33.4_{0.4}$ | | $7.6_{0.3}$ | $52.1_{1.0}$ | $43.9_{0.5}$ |
| SynthLabsAI/Big-Math... | $33.0_{0.3}$ | | $9.4_{0.2}$ | $53.2_{1.0}$ | $38.1_{0.4}$ |
| TIGER-Lab/MATH-plus | $32.7_{0.4}$ | | $8.8_{0.3}$ | $51.6_{0.9}$ | $40.3_{0.9}$ |
| Asap7772/hendrycks-math-... | $32.3_{0.4}$ | | $8.9_{0.5}$ | $52.4_{1.1}$ | $37.4_{0.8}$ |
| ibivibiv/math_instruct | $30.6_{0.4}$ | | $8.2_{0.4}$ | $48.2_{1.0}$ | $37.6_{0.7}$ |
| ajibawa-2023/Maths-College | $30.1_{0.4}$ | | $2.4_{0.2}$ | $51.6_{1.1}$ | $39.5_{0.7}$ |
| BAAI/InfinityMATH[*] | $29.7_{0.3}$ | | $7.4_{0.3}$ | $47.6_{1.0}$ | $36.5_{0.2}$ |
| MetaMath[*] | $29.3_{0.4}$ | | $6.1_{0.5}$ | $48.4_{1.2}$ | $35.3_{0.5}$ |
| allenai/math_qa | $27.8_{0.4}$ | | $4.9_{0.3}$ | $46.4_{0.8}$ | $34.1_{1.1}$ |
| deepmind/math_dataset | $25.9_{0.4}$ | | $5.1_{0.2}$ | $43.2_{1.2}$ | $31.1_{0.8}$ |
| Lap1official/Math[*] | $24.4_{0.3}$ | | $7.3_{0.3}$ | $38.6_{1.0}$ | $28.5_{0.3}$ |

Table 32: **Full Ablation for Math Question Sources**

| SFT Datasets | | | Benchmarks | | |
| --- | --- | --- | --- | --- | --- |
| Question Generation Strategy | Average | | Code Avg | Math Avg | Science Avg |
| StackExchange Physics | $\mathbf{34.3}_{0.4}$ | | $\mathbf{11.9}_{0.5}$ | $\mathbf{50.9}_{0.8}$ | $43.2_{0.7}$ |
| Organic Chemistry PDF Pipeline | $\mathbf{34.0}_{0.3}$ | | $8.4_{0.3}$ | $\mathbf{52.1}_{0.7}$ | $\mathbf{45.3}_{0.8}$ |
| mteb/cqadupstack-physics | $33.3_{0.4}$ | | $7.4_{0.3}$ | $\mathbf{51.9}_{1.1}$ | $\mathbf{44.1}_{0.9}$ |
| camel-ai/physics | $30.9_{0.5}$ | | $8.6_{0.2}$ | $48.0_{1.1}$ | $38.9_{1.2}$ |
| Josephgflowers/Par-Four-Fineweb... | $30.9_{0.4}$ | | $8.2_{0.5}$ | $48.4_{1.0}$ | $38.8_{0.6}$ |
| mattany/wikipedia-biology | $29.3_{0.4}$ | | $5.2_{0.2}$ | $47.3_{1.3}$ | $38.4_{0.7}$ |
| millawell/wikipedia_field_of_science | $29.1_{0.4}$ | | $4.8_{0.4}$ | $47.7_{1.0}$ | $37.5_{0.9}$ |
| zeroshot/arxiv-biology | $28.7_{0.4}$ | | $5.5_{0.2}$ | $46.7_{0.9}$ | $36.4_{1.2}$ |
| camel-ai/chemistry | $27.8_{0.4}$ | | $3.6_{0.3}$ | $46.4_{1.1}$ | $36.1_{0.7}$ |
| Sangeetha/Kaggle... | $27.5_{0.6}$ | | $6.0_{0.6}$ | $43.8_{1.4}$ | $35.2_{1.0}$ |
| marcov/pubmed_qa... | $25.6_{0.5}$ | | $5.5_{0.2}$ | $41.8_{1.5}$ | $31.4_{0.5}$ |
| StackExchange Biology | $25.1_{0.4}$ | | $4.0_{0.3}$ | $39.6_{1.1}$ | $34.8_{0.9}$ |
| camel-ai/biology | $25.0_{0.4}$ | | $2.7_{0.2}$ | $44.1_{1.0}$ | $29.7_{1.0}$ |
| AdapterOcean/biology_dataset... | $21.9_{0.4}$ | | $3.1_{0.3}$ | $41.3_{1.1}$ | $21.1_{0.8}$ |

Table 33: **Full Ablation for Science Question Sources**

| SFT Datasets | | | Benchmarks | | |
|---|---|---|---|---|---|
| Mixing Strategy | Average | Code Avg | Math Avg | Science Avg | |
| Top 2 Code Sources | $\mathbf{41.3}_{0.4}$ | $\mathbf{27.3}_{0.3}$ | $\mathbf{54.7}_{0.9}$ | $\mathbf{42.1}_{1.0}$ | |
| Top 1 Code Sources | $39.9_{0.6}$ | $23.1_{1.0}$ | $\mathbf{54.5}_{0.8}$ | $43.1_{1.2}$ | |
| Top 4 Code Sources | $38.6_{0.4}$ | $24.2_{0.6}$ | $52.2_{0.8}$ | $39.8_{0.9}$ | |
| Top 8 Code Sources | $37.0_{0.4}$ | $21.8_{0.3}$ | $51.9_{1.2}$ | $37.7_{0.6}$ | |
| Top 16 Code Sources | $36.4_{0.4}$ | $20.8_{0.4}$ | $50.1_{0.9}$ | $39.1_{1.0}$ | |

Table 34: **Full Ablation for Code Question Source Mixing**

| SFT Datasets | | | Benchmarks | | |
|---|---|---|---|---|---|
| Mixing Strategy | Average | Code Avg | Math Avg | Science Avg | |
| Top 1 Math Sources | $\mathbf{37.6}_{0.5}$ | $\mathbf{12.1}_{0.5}$ | $\mathbf{60.1}_{1.5}$ | $\mathbf{41.9}_{0.7}$ | |
| Top 8 Math Sources | $35.8_{0.5}$ | $\mathbf{12.6}_{0.4}$ | $54.8_{0.9}$ | $42.3_{1.2}$ | |
| Top 4 Math Sources | $34.7_{0.3}$ | $9.0_{0.2}$ | $56.0_{0.9}$ | $41.4_{0.5}$ | |
| Top 2 Math Sources | $34.3_{0.3}$ | $6.9_{0.3}$ | $55.9_{0.9}$ | $\mathbf{43.0}_{0.5}$ | |
| Top 16 Math Sources | $33.8_{0.3}$ | $11.2_{0.4}$ | $50.3_{0.8}$ | $\mathbf{43.1}_{0.7}$ | |

Table 35: **Full Ablation for Math Question Source Mixing**

| SFT Datasets | | | Benchmarks | | |
|---|---|---|---|---|---|
| Mixing Strategy | Average | Code Avg | Math Avg | Science Avg | |
| Top 2 Science Sources | $\mathbf{33.7}_{0.4}$ | $9.5_{0.3}$ | $\mathbf{50.3}_{0.9}$ | $\mathbf{44.8}_{1.2}$ | |
| Top 1 Science Sources | $\mathbf{33.6}_{0.5}$ | $\mathbf{12.0}_{0.3}$ | $49.6_{0.9}$ | $42.0_{1.3}$ | |
| Top 4 Science Sources | $32.2_{0.4}$ | $8.1_{0.2}$ | $\mathbf{49.3}_{1.3}$ | $\mathbf{42.5}_{0.6}$ | |
| Top 8 Science Sources | $31.1_{0.4}$ | $6.4_{0.4}$ | $\mathbf{49.0}_{1.1}$ | $41.4_{0.9}$ | |
| Top 16 Science Sources | $30.8_{0.3}$ | $6.7_{0.2}$ | $48.4_{1.0}$ | $40.3_{0.6}$ | |

Table 36: **Full Ablation for Science Question Source Mixing**

## P.2 MIXING QUESTION GENERATION STRATEGIES

Our results for the code question mixing ablation are in Table 34. Our results for the math question mixing ablation are in Table 35. Our results for the science question mixing ablation are in Table 36. The trend of less mixing being better holds across all domains.

| SFT Datasets | | Benchmarks | | |
|---|---|---|---|---|
| Filtering Strategy | Average | Code Avg | Math Avg | Science Avg |
| Difficulty-based Selection | $\mathbf{43.0}_{0.5}$ | $27.7_{0.4}$ | $\mathbf{56.0}_{1.3}$ | $\mathbf{46.4}_{0.7}$ |
| Length-based Selection (GPT-4.1-nano) | $\mathbf{42.2}_{0.4}$ | $26.6_{0.5}$ | $\mathbf{55.4}_{1.3}$ | $\mathbf{46.0}_{0.2}$ |
| AskLLM Selection | $41.6_{0.5}$ | $\mathbf{28.8}_{0.5}$ | $52.1_{1.2}$ | $\mathbf{45.2}_{0.8}$ |
| Length-based Selection (GPT-4o-mini) | $40.8_{0.5}$ | $25.6_{0.5}$ | $53.1_{0.9}$ | $\mathbf{45.2}_{1.1}$ |
| FastText (P: Codeforces; N: CodeReview) | $40.5_{0.3}$ | $26.3_{0.4}$ | $\mathbf{53.9}_{0.9}$ | $41.8_{0.5}$ |
| Length-based Selection (GPT-4.1-mini) | $40.5_{0.3}$ | $26.8_{0.3}$ | $51.3_{0.8}$ | $44.9_{0.8}$ |
| Random Selection | $39.7_{0.5}$ | $\mathbf{28.6}_{0.5}$ | $50.2_{1.3}$ | $40.5_{0.7}$ |
| FastText (P: LeetCode; N: SQL) | $39.6_{0.4}$ | $25.2_{0.5}$ | $52.8_{1.0}$ | $41.7_{0.5}$ |
| FastText (P: Code Golf; N: SQL) | $39.0_{0.5}$ | $23.9_{0.5}$ | $52.4_{1.3}$ | $41.5_{0.7}$ |
| FastText (P: All; N: SQL) | $38.9_{0.5}$ | $26.3_{0.4}$ | $50.5_{1.0}$ | $40.7_{1.2}$ |
| FastText (P: IOI; N: SQL) | $38.5_{0.4}$ | $24.8_{0.4}$ | $50.2_{0.8}$ | $41.4_{0.7}$ |
| FastText (P: Codeforces; N: All) | $38.5_{0.4}$ | $25.1_{0.4}$ | $50.9_{1.1}$ | $39.9_{0.8}$ |
| FastText (P: Codeforces; N: SQL) | $38.4_{0.4}$ | $25.0_{0.2}$ | $50.0_{1.3}$ | $41.1_{0.6}$ |
| Embedding-based Selection | $36.9_{0.6}$ | $16.2_{0.4}$ | $\mathbf{53.4}_{1.2}$ | $43.1_{1.6}$ |

Table 37: **Full Ablation for Code Question Filtering**

| SFT Datasets | | Benchmarks | | |
|---|---|---|---|---|
| Filtering Strategy | Average | Code Avg | Math Avg | Science Avg |
| Length-based Selection (GPT-4.1-mini) | $\mathbf{41.9}_{0.3}$ | $\mathbf{13.4}_{0.3}$ | $\mathbf{66.0}_{0.8}$ | $\mathbf{48.6}_{0.4}$ |
| Length-based Selection (GPT-4.1-nano) | $39.4_{0.3}$ | $11.0_{0.4}$ | $\mathbf{64.5}_{0.7}$ | $44.3_{0.7}$ |
| AskLLM Selection | $36.3_{0.4}$ | $9.5_{0.5}$ | $58.1_{1.1}$ | $43.8_{0.6}$ |
| FastText (P: Numina; N: Lap1official) | $35.6_{0.4}$ | $11.0_{0.2}$ | $54.9_{1.1}$ | $43.5_{0.8}$ |
| Difficulty-based Selection | $35.5_{0.5}$ | $8.2_{0.3}$ | $59.3_{1.4}$ | $40.8_{1.1}$ |
| Embedding-based Selection | $35.4_{0.4}$ | $6.3_{0.3}$ | $57.0_{1.0}$ | $46.5_{0.9}$ |
| Random Selection | $35.2_{0.5}$ | $8.1_{0.2}$ | $56.6_{1.3}$ | $43.8_{1.1}$ |
| FastText (P: S1.1; N: Lap1official) | $34.9_{0.4}$ | $10.8_{0.4}$ | $55.5_{1.1}$ | $40.5_{0.3}$ |
| FastText (P: Olympiad; N: Lap1official) | $34.9_{0.3}$ | $8.7_{0.3}$ | $56.2_{1.0}$ | $42.2_{0.4}$ |
| FastText (P: OpenR1; N: Lap1official) | $34.4_{0.4}$ | $8.7_{0.3}$ | $55.1_{1.2}$ | $41.7_{0.8}$ |
| FastText (P: All; N: Lap1official) | $34.4_{0.4}$ | $9.8_{0.5}$ | $54.3_{1.0}$ | $41.4_{0.4}$ |
| FastText (P: Numina; N: All) | $32.8_{0.4}$ | $7.8_{0.2}$ | $53.9_{1.2}$ | $38.5_{0.4}$ |
| FastText (P: Numina; N: Natural Reasoning) | $32.6_{0.6}$ | $6.8_{0.4}$ | $53.9_{1.2}$ | $39.4_{1.5}$ |
| Length-based Selection (GPT-4o-mini) | $6.8_{0.3}$ | $2.3_{0.4}$ | $10.2_{0.9}$ | $8.4_{0.3}$ |

Table 38: **Full Ablation for Math Question Filtering**

### P.3 FILTERING QUESTIONS

Our results for the code question filtering ablation are in Table 37. Our results for the math question filtering ablation are in Table 38. Our results for the science question filtering ablation are in Table 39. For each data domain, we try each question filtering strategy from Appendix O.2. The ablations contain different combinations of FastText positives and negatives. They also include various models for the length-based filtering, such as GPT-4o-mini, GPT-4.1-mini, and GPT-4.1-nano. Across both science and math, length-based filtering with GPT-4.1 models works well. Around half of the filtering strategies improve over random filtering for each data domain. On all data domains, AskLLM filtering and difficulty-based filtering work relatively well.

| SFT Datasets | | | Benchmarks | | |
|---|---|---|---|---|---|
| Filtering Strategy | | Average | Code Avg | Math Avg | Science Avg |
| Length-based Selection (GPT-4.1-mini) | | $\mathbf{35.9}_{0.4}$ | $7.3_{0.5}$ | $\mathbf{54.4}_{0.9}$ | $\mathbf{50.9}_{0.9}$ |
| Length-based Selection (GPT-4.1-nano) | | $\mathbf{35.6}_{0.3}$ | $7.7_{0.3}$ | $\mathbf{53.7}_{0.7}$ | $\mathbf{50.2}_{0.8}$ |
| AskLLM Selection | | $\mathbf{35.3}_{0.3}$ | $\mathbf{11.1}_{0.1}$ | $51.5_{0.8}$ | $47.2_{0.7}$ |
| FastText (P: SciQ; N: Wikipedia w/ Arxiv) | | $\mathbf{35.1}_{0.5}$ | $9.3_{0.3}$ | $51.8_{1.3}$ | $48.7_{1.1}$ |
| Embedding-based Selection | | $34.9_{0.4}$ | $9.7_{0.4}$ | $51.2_{1.0}$ | $48.4_{1.0}$ |
| Length-based Selection (GPT-4o-mini) | | $34.9_{0.3}$ | $\mathbf{10.8}_{0.3}$ | $52.9_{0.9}$ | $44.3_{0.6}$ |
| FastText (P: SCP, SciQ, ExpertQA; N: Arxiv) | | $34.4_{0.6}$ | $9.5_{0.4}$ | $51.3_{1.1}$ | $46.4_{1.7}$ |
| Random Selection | | $33.8_{0.4}$ | $8.1_{0.4}$ | $52.1_{1.1}$ | $44.6_{0.6}$ |
| FastText (P: SciQ; N: Wikipedia w/ Arxiv) | | $33.5_{0.3}$ | $8.2_{0.3}$ | $51.3_{1.0}$ | $44.6_{0.4}$ |
| Difficulty-based Selection | | $33.5_{0.4}$ | $7.7_{0.4}$ | $50.7_{0.9}$ | $46.4_{1.1}$ |
| FastText (P: SciQ; N: Wikipedia w/ Arxiv) | | $33.4_{0.4}$ | $10.2_{0.6}$ | $50.7_{1.0}$ | $42.2_{0.5}$ |
| FastText (P: ExpertQA; N: Arxiv) | | $31.5_{0.4}$ | $\mathbf{11.4}_{0.3}$ | $45.9_{1.1}$ | $40.2_{0.8}$ |

Table 39: **Full Ablation for Science Question Filtering**

| SFT Datasets | | | Benchmarks | | |
|---|---|---|---|---|---|
| Annotation Strategy | Average | Code Avg | Math Avg | Science Avg |
| Exact Dedup w/ $4\times$ sampling | $\mathbf{41.3}_{0.5}$ | $\mathbf{28.6}_{0.4}$ | $\mathbf{53.5}_{1.2}$ | $42.1_{1.0}$ |
| No Dedup w/ $16\times$ sampling | $\mathbf{41.3}_{0.4}$ | $25.9_{0.3}$ | $\mathbf{53.8}_{1.2}$ | $\mathbf{45.8}_{0.3}$ |
| No Dedup w/ $4\times$ sampling | $\mathbf{41.2}_{0.4}$ | $27.3_{0.3}$ | $\mathbf{54.3}_{1.2}$ | $42.2_{0.7}$ |
| Fuzzy Dedup w/ $4\times$ sampling | $\mathbf{41.2}_{0.4}$ | $\mathbf{28.1}_{0.5}$ | $51.8_{1.1}$ | $45.0_{0.6}$ |
| Fuzzy Dedup w/ $16\times$ sampling | $\mathbf{41.2}_{0.4}$ | $24.8_{0.4}$ | $\mathbf{54.3}_{1.0}$ | $\mathbf{46.1}_{0.4}$ |
| Exact Dedup w/ $16\times$ sampling | $\mathbf{40.6}_{0.5}$ | $25.5_{0.4}$ | $\mathbf{53.5}_{1.2}$ | $44.1_{1.1}$ |
| No Dedup w/ $1\times$ sampling | $39.8_{0.6}$ | $26.5_{0.4}$ | $\mathbf{53.0}_{1.2}$ | $39.9_{1.5}$ |

Table 40: **Full Ablation for Code Deduplication and Multiple Sampling**

| SFT Datasets | | | Benchmarks | | |
|---|---|---|---|---|---|
| Annotation Strategy | Average | Code Avg | Math Avg | Science Avg |
| Exact Dedup w/ $1\times$ sampling | $\mathbf{41.7}_{0.3}$ | $\mathbf{14.0}_{0.3}$ | $\mathbf{66.0}_{1.0}$ | $46.6_{0.2}$ |
| Exact Dedup w/ $16\times$ sampling | $40.1_{0.4}$ | $11.0_{0.3}$ | $63.6_{1.3}$ | $\mathbf{48.7}_{0.5}$ |
| Exact Dedup w/ $4\times$ sampling | $39.2_{0.4}$ | $12.0_{0.3}$ | $62.6_{1.1}$ | $44.7_{0.5}$ |
| Fuzzy Dedup w/ $4\times$ sampling | $38.8_{1.2}$ | $12.4_{0.5}$ | $60.6_{3.9}$ | $45.7_{0.9}$ |
| No Dedup w/ $1\times$ sampling | $38.3_{1.1}$ | $10.8_{0.3}$ | $59.8_{3.7}$ | $47.4_{1.0}$ |
| No Dedup w/ $4\times$ sampling | $37.7_{0.4}$ | $3.3_{0.3}$ | $\mathbf{64.6}_{1.2}$ | $\mathbf{49.0}_{0.6}$ |
| Fuzzy Dedup w/ $1\times$ sampling | $37.4_{1.5}$ | $9.3_{1.7}$ | $60.6_{4.1}$ | $44.8_{2.0}$ |
| No Dedup w/ $16\times$ sampling | $36.5_{1.2}$ | $\mathbf{13.9}_{0.4}$ | $55.4_{3.9}$ | $42.1_{1.1}$ |
| Fuzzy Dedup w/ $16\times$ sampling | $36.0_{0.4}$ | $5.5_{0.2}$ | $61.1_{1.1}$ | $43.9_{0.9}$ |

Table 41: **Full Ablation for Math Deduplication and Multiple Sampling**

| SFT Datasets | | | Benchmarks | | |
|---|---|---|---|---|---|
| Annotation Strategy | Average | Code Avg | Math Avg | Science Avg |
| Exact Dedup w/ $16\times$ sampling | $\mathbf{36.2}_{0.5}$ | $9.0_{0.4}$ | $\mathbf{54.5}_{1.0}$ | $49.7_{1.2}$ |
| Fuzzy Dedup w/ $16\times$ sampling | $\mathbf{36.1}_{0.4}$ | $\mathbf{10.9}_{0.2}$ | $52.9_{1.3}$ | $48.8_{0.5}$ |
| Exact Dedup w/ $4\times$ sampling | $\mathbf{35.8}_{0.5}$ | $10.6_{0.7}$ | $51.8_{1.0}$ | $49.6_{1.2}$ |
| No Dedup w/ $4\times$ sampling | $\mathbf{35.8}_{0.4}$ | $10.0_{0.4}$ | $\mathbf{55.2}_{0.8}$ | $45.4_{0.9}$ |
| No Dedup w/ $16\times$ sampling | $\mathbf{35.7}_{0.4}$ | $7.6_{0.5}$ | $53.8_{1.0}$ | $\mathbf{50.9}_{0.5}$ |
| No Dedup w/ $1\times$ sampling | $\mathbf{35.5}_{0.3}$ | $9.3_{0.3}$ | $54.2_{1.1}$ | $46.9_{0.2}$ |
| Exact Dedup w/ $1\times$ sampling | $35.0_{0.4}$ | $7.6_{0.4}$ | $54.0_{1.2}$ | $47.5_{0.5}$ |
| Fuzzy Dedup w/ $4\times$ sampling | $34.9_{0.4}$ | $7.4_{0.5}$ | $55.0_{1.0}$ | $46.0_{0.7}$ |
| Fuzzy Dedup w/ $1\times$ sampling | $34.2_{0.3}$ | $5.8_{0.4}$ | $52.5_{0.7}$ | $49.5_{0.4}$ |

Table 42: **Full Ablation for Science Deduplication and Multiple Sampling**

## P.4 DEDUPLICATION AND MULTIPLE SAMPLING

Our results for the code deduplication and multiple sampling ablation are in Table 40. Our results for the math deduplication and multiple sampling ablation are in Table 41. Our results for the science deduplication and multiple sampling ablation are in Table 42. Across all data domains, doing exact deduplication or no deduplication was better than doing fuzzy deduplication. Moreover, doing multiple sampling performed as well as or equal to annotating each question one time. The main exception here is in Table 41, where doing 1 annotation per question is better. However, the second-best strategy empirically is doing exact deduplication with $16\times$ sampling. The $16\times$ provides an axis for scaling data more for OpenThoughts3-1.2M, so we chose this annotation strategy for our final pipeline.

| SFT Datasets | | Benchmarks | | |
| --- | --- | --- | --- | --- |
| Filtering Strategy | Average | Code Avg | Math Avg | Science Avg |
| FastText Selection | $\mathbf{42.3}_{0.5}$ | $\mathbf{27.2}_{0.5}$ | $\mathbf{54.7}_{1.4}$ | $\mathbf{46.5}_{0.3}$ |
| No Filtering | $\mathbf{42.2}_{0.5}$ | $27.4_{0.6}$ | $54.5_{1.1}$ | $46.1_{0.9}$ |
| Shortest Answers Selection | $\mathbf{42.0}_{0.5}$ | $26.5_{0.5}$ | $54.4_{1.1}$ | $\mathbf{46.9}_{1.1}$ |
| Python Tag Based Selection | $\mathbf{41.7}_{0.5}$ | $\mathbf{27.8}_{0.3}$ | $54.6_{1.0}$ | $43.4_{1.2}$ |
| Majority Consensus Selection | $\mathbf{41.3}_{0.5}$ | $26.6_{0.2}$ | $53.6_{1.2}$ | $\mathbf{45.0}_{1.0}$ |
| Longest Answers Selection | $41.0_{0.4}$ | $26.9_{0.4}$ | $\mathbf{55.4}_{1.2}$ | $40.5_{0.7}$ |
| GPT Verification | $40.7_{0.4}$ | $25.5_{0.4}$ | $53.6_{0.8}$ | $44.2_{1.0}$ |
| Removing Non-English Answers | $40.6_{0.4}$ | $26.2_{0.5}$ | $54.5_{1.1}$ | $41.6_{0.6}$ |
| Random Filtering | $39.8_{0.4}$ | $25.1_{0.5}$ | $52.2_{1.1}$ | $43.0_{0.5}$ |
| Removing Long Paragraphs | $30.3_{0.5}$ | $19.6_{0.2}$ | $40.5_{1.5}$ | $31.3_{0.9}$ |

Table 43: **Full Ablation for Code Question-Answer Filtering**

| SFT Datasets | | Benchmarks | | |
| --- | --- | --- | --- | --- |
| Filtering Strategy | Average | Code Avg | Math Avg | Science Avg |
| No Filtering | $\mathbf{41.9}_{0.4}$ | $\mathbf{15.2}_{0.5}$ | $\mathbf{65.6}_{0.9}$ | $46.4_{0.7}$ |
| Random Filtering | $\mathbf{41.6}_{0.4}$ | $14.9_{0.4}$ | $\mathbf{64.8}_{0.9}$ | $46.7_{0.5}$ |
| Shortest Answers Selection | $\mathbf{41.1}_{0.4}$ | $14.8_{0.4}$ | $63.7_{1.1}$ | $46.7_{0.7}$ |
| Removing Non-English Answers | $\mathbf{41.1}_{0.5}$ | $14.2_{0.5}$ | $63.1_{1.0}$ | $\mathbf{48.6}_{1.0}$ |
| Majority Consensus Selection | $41.0_{0.4}$ | $\mathbf{14.5}_{0.5}$ | $62.3_{0.8}$ | $\mathbf{48.8}_{0.8}$ |
| FastText Selection | $40.7_{0.5}$ | $13.5_{0.2}$ | $62.8_{1.4}$ | $\mathbf{48.4}_{0.8}$ |
| Longest Answers Selection | $40.5_{0.5}$ | $12.9_{0.4}$ | $\mathbf{63.9}_{1.4}$ | $46.7_{1.0}$ |
| GPT Verification | $40.0_{0.5}$ | $13.1_{0.3}$ | $61.4_{1.1}$ | $\mathbf{48.3}_{1.1}$ |
| Removing Long Paragraphs | $38.0_{0.4}$ | $5.7_{0.2}$ | $\mathbf{64.5}_{0.9}$ | $46.8_{1.0}$ |

Table 44: **Full Ablation for Math Question-Answer Filtering**

| SFT Datasets | | Benchmarks | | |
| --- | --- | --- | --- | --- |
| Filtering Strategy | Average | Code Avg | Math Avg | Science Avg |
| No Filtering | $\mathbf{38.3}_{0.4}$ | $10.6_{0.3}$ | $\mathbf{56.9}_{0.9}$ | $\mathbf{51.9}_{0.7}$ |
| Longest Answers Selection | $\mathbf{37.5}_{0.3}$ | $12.0_{0.2}$ | $\mathbf{55.4}_{0.8}$ | $48.8_{0.7}$ |
| Removing Non-English Answers | $37.4_{0.4}$ | $\mathbf{13.4}_{0.4}$ | $54.5_{1.0}$ | $47.6_{0.9}$ |
| FastText Selection | $36.8_{0.4}$ | $11.5_{0.4}$ | $54.7_{1.1}$ | $47.7_{0.5}$ |
| Random Filtering | $36.5_{0.4}$ | $11.0_{0.4}$ | $53.9_{1.1}$ | $48.8_{0.7}$ |
| Shortest Answers Selection | $36.2_{0.5}$ | $11.4_{0.7}$ | $53.6_{1.2}$ | $47.5_{0.4}$ |
| Removing Long Paragraphs | $35.9_{0.5}$ | $11.4_{0.6}$ | $53.7_{1.0}$ | $45.7_{1.2}$ |
| Majority Consensus Selection | $35.7_{0.5}$ | $9.1_{0.5}$ | $54.6_{1.1}$ | $47.1_{0.9}$ |
| GPT Verification | $35.6_{0.5}$ | $11.4_{0.5}$ | $52.6_{1.3}$ | $46.4_{1.0}$ |

Table 45: **Full Ablation for Science Question-Answer Filtering**

## P.5 QUESTION ANSWER FILTERING

Our results for the code question-answer filtering ablation are in Table 43. Our results for the math question-answer filtering ablation are in Table 44. Our results for the science question-answer filtering ablation are in Table 45. Across all data domains, not doing filtering at all performs similarly or outperforms the best question-answer filtering strategy.

| SFT Datasets | | | Benchmarks | | |
|---|---|---|---|---|---|
| Teacher Models | Average | Code Avg | Math Avg | Science Avg |
| Qwen/QwQ-32B | $\mathbf{44.2}_{0.5}$ | $\mathbf{29.5}_{0.3}$ | $\mathbf{58.7}_{1.1}$ | $\mathbf{44.6}_{1.0}$ |
| deepseek-ai/DeepSeek-R1 | $38.0_{1.5}$ | $19.2_{3.4}$ | $54.3_{1.6}$ | $41.8_{1.0}$ |
| microsoft/Phi-4-reasoning-plus | $29.0_{0.4}$ | $0.5_{0.1}$ | $52.1_{1.2}$ | $37.2_{0.6}$ |

Table 46: **Full Ablation for Teacher Model for Code**

| SFT Datasets | | | Benchmarks | | |
|---|---|---|---|---|---|
| Teacher Models | Average | Code Avg | Math Avg | Science Avg |
| Qwen/QwQ-32B | $\mathbf{44.2}_{0.4}$ | $10.9_{0.4}$ | $\mathbf{71.6}_{1.1}$ | $\mathbf{53.2}_{0.4}$ |
| deepseek-ai/DeepSeek-R1 | $40.6_{0.3}$ | $\mathbf{13.3}_{0.4}$ | $62.5_{0.9}$ | $48.5_{0.4}$ |
| microsoft/Phi-4-reasoning-plus | $30.6_{0.6}$ | $7.1_{0.4}$ | $49.0_{1.6}$ | $38.2_{0.9}$ |

Table 47: **Full Ablation for Teacher Model for Math**

| SFT Datasets | | | Benchmarks | | |
|---|---|---|---|---|---|
| Teacher Models | Average | Code Avg | Math Avg | Science Avg |
| Qwen/QwQ-32B | $\mathbf{39.1}_{0.4}$ | $\mathbf{10.1}_{0.5}$ | $\mathbf{62.1}_{1.2}$ | $48.0_{0.2}$ |
| deepseek-ai/DeepSeek-R1 | $35.9_{0.7}$ | $7.1_{1.6}$ | $55.9_{0.9}$ | $\mathbf{49.0}_{0.4}$ |
| microsoft/Phi-4-reasoning-plus | $21.7_{0.4}$ | $4.8_{0.2}$ | $28.8_{1.3}$ | $36.4_{0.7}$ |

Table 48: **Full Ablation for Teacher Model for Science**

## P.6    TEACHER MODEL EXPERIMENTS

Our results for the teacher model ablations for code are in Table 46. Our results for the teacher model ablations for math are in Table 47. Our results for the teacher model ablations for science are in Table 48. Across all data domains, QwQ-32B is the best teacher by a statistically significant margin.

