# OpenReview forum: "OpenThoughts: Data Recipes for Reasoning Models"
_ICLR.cc/2026/Conference — ICLR 2026 Oral_

### Official Review · Reviewer_1sri · 2025-10-28

**Soundness:** 3
**Presentation:** 4
**Contribution:** 3
**Rating:** 6
**Confidence:** 3

**Summary:**

The paper addresses the challenge of creating publicly available data recipes for training reasoning models. It introduces a data generation pipeline for creating open-source reasoning data by empirically selecting the most effective approach at each stage. Using this pipeline, the authors construct the OpenThoughts2-1M and OpenThoughts3 datasets. A model trained on this data, named OpenThinker3-7B, achieves state-of-the-art performance.

**Strengths:**

S1: The paper conducts thorough experiments and constructs open-source datasets, which paves the way for future research on reasoning models.

S2: The paper is well-structured and easy to follow.

S3: The paper addresses a practical data problem for reasoning model training and research, which is an important contribution to the development of AI.

**Weaknesses:**

W1: The paper's aim to create open-source datasets is undermined by the proposed pipeline's high dependency on closed-source LLM APIs, such as GPT-4o. This creates a contradiction and harms the reproducibility of the work, as results can vary significantly depending on the specific API version used.

W2: Some experimental conclusions lack rigor. For instance, in Section 3.6, the paper dismisses answer filtering strategies as ineffective because they did not outperform the baseline. However, this conclusion is not well-supported because the comparison is unfair: the baseline was trained on 63,200 samples, while the answer filtering strategies used only 31,600. This confounding variable makes it impossible to rule out that the baseline's superior performance stems from having more training data, rather than the ineffectiveness of filtering.

W3: The paper lacks an in-depth analysis of some interesting and counterintuitive experimental results, failing to provide deeper insights. For example, Section 3.7 shows that while Qwen-32B has a lower average score than DeepSeek-R1, it outperforms all other models when used as a teacher. This phenomenon is counterintuitive, and a more thorough analysis of why a lower-scoring model makes a better teacher could offer valuable insights for future work.

W4: While it is understandable that some experimental results are placed in the appendix due to space limitations, the paper frequently has a significant separation between the presentation of results and their corresponding analysis. This disjointed structure makes the paper difficult to read.

**Questions:**

See above.

---

> ### Author Response · Authors · 2025-11-21
>
> We thank the reviewer for their thoughtful review!
>
> > The paper's aim to create open-source datasets is undermined by the proposed pipeline's high dependency on closed-source LLM APIs, such as GPT-4o. This creates a contradiction and harms the reproducibility of the work, as results can vary significantly depending on the specific API version used.
>
> Throughout this work, we used DeepSeek-R1, QwQ-32B, and Phi-4 for generating data. We mainly use LLM APIs such as GPT-4o for filtering. However, filtering could be replaced with different models, and our pipeline is not fundamentally dependent on these closed-source models. We do consistently use gpt-4o-mini-2024-07-18 for data generation.
>
>
> > Some experimental conclusions lack rigor. For instance, in Section 3.6, the paper dismisses answer filtering strategies as ineffective because they did not outperform the baseline. However, this conclusion is not well-supported because the comparison is unfair: the baseline was trained on 63,200 samples, while the answer filtering strategies used only 31,600. This confounding variable makes it impossible to rule out that the baseline's superior performance stems from having more training data, rather than the ineffectiveness of filtering.
>
> Please see Tables 43, 44, and 45 for full results. We include both a no-filtering and a random-filtering baseline, with random filtering being compute-controlled. Our conclusion is that answer filters do not improve results. For example, in the math domain, random filtering performs best up to the standard error. While it may be desirable for the training compute-controlled setting to use a smaller dataset, all of these methods are controlled for data-generation compute. That is, we collect the same number of teacher annotations for each method, which is the main computational expense in our experiments. Moreover, our goal is to achieve the best possible performance, and the results of this experiment demonstrate that doing no answer-based filtering yielded the best downstream results.
>
>
> > The paper lacks an in-depth analysis of some interesting and counterintuitive experimental results, failing to provide deeper insights. For example, Section 3.7 shows that while Qwen-32B has a lower average score than DeepSeek-R1, it outperforms all other models when used as a teacher. This phenomenon is counterintuitive, and a more thorough analysis of why a lower-scoring model makes a better teacher could offer valuable insights for future work.
>
> The primary goal of the paper is to achieve high performance, and understanding some of the phenomena in more depth is a good direction for future work. We agree that analysis here would be helpful, but we can only suggest possible hypotheses. See Section E.2 for our study of what makes a good teacher model. We analyzed different teacher models across smaller datasets to assess their impact on downstream performance. We found, for example, that Claude’s performance increases linearly with the number of tokens, suggesting that more verbose teachers are better. We tried filtering the teacher annotations in section E.3 and found that compressing or removing “wait” tokens significantly hurt performance. This supports that verbose teachers are better.
>
>
> > While it is understandable that some experimental results are placed in the appendix due to space limitations, the paper frequently has a significant separation between the presentation of results and their corresponding analysis. This disjointed structure makes the paper difficult to read.
>
> Given the large number of experiments, fitting all our results into nine pages is infeasible. With an extra page for the camera-ready, we will be able to improve our structure greatly. We would greatly appreciate any concrete examples, so we can fix them!

---

> > ### Comment · Reviewer_1sri · 2025-11-26
> >
> > Thank you for your comprehensive response. Most of my concerns have been addressed. After reading the rebuttal and the review comments from the other reviewers, I have decided to maintain my positive score.

---

### Official Review · Reviewer_KbX7 · 2025-10-28

**Soundness:** 3
**Presentation:** 3
**Contribution:** 3
**Rating:** 6
**Confidence:** 3

**Summary:**

This paper presents a large-scale empirical study aimed at identifying effective data recipes for improving LLM reasoning performance. The authors systematically analyze the impact of various data curation strategies, including question sourcing, filtering, deduplication, teacher selection, and dataset scaling. Many controlled experiments are conducted to evaluate how different design choices affect model reasoning ability under supervised fine-tuning. The paper also introduces an open dataset and provides detailed ablations to ensure reproducibility and transparency.

**Strengths:**

- The paper constructs a large and well-curated dataset specifically designed to enhance LLM reasoning capabilities. This contribution is practically meaningful, as data quality and composition have become increasingly crucial for reasoning-oriented LLM development. The released dataset and accompanying analysis provide a useful foundation for future research in reasoning and instruction tuning.

- The empirical study is extensive and carefully executed. The authors evaluate a wide range of data curation factors, such as filtering, question mixing, deduplication, teacher selection, and dataset scaling. The conclusions are supported with detailed ablation studies. The experiments are systematic and transparent, offering clear evidence for each design choice and contributing valuable insights for the broader community.

**Weaknesses:**

- This work feels closer to a technical report than a research paper. The study is solid and well-executed, but it lacks significant technical novelty. The proposed pipeline is straightforward and largely follows existing practices in data curation and reasoning dataset construction, with limited conceptual innovation.

- The proposed dataset provides only marginal improvements over existing baselines. As shown in Table 1, the average performance gain over previous datasets is relatively small (around 2.1), which raises questions about the practical significance of the proposed data recipes. While the results are consistent and reproducible, the improvement is quite modest and may not justify the large experimental effort.

**Questions:**

I do not have specific questions for the authors.

---

> ### Author Response · Authors · 2025-11-21
>
> We thank the reviewer for recognizing the practical value of our work for developing reasoning LLMs. We also appreciate the acknowledgement that our extensive study is a strong foundation for future research.
>
> > This work feels closer to a technical report than a research paper. The study is solid and well-executed, but it lacks significant technical novelty. The proposed pipeline is straightforward and largely follows existing practices in data curation and reasoning dataset construction, with limited conceptual innovation.
>
> Regarding the first critique of technical novelty, we emphasize our systematic ablations include several experiments which explore dataset construction practices not already existing in the literature. We include these new methods alongside existing practices for comparison (e.g. deduplication with multiple sampling in Table 5, all starred question sourcing methods in Table 2, answer filtering via answer length, etc.). While the new methods did not always outperform other approaches, some stages of the pipeline resulted in novel observations (e.g. using a weaker teacher model resulted in stronger downstream reasoning performance).
>
> Ours is the first work to define a reasoning dataset construction pipeline and rigorously experiment on each element of it. We feel this formalization is in itself a contribution.
>
> Furthermore, the previous popular work in reasoning dataset curation (S1, LIMO) argued for smaller datasets whereas we demonstrate that scale is an important factor in improving downstream performance.
>
> > The proposed dataset provides only marginal improvements over existing baselines. As shown in Table 1, the average performance gain over previous datasets is relatively small (around 2.1), which raises questions about the practical significance of the proposed data recipes. While the results are consistent and reproducible, the improvement is quite modest and may not justify the large experimental effort.
>
> Considering the second critique on magnitude of the improvement over baselines, we emphasize the 30 point increase over the base model. Our initial goal was to recreate the DeepSeek-R1-Qwen-7B model with open data and in the end, we surpassed this model by 12 points. OpenThinker3-7B also has a 8 point improvement over the strongest SFT-only baseline (NemoNano-1M) in a controlled experimental setting using the exact same hyperparameters.
>
> When we expand our comparison beyond datasets to include additional training methods (namely RL on top of SFT) our model still performs well, showing a 2.1 improvement over the strongest compared RL+SFT baseline (Nemotron-Nano-8B). Further RL on top of our SFT-only model presents an opportunity to improve performance even more, and we leave this opportunity for future work.
>
> Overall, our large experimental effort resulted in a fully open dataset with full details on construction and experimental justification behind each step in the construction pipeline. Not only was our model competitive with the most performant models, it is also, by far, the most open model. We also define a scalable data recipe, which can be used to create even larger datasets to achieve higher performance.

---

> > ### Comment · Reviewer_KbX7 · 2025-11-26
> >
> > Thanks for the clarifications provided in the rebuttal. I appreciate the authors’ efforts in addressing the concerns raised. I have no further comments at this stage.

---

### Official Review · Reviewer_1PHv · 2025-10-30

**Soundness:** 3
**Presentation:** 3
**Contribution:** 3
**Rating:** 6
**Confidence:** 4

**Summary:**

This paper presents OpenThoughts3, a scalable and systematic data curation pipeline for building reasoning models through supervised fine-tuning. The authors carefully analyze each step of the data generation process—question sourcing, filtering, deduplication, answer sampling, and teacher selection—through more than 1,000 controlled experiments, resulting in a high-quality open dataset of 1.2M examples. The resulting model, OpenThinker3-7B, achieves state-of-the-art results among open-data models, surpassing comparable baselines like DeepSeek-R1-Distill-7B and Nemotron-Nano-8B across math, code, and science benchmarks. The paper emphasizes that careful dataset design can rival or exceed RL-based approaches, and provides a reproducible open-source recipe for training reasoning models.

**Strengths:**

1.	The work excels in its thorough dissection of the data curation process, covering multiple strategies at every pipeline stage. This level of experimental rigor goes beyond most existing work, making the conclusions highly credible and reproducible.
2.	By scaling the dataset to 1.2M examples and carefully selecting teacher models, the authors deliver competitive results on multiple reasoning benchmarks. The scaling curves and ablation studies clearly demonstrate the effectiveness of their approach.
3.	Unlike many proprietary reasoning models, the authors commit to full open-source release of datasets and models. This can significantly accelerate community research and lower the entry barrier for reasoning model development.

**Weaknesses:**

1.	I appreciate the design of the SFT data pipeline, but it’s hard not to notice how the entire work leans almost exclusively on this single training paradigm. In recent reasoning models, RL or curriculum strategies often play a big role in pushing performance further. Even if the authors didn’t run those experiments, a more thoughtful discussion or positioning would have made the contribution feel less one-dimensional.
2.	I’m left unsure whether the gains translate to broader reasoning capabilities. A few results on less standard or more language-heavy tasks would help a lot here. Right now, the claims around generalization feel more suggestive than demonstrated.
3.	The paper runs a huge number of experiments, but says very little about why certain design choices actually help. For instance, why does a relatively simple teacher mix outperform a seemingly stronger one? What kinds of examples drive the improvements? Without some interpretability or qualitative perspective, the work risks feeling like an “empirical recipe” rather than a deeper contribution.

**Questions:**

Please refer to Weaknesses.

---

> ### Author Response · Authors · 2025-11-21
>
> We appreciate the reviewer's acknowledgement of our experimental rigour and the comprehensive study at every stage in the dataset construction pipeline. Thank you for acknowledging that our findings are strong and effective at improving model performance.
>
> > I appreciate the design of the SFT data pipeline, but it’s hard not to notice how the entire work leans almost exclusively on this single training paradigm. In recent reasoning models, RL or curriculum strategies often play a big role in pushing performance further. Even if the authors didn’t run those experiments, a more thoughtful discussion or positioning would have made the contribution feel less one-dimensional.
>
> We designed our work around a single training paradigm. By holding the training paradigm constant, we are able to run extensive, controlled experiments to determine which variables in the dataset pipeline have the largest impact on improving downstream performance. This approach enables us to run this rigorous study on many dimensions of reasoning dataset construction.
>
> The fact that our SFT-only trained model is competitive to RL+SFT models (see Table 1) is a strength and demonstrates that this simple training paradigm with a strong dataset pipeline can go a long way in improving model performance.
>
> Even in more complex training paradigms, which can improve model performance, the first step is usually SFT training. For those who are interested in developing more complex training paradigms, our work is an effective foundation and our models can be subsequently trained (e.g. adding an RL step or an additional SFT step on a more challenging curriculum). Studying reasoning recipes for RL is a promising area of future work, but did not fit into the scope of this paper.
>
>
> > I’m left unsure whether the gains translate to broader reasoning capabilities. A few results on less standard or more language-heavy tasks would help a lot here. Right now, the claims around generalization feel more suggestive than demonstrated.
>
> When comparing our final model to the baselines, we evaluated performance on a held out set of reasoning benchmarks. This set was not measured or used as a signal during our large scale ablations on the dataset pipeline. We demonstrate generalization in Table 1 showing improvements not only in the core development reasoning benchmarks, but also in the held out set.
>
>
> Our 8 core datasets and 4 held out datasets over the 3 domains of math, code, and science meaningfully measure a diverse set of reasoning ability. Future work on new reasoning benchmarks is an exciting direction.
>
> > The paper runs a huge number of experiments, but says very little about why certain design choices actually help. For instance, why does a relatively simple teacher mix outperform a seemingly stronger one? What kinds of examples drive the improvements? Without some interpretability or qualitative perspective, the work risks feeling like an “empirical recipe” rather than a deeper contribution.
>
> Ours is the first work to define a reasoning dataset construction pipeline and rigorously experiment on each element of it. We feel this formalization is in itself a contribution.
>
> We agree that analysis here would be helpful, but we can only suggest possible hypotheses. See Section E.2 for our study of what makes a good teacher model. We analyzed different teacher models across smaller datasets to assess their impact on downstream performance. We found, for example, that Claude’s performance increases linearly with the number of tokens, suggesting that more verbose teachers are better. We tried filtering the teacher annotations in section E.3 and found that compressing or removing “wait” tokens significantly hurt performance. This supports verbose teachers being better.
>
> Additional analysis and rigorous study to provide causal explanation for some of the observed results is an interesting future research direction.

---

### Official Review · Reviewer_wnkN · 2025-11-01

**Soundness:** 4
**Presentation:** 3
**Contribution:** 3
**Rating:** 8
**Confidence:** 4

**Summary:**

This paper introduces OpenThoughts2-1M and OpenThinker2-32B, the first publicly available open source dataset and model for reasoning tasks that match the performance of DeepSeek-R1-Distill-32B on AIME and LiveCodeBench. They also perform a series of 1000+ experiments to systematically improve their data generation pipeline to develop OpenThoughts3, which, combined with scaling to 1.2M examples and using QwQ-3B, yields OpenThinker3-7B, which vastly outperforms DeepSeek-R1-Distill-Qwen-7B on AIME, LiveCodeBench, and GPQA. Key findings include that sampling multiple answers from a teacher model is an effective strategy to scale the size of the training data, models with better performance are not necessarily better teachers, verification and answer filtering methods don’t lead to significant performance improvements, data quality trumps data diversity, and filtering questions based on LLM labeled difficulty or response length yields better results than using embedding based filters typical to pre-training.

**Strengths:**

1. Well-motivated and carefully designed experiments leading to clear takeaways about best practices for SFT training data design for reasoning models.
2. OpenThinker3-7B achieves the best average performance across all the evaluation benchmarks.
3. Efforts to decontaminate training data by removing samples with high similarity to the benchmark instances.
4. Interesting results: Question sourcing (simple synthetic questions perform comparably or even better than complex or manually curated pipelines), Question filtering (difficulty filtering and response length filtering work well for code and math, compared to fasttext classifiers or embedding-based methods that work well for pre-training), Best teacher model is not necessarily the best performing model.

**Weaknesses:**

1. Should further investigate the impact of deduplication and sampling multiple answers because the results are very inconclusive, and exhibit too much variance across code, math, and science. It is also weird that the authors choose varying deduplication strategies across math, code, and science, but choose to pick x16 answer sampling per question for all domains, even though it is not the best across the board. This choice seems arbitrary, and the reasoning that it is better for scaling seems weird since training on much more data for essentially the same or worse performance is potentially just a waste of compute.
2. Again, the conclusions about answer filtering seem weird, since they bring up the point that training on all the data instead of filtering low-quality data makes no difference, so training on all the data is better for scaling. However, an alternative perspective would be that training on these low-quality instances doesn’t add to the performance, so why should one waste compute on training with these instances? In other words, answer filtering strategies could be used to find a smaller, more effective dataset that retains the same level of performance as the full dataset.
3. The difficulty filters used for answer filtering do not use any verifiers, like code execution with test suites, etc., which, while hard to scale, have shown a lot of promise in rejection fine-tuning-based pipelines (like RAFT) [1, 2, 3].
4. There is seemingly a big gap in performance between the best-performing OpenThoughts3-7B model and the teacher models used (QwQ, DeepSeekReasoner). While this is reasonable given the parameter size difference, the 10-point difference is pretty significant.
5. Some of the more intriguing results warrant further exploration, like why a less capable teacher model is a better teacher or why different question filtering strategies work for specific domains.

[1] Zheng, Kunhao, et al. "What Makes Large Language Models Reason in (Multi-Turn) Code Generation?." arXiv preprint arXiv:2410.08105 (2024).
[2] Xiong, Wei, et al. "A minimalist approach to llm reasoning: from rejection sampling to reinforce." arXiv preprint arXiv:2504.11343 (2025).
[3] Dong, Hanze, et al. "Raft: Reward ranked finetuning for generative foundation model alignment." arXiv preprint arXiv:2304.06767 (2023).

**Questions:**

Do you explore potential explanations of the QwQ being the best teacher model as an observation of the capacity gap phenomenon [1]?

[1] Zhang, Chen, et al. "Towards the law of capacity gap in distilling language models." arXiv preprint arXiv:2311.07052 (2023).

---

> ### Author Response · Authors · 2025-11-21
>
> We thank the reviewer for their thoughtful review!
>
> > Should further investigate the impact of deduplication and sampling multiple answers because the results are very inconclusive, and exhibit too much variance across code, math, and science.
>
> Since these are very different domains and data sources, variance across domains is reasonable. The central claim here is that doing 16x fewer unique questions and answering each question 16x times achieves the same or better performance than answering 16x questions once. This is an unintuitive yet powerful result for data curation, as it gives us a meaningful axis to scale our data.
>
>
> > It is also weird that the authors choose varying deduplication strategies across math, code, and science, but choose to pick x16 answer sampling per question for all domains, even though it is not the best across the board.
>
>
> Each method is compute-controlled here, with 31,600 datapoints for each. We are only varying the number of unique instructions for each. Thus, doing 16x sampling means this dataset has 16x fewer unique questions compared to 1x sampling. Please see tables 40, 41, and 42 for the main results. 16x sampling is the best up to standard noise for code and science. It is only second in math, but 16x sampling allows us to stretch the amount of source questions we have by reusing them more times. This is a powerful method to scale a dataset. Thus, we consider picking 16x scaling in this one case is a reasonable exception.
>
> > However, an alternative perspective would be that training on these low-quality instances doesn’t add to the performance, so why should one waste compute on training with these instances? In other words, answer filtering strategies could be used to find a smaller, more effective dataset that retains the same level of performance as the full dataset.
>
> Please see Tables 43, 44, and 45 for full results. We include both a no-filtering and random filtering baseline, where random filtering is compute-controlled. Our conclusion is that answer filters do not improve results. For example, for math domain data, random filtering performs the best up to the standard error. While it may be desirable for the training compute-controlled setting to use a smaller dataset, all of these methods are controlled with respect to data-generation compute. That is, we collect the same number of teacher annotations for each method, which is a considerable computational expense in our experiments. Moreover, our goal is to achieve the best performance possible, and the results from this experiment demonstrate that doing no answer-based filtering was optimal for providing the best downstream results.
>
> > The difficulty filters used for answer filtering do not use any verifiers, like code execution with test suites, etc., which, while hard to scale, have shown a lot of promise in rejection fine-tuning-based pipelines (like RAFT) [1, 2, 3].
>
> In Table 17, we use code execution with unit tests to verify answers (similar to the TACO pipeline).  We found that this degrades downstream performance. Moreover, in Table 14, we use verifiers for math, science, and code data and find no generalizable improvements from verification, supporting our above results. Thus, our empirical evidence did not support the use of verifiers to filter out answers. This is one of the main counterintuitive results of our paper.
>
> | Models | Average | Code Avg | Math Avg | Science Avg |
> |---|---|---|---|---|
> | OpenThinker-32B | **64.5** | **45.8** | **83.7** | **63.7** |
> | OpenThinker-32B-Unverified | 62.1 | 43.8 | 81.4 | 60.5 |
> | OpenThinker-7B-Unverified | **45.0** | **24.2** | **64.6** | **46.9** |
> | OpenThinker-7B | 41.9 | 21.8 | 62.0 | 41.9 |
>
> | SFT Datasets | Benchmarks | | |
> |---|---|---|---|
> | **Datasets** | **LiveCodeBench** | **CodeElo** | **CodeForces** |
> | Verified via Unit Tests | 36.0 | 9.4 | 10.4 |
> | Unfiltered (Random Sample) | 38.5 | 10.7 | 13.54 |

---

> > ### Author Response · Authors · 2025-11-21
> >
> > > There is seemingly a big gap in performance between the best-performing OpenThoughts3-7B model and the teacher models used (QwQ, DeepSeekReasoner). While this is reasonable given the parameter size difference, the 10-point difference is pretty significant.
> >
> > There is a performance gap between QwQ-32B and DeepSeekReasoner. However, these are much larger models trained by frontier labs, so getting close to that in open source is an impressive accomplishment. We weren’t able to train an OpenThinker3-32B model due to the prohibitive cost of doing so.
> >
> > > Some of the more intriguing results warrant further exploration, like why a less capable teacher model is a better teacher or why different question filtering strategies work for specific domains.
> >
> > The primary goal of the paper is to achieve high performance, and understanding some of the phenomena in more depth is a good direction for future work. We agree that analysis here would be helpful, but we can only suggest possible hypotheses. See Section E.2 for our study of what makes a good teacher model. We analyzed different teacher models across smaller datasets to assess their impact on downstream performance. We found, for example, that Claude’s performance increases linearly with the number of tokens, suggesting that more verbose teachers are better. We tried filtering the teacher annotations in section E.3 and found that compressing or removing “wait” tokens significantly hurt performance. This supports the idea that verbose teachers are better.

---

### Meta-Review · Area_Chair_MW6P · 2026-01-07

**Summary:**

This paper presents a systematic empirical investigation of reasoning data curation through more than 1000 controlled experiments across math, code, and science domains. The resulting OpenThinker3-7B model achieves state-of-the-art performance among open-data 7B reasoning models (53% AIME 2025, 51% LiveCodeBench, 54% GPQA Diamond), with improvements of 15-20 percentage points over existing open baselines. Reviewers appreciated the exceptional experimental rigor, strong results, and comprehensive reproducibility documentation, though raised minor concerns about limited conceptual novelty and lack of deep interpretability for counterintuitive findings.

**Reviewer Concerns:**

The rebuttal successfully addressed most methodological concerns: (1) Experimental rigor issues were resolved by clarifying compute-controlled baselines; (2) Answer filtering conclusions were justified with random filtering baselines; (3) Verifier usage was shown to degrade performance; (4) SFT-only approach was justified for controlled experiments and shown competitive with RL+SFT models; (5) Sampling strategy choices were empirically validated across domains; (6) Significance concerns were reframed by highlighting 30-point gain over base model and 8-point gain over SFT-only baselines. The authors provided specific table references and additional analysis sections for most concerns.

**Reviewer Scores:**

The consensus is positive with all reviewers accepting.

---

### Decision · Program_Chairs · 2026-01-26

Accept (Oral)